# Sampling by averaging: A multiscale approach to score estimation

**Paula Cordero-Encinar**
Department of Mathematics
Imperial College London
paula.cordero-encinar22@imperial.ac.uk

**Andrew B. Duncan**
Department of Mathematics
Imperial College London
a.duncan@imperial.ac.uk

**Sebastian Reich**
Institut für Mathematik
Universität Potsdam
sereich@uni-potsdam.de

**O. Deniz Akyildiz**
Department of Mathematics
Imperial College London
deniz.akyildiz@imperial.ac.uk

## Abstract

We introduce a novel framework for efficient sampling from complex, unnormalised target distributions by exploiting multiscale dynamics. Traditional score-based sampling methods either rely on learned approximations of the score function or involve computationally expensive nested Markov chain Monte Carlo (MCMC) loops. In contrast, the proposed approach leverages stochastic averaging within a slow-fast system of stochastic differential equations (SDEs) to estimate intermediate scores along a diffusion path without training or inner-loop MCMC. Two algorithms are developed under this framework: MULTALMC, which uses multiscale annealed Langevin dynamics, and MULTCDIFF, based on multiscale controlled diffusions for the reverse-time Ornstein-Uhlenbeck process. Both overdamped and underdamped variants are considered, with theoretical guarantees of convergence to the desired diffusion path. The framework is extended to handle heavy-tailed target distributions using Student's t-based noise models and tailored fast-process dynamics. Empirical results across synthetic and real-world benchmarks, including multimodal and high-dimensional distributions, demonstrate that the proposed methods are competitive with existing samplers in terms of accuracy and efficiency, without the need for learned models.

## 1   Introduction

Efficiently sampling from complex unnormalised probability distributions is a fundamental challenge across many scientific domains, including statistics, chemistry, computational physics, and biology, see, e.g. Gelman et al. [2013], Lelièvre et al. [2010], Liu [2008]. Classical Markov Chain Monte Carlo (MCMC) methods provide asymptotically unbiased samples under mild assumptions on the target [Durmus and Moulines, 2017, Mousavi-Hosseini et al., 2023]. However, they often become computationally inefficient in practice, especially for high-dimensional, multimodal targets, due to the need for long Markov chains to ensure adequate mixing and convergence. To alleviate these issues, annealing-based strategies construct a sequence of smoother intermediate distributions bridging a simple base distribution and the target. This principle underlies algorithms such as Parallel Tempering [Geyer and Thompson, 1995, Swendsen and Wang, 1986], Annealed Importance Sampling [Neal, 2001], and Sequential Monte Carlo [Del Moral et al., 2006, Doucet et al., 2001].

Recently, score-based diffusion methods [Hyvärinen, 2005, Song and Ermon, 2020] have emerged as powerful models capable of generating high-quality samples. These approaches simulate a reverse

39th Conference on Neural Information Processing Systems (NeurIPS 2025).

diffusion process guided by time-dependent score functions. While successful in generative modelling, they rely on access to training samples—a setting that differs from sampling problems, where only the unnormalised target density is available. Therefore, traditional score-matching techniques are not applicable. To address this, a growing body of work has explored ways to estimate the score function using only the target density, either through neural networks using variational objectives [Chen et al., 2025, Richter and Berner, 2024, Vargas et al., 2024] or training-free alternatives based on MCMC [Chen et al., 2024, Grenioux et al., 2024, Huang et al., 2024, Phillips et al., 2024, Saremi et al., 2024]. In this work, we present a novel training-free sampling framework that eliminates the need for the inner MCMC loop required in previous works to approximate the score function. Our main contributions are as follows:

- We demonstrate how a multiscale system of SDEs can be leveraged to enable efficient sampling along a diffusion path, obviating the need to estimate the score function.

- In particular, we propose two algorithms: MULTALMC: Multiscale Annealed Langevin Monte Carlo (Section 3.1) and MULTCDIFF: Multiscale Controlled Diffusions (Section 3.2) based on discretisations of two different multiscale SDEs: annealed Langevin dynamics for general noise schedules and the reverse SDE associated with an Ornstein-Uhlenbeck (OU) noising process, respectively. We also explore accelerated versions using underdamped dynamics.

- We provide a theoretical analysis of the proposed methods (Section 4) and demonstrate their effectiveness across different synthetic and real-world benchmarks (Section 5).

## 2 Background and related work

### 2.1 Diffusion paths

The reverse process in diffusion models consists in sampling along a path of probability distributions $(\mu_t)_{t \in [0,1]}$, which starts at a simple distribution and ends at the target distribution $\pi \propto e^{-V_\pi}$ on $\mathbb{R}^d$. Following Chehab and Korba [2024], the intermediate distributions can be expressed as follows

$$\mu_t(x) = \frac{\pi(x/\sqrt{\lambda_t})}{\lambda_t^{d/2}} * \frac{\nu\left(x/\sqrt{1-\lambda_t}\right)}{(1-\lambda_t)^{d/2}}, \qquad (1)$$

where $*$ denotes the convolution operation, $\nu$ describes the base or *noising* distribution, and $\lambda_t$ is an increasing function called schedule, such that, $\lambda_t \in [0,1]$ and $\lambda_1 = 1$. We refer to the path $(\mu_t)_{t \in [0,1]}$ in (1) as the *diffusion path*. By choosing $\lambda_t = e^{-2T(1-t)}$ for some fixed $T$, we recover the path corresponding to a forward OU noising process, a widely used approach in diffusion models [Benton et al., 2024, Chen et al., 2023, Song et al., 2021]. In this case, the reverse-time dynamics can be characterised [Anderson, 1982], requiring only to estimate the score functions $\nabla \log \mu_t$ along the path. However, for general diffusion paths, the reverse process cannot be described by a closed-form SDE due to intractable control terms in the drift, which can be estimated using neural networks as in Albergo et al. [2023]. Alternatively, sampling from the diffusion path can be achieved through *annealed Langevin dynamics* [Cordero-Encinar et al., 2025, Song and Ermon, 2019], which also relies on score estimation but avoids dealing with intractable drift terms.

The diffusion path has demonstrated very good performance in the generative modelling literature where the score of the intermediate distributions can be estimated empirically using score matching techniques [Song and Ermon, 2019]. In the context of sampling, however, estimating the score is more challenging, as samples from the data distribution are not available and instead we only have access to an unnormalised target probability distribution. In particular, under the assumption that $\nu, \pi$ are bounded and have finite second-order moments, and that $|\nabla \log \nu|$ and $|\nabla V_\pi|$ are bounded, the expression of the score of the intermediate distributions is given by

$$\nabla \log \mu_t(x) = \frac{1}{\sqrt{1-\lambda_t}} \mathbb{E}_{\rho_{t,x}(y)}\left[\nabla \log \nu\left(\frac{x-Y}{\sqrt{1-\lambda_t}}\right)\right] = -\frac{1}{\sqrt{\lambda_t}} \mathbb{E}_{\rho_{t,x}(y)}\left[\nabla V_\pi\left(\frac{Y}{\sqrt{\lambda_t}}\right)\right], \qquad (2)$$

where $\rho_{t,x}(y) \propto \nu\left(\frac{x-y}{\sqrt{1-\lambda_t}}\right) e^{-V_\pi(y/\sqrt{\lambda_t})}$ and we have used that $\nabla(f * g) = (\nabla f) * g = f * (\nabla g)$ when $f$ and $g$ are differentiable functions. Notably, in the limits $\lambda_t = 0$ and $\lambda_t = 1$, the conditional distribution $\rho_{t,x}$ converges to a Dirac delta function. The score in (2) involves an expectation over $\rho_{t,x}$, which typically requires MCMC sampling and results in a computationally expensive nested-loop structure. We avoid this by exploiting multiscale dynamics, as described in the following section.

## 2.2 Multiscale dynamics and stochastic averaging

Consider the slow-fast system of SDEs

$$\begin{cases} \mathrm{d}X_t = b(t, X_t, Y_t)\mathrm{d}t + \sqrt{2}\mathrm{d}B_t \\ \mathrm{d}Y_t = \frac{1}{\varepsilon}f(t, X_t, Y_t)\mathrm{d}t + \sqrt{\frac{2}{\varepsilon}}\mathrm{d}\tilde{B}_t \end{cases}$$

where $0 < \varepsilon \ll 1$ controls the scale separation between the slow $X_t$ and the fast component $Y_t$. We assume that, for $\varepsilon = 1$ and fixed $t$ and $X_t$, $Y_t$ is ergodic. As $\varepsilon \to 0$, the fast component $Y_t$ evolves on a much shorter timescale than $X_t$ due to the time-rescaling property of Brownian motion. Intuitively, this means that $Y_t$ rapidly reaches its stationary distribution while $X_t$ remains substantially unchanged. In the limit $\varepsilon \to 0$, the slow-process $X_t$ converges to the averaged dynamics $\bar{X}_t$ which is given by

$$\mathrm{d}\bar{X}_t = \bar{b}(t, \bar{X}_t)\mathrm{d}t + \sqrt{2}\mathrm{d}B_t, \qquad \bar{b}(t, x) = \mathbb{E}_{Y \sim \rho_{t,x}}[b(t, x, Y)],$$

where $\rho_{t,x}$ is the invariant measure of the frozen process $\mathrm{d}Y_s = f(t, x, Y_s)\mathrm{d}s + \sqrt{2}\mathrm{d}\tilde{B}_s$. In computational statistics, stochastic slow-fast systems have been leveraged to simulate the averaged dynamics efficiently [Akyildiz et al., 2024, Harvey et al., 2011, Pavliotis and Stuart, 2008, Weinan et al., 2005]. In our context of sampling from diffusion paths, the averaged drift $\bar{b}(t, x)$ corresponds to the score $\nabla \log \mu_t(x)$, which is defined as an expectation (see Eq. (2)). By leveraging multiscale dynamics, we can approximate this expectation without relying on a nested-loop sampling structure, a key insight that underpins our method, as summarised in Section 3.

## 2.3 Related work

**Diffusion-based samplers** Several recent works have proposed sampling algorithms based on the diffusion path. We divide them into two categories: neural samplers and learning-free samplers. Neural samplers such as those in Chen et al. [2025], Noble et al. [2025], Richter and Berner [2024], Vargas et al. [2024], Zhang and Chen [2022] estimate the time-dependent drift function of the diffusion process using a parametrised model, typically a neural network, by solving a variational inference problem defined over the space of path measures. As in diffusion generative models, their performance is limited by the expressiveness of the chosen model class. In contrast, learning-free samplers [Chen et al., 2024, Grenioux et al., 2024, He et al., 2024b, Huang et al., 2024, Phillips et al., 2024, Saremi et al., 2024] do not require any training similar to our work. Instead, they estimate the score function $\nabla \log \mu_t$ using MCMC samples from the conditional distribution $\rho_{t,x}$. That is, these approaches involve a bi-level structure, with an inner MCMC loop used at each step of the outer diffusion-based sampler. In this sense, they implement a *Langevin-within-Langevin* strategy. Our approach avoids nested loops by leveraging multiscale dynamics along the diffusion path. This results in a *Langevin-by-Langevin* sampler that is both simpler and more computationally efficient.

Besides, while acceleration techniques based on underdamped dynamics [Duncan et al., 2017, Monmarché, 2020] have shown promise in the generative modelling literature [Dockhorn et al., 2022a,b, Holderrieth et al., 2024], and recent work by Blessing et al. [2025] introduces neural underdamped diffusion samplers, no training-free accelerated samplers based on diffusion processes have been proposed. Addressing this gap is one of the key contributions of our work.

**Proximal samplers** Our approach is also closely related to the class of proximal samplers [Chen et al., 2022, Lee et al., 2021b]. These methods define an auxiliary joint distribution of the form $\tilde{\pi}(x, z) \propto \pi(x)\exp(-\|x - z\|^2/(2\eta))$ and perform Gibbs sampling by alternating between the convolutional distribution $\tilde{\pi}(z|x)$ and the denoising posterior $\tilde{\pi}(x|z)$. In practice, sampling from the denoising posterior $\tilde{\pi}(x|z)$ involves finding a mode using gradient-based optimisation techniques, followed by rejection sampling around that mode. This step can be computationally expensive and inefficient, especially in high-dimensional settings. Our method improves upon these approaches by replacing the proximal-point-style posterior sampling step with a fast-timescale dynamics, enabling more efficient exploration of the denoising posterior. Furthermore, unlike proximal samplers that operate at a fixed noise level, our approach targets the entire diffusion path, smoothly interpolating between the base distribution and the target.

# 3 Sampling using multiscale dynamics

A key challenge in sampling from the diffusion path in (1) is accurately estimating the score function when we do not have direct access to samples. Since the score of the intermediate distributions along the diffusion path is computed as an expectation over the conditional distribution $\rho_{t,x}$ (see Eq. (2)), we propose to sample from the time-dependent conditional distribution $\rho_{t,x}$ using an alternative diffusion process with a shorter timescale than the original dynamics following the path in (1). This enables both processes to run in parallel, unlike existing methods that execute them sequentially. In those approaches, each iteration of the original diffusion process (outer loop) requires multiple time steps of the process targeting the conditional distribution $\rho_{t,x}$ (inner loop) to learn the score function, leading to increased computational cost and a higher number of evaluations of the target potential. By employing different timescales for each diffusion process, our method improves sampling efficiency while ensuring convergence to the correct target distribution, as justified by stochastic averaging theory [Liu et al., 2020]. We explore this approach in two settings: (i) when the original diffusion is a Langevin dynamics driven by the scores of the intermediate probability distribution $\nabla \log \mu_t$ (Section 3.1), and (ii) when we use the reverse SDE corresponding to an OU noising process (Section 3.2).

## 3.1 MULTALMC: Multiscale Annealed Langevin Monte Carlo

Following Cordero-Encinar et al. [2025], Song and Ermon [2019], we use a time inhomogeneous Langevin SDE to sample from the diffusion path (1) given by

$$d\bar{X}_t = \nabla \log \hat{\mu}_t(\bar{X}_t)dt + \sqrt{2}dB_t, \quad t \in [0, 1/\kappa], \tag{3}$$

where $\hat{\mu}_t$ denotes the reparametrised probability distributions from the diffusion path ($\hat{\mu}_t = \mu_{\kappa t})_{t \in [0,1/\kappa]}$, for some $0 < \kappa < 1$, $\bar{X}_0 \sim \mu_0 = \nu$ and $(B_t)_{t \geq 0}$ is a Brownian motion. We consider both Gaussian diffusion paths where the base distribution $\nu$ is a Normal distribution and heavy-tailed diffusions corresponding to a Student's t base distribution. Since the scores $\nabla \log \hat{\mu}_t$ are intractable and computed as expectations over $\rho_{t,x}$ (see Eq. (2)), we adopt a multiscale system to approximate the averaged dynamics in (3). The fast process targetting the conditional distribution $\rho_{t,x}$ can follow any fast-mixing dynamics. In particular, we explore the use of overdamped Langevin dynamics or a modified Itô diffusion [He et al., 2024a], but it is important to remark that our framework is more general. When using overdamped Langevin dynamics, the fast process is given by

$$dY_t = \frac{1}{\varepsilon}\nabla \log \hat{\rho}_{t,X_t}(Y_t)dt + \sqrt{\frac{2}{\varepsilon}}d\tilde{B}_t, \quad t \in [0, 1/\kappa],$$

$$\nabla \log \hat{\rho}_{t,X_t}(y) = -\frac{1}{\sqrt{\lambda_{\kappa t}}}\nabla V_\pi\left(\frac{Y_t}{\sqrt{\lambda_{\kappa t}}}\right) + \frac{1}{\sqrt{1-\lambda_{\kappa t}}}\nabla \log \nu\left(\frac{Y_t - X_t}{\sqrt{1-\lambda_{\kappa t}}}\right),$$

where $\hat{\rho}_{t,X_t}$ is a reparametrised version of $\rho_{t,X_t}$ and $(\tilde{B}_t)_{t \geq 0}$ is a Brownian motion on $\mathbb{R}^d$. This suggests sampling from the following stochastic slow-fast system

$$\begin{cases} dX_t &= \begin{cases} \frac{1}{\sqrt{1-\lambda_{\kappa t}}}\nabla \log \nu\left(\frac{X_t - Y_t}{\sqrt{1-\lambda_{\kappa t}}}\right)dt + \sqrt{2}dB_t & \text{if } \lambda_{\kappa t} < \tilde{\lambda} \\ -\frac{1}{\sqrt{\lambda_{\kappa t}}}\nabla V_\pi\left(\frac{Y_t}{\sqrt{\lambda_{\kappa t}}}\right)dt + \sqrt{2}dB_t & \text{if } \lambda_{\kappa t} \geq \tilde{\lambda} \end{cases} \\ dY_t &= \frac{1}{\varepsilon}\left(-\frac{1}{\sqrt{\lambda_{\kappa t}}}\nabla V_\pi\left(\frac{Y_t}{\sqrt{\lambda_{\kappa t}}}\right) + \frac{1}{\sqrt{1-\lambda_{\kappa t}}}\nabla \log \nu\left(\frac{Y_t - X_t}{\sqrt{1-\lambda_{\kappa t}}}\right)\right)dt + \sqrt{\frac{2}{\varepsilon}}d\tilde{B}_t. \end{cases} \tag{4}$$

This system converges to the averaged dynamics described in (3) as $\varepsilon \to 0$, since taking the expectation of the drift term in the $X_t$ dynamics with respect to $Y_t \sim \hat{\rho}_{t,X_t}$ recovers the score function $\nabla \log \hat{\mu}_t$ (see Equation (2) and Section 4). For numerical implementation, we use an adaptive scheme that alternates between the two expressions for $X_t$ depending on the value of $\lambda_{\kappa t}$ in order to avoid numerical instabilities as $\lambda_{\kappa t}$ approaches 0 or 1. The resulting system of SDEs can be discretised using different numerical schemes, however, standard integrators such as Euler-Maruyama (EM) will be unstable when $\varepsilon$ is small, due to the large-scale separation between the slow and fast processes. To mitigate this, we leverage discretisations which remain stable, despite the stiffness of the SDE, such as the exponential integrator [Hochbruck and Ostermann, 2010] and the SROCK method [Abdulle et al., 2018]. Furthermore, since $\rho_{t,x}$ converges to a Dirac delta distribution centred at 0 when $\lambda_{\kappa t} = 0$ and centred at $x$ when $\lambda_{\kappa t} = 1$, we apply a further slight modification to the slow-fast system to improve numerical stability (see App. B.1.1 for details).

**Accelerated MULTALMC**  When implemented in practice, the overdamped system in (4) requires a very small value of $\kappa$ (which corresponds to a slowly varying dynamics driven by $\nabla \log \hat{\mu}_t$) to accurately follow the marginals of the path $(\hat{\mu}_t)_{t \in [0, 1/\kappa]}$ and ultimately sample from the target distribution. This results in a large number of discretisation steps, thus making the method computationally expensive, see App. E.1 for more details. In contrast, underdamped dynamics exhibit faster mixing times [Bou-Rabee and Eberle, 2023, Eberle and Lörler, 2024], which intuitively helps the system explore the space more efficiently and better track the intermediate distributions along the path $(\hat{\mu}_t)_{t \in [0, 1/\kappa]}$. These dynamics have also shown strong empirical performance in the neural sampling literature [Blessing et al., 2025], requiring fewer discretisation steps. Motivated by these advantages, we propose augmenting the state space with auxiliary velocity variables $\bar{V}_t$, and adopting the following underdamped Langevin diffusion to sample from the path $(\hat{\mu}_t)_{t \in [0, 1/\kappa]}$

$$\begin{cases} d\bar{X}_t &= M^{-1}\bar{V}_t dt \\ d\bar{V}_t &= \nabla \log \hat{\mu}_t(\bar{X}_t)dt - \Gamma M^{-1}\bar{V}_t dt + \sqrt{2\Gamma}dB_t. \end{cases} \quad \text{for } t \in [0, 1/\kappa]. \quad (5)$$

The mass parameter $M$ determines the coupling between position $X_t$ and velocity $V_t$, while the friction coefficient $\Gamma$ controls the strength of noise injected into the velocity component. Both $M$ and $\Gamma$ are symmetric positive definite matrices, in practice we consider $M = mId$ and $\Gamma = \gamma Id$. The behaviour of the system is governed by the interplay between $M$ and $\Gamma$ [McCall, 2010]. Based on this, we propose to use the following underdamped slow-fast system to sample from the target distribution

$$\begin{cases} dX_t = & M^{-1}V_t dt \\ dV_t = & \begin{cases} \left( \frac{1}{\sqrt{1-\lambda_{\kappa t}}}\nabla \log \nu \left( \frac{X_t - Y_t}{\sqrt{1-\lambda_{\kappa t}}} \right) - \Gamma M^{-1}V_t \right) dt + \sqrt{2\Gamma}dB_t & \text{if } \lambda_{\kappa t} < \tilde{\lambda} \\ \left( -\frac{1}{\sqrt{\lambda_{\kappa t}}}\nabla V_\pi \left( \frac{Y_t}{\sqrt{\lambda_{\kappa t}}} \right) - \Gamma M^{-1}V_t \right) dt + \sqrt{2\Gamma}dB_t & \text{if } \lambda_{\kappa t} \geq \tilde{\lambda} \end{cases} \\ dY_t = & \frac{1}{\varepsilon}\left( -\frac{1}{\sqrt{\lambda_{\kappa t}}}\nabla V_\pi \left( \frac{Y_t}{\sqrt{\lambda_{\kappa t}}} \right) + \frac{1}{\sqrt{1-\lambda_{\kappa t}}}\nabla \log \nu \left( \frac{Y_t - X_t}{\sqrt{1-\lambda_{\kappa t}}} \right) \right) dt + \sqrt{\frac{2}{\varepsilon}}d\tilde{B}_t. \end{cases} \quad (6)$$

Note that in the underdamped setting, both $X_t, V_t$ are treated as slow variables, and as $\varepsilon \to 0$, this system converges to the averaged dynamics in (5). To implement the system in practice, we combine a symmetric splitting scheme (OBABO) [Monmarché, 2020] for the slow-dynamics, and discretise the fast-dynamics using SROCK method [Abdulle et al., 2018]. We will evaluate the performance of the proposed sampler based on this underdamped slow-fast system (Algorithm 3), referred to as MULTALMC, under two choices of the reference distribution $\nu$: a standard Gaussian and a Student's t distribution. A simplified version of the implementation is presented in Alg. 1, while a more detailed description is provided in App. B.1.2.

**Heavy-tailed diffusions**  When the target distribution has heavy tails, standard Gaussian diffusions often fail to capture the correct tail behaviour [Pandey et al., 2025, Shariatian et al., 2025]. This motivates the use of heavy-tailed noising processes, which have been shown to offer theoretical guarantees in such settings [Cordero-Encinar et al., 2025].

We propose sampling along a heavy-tailed diffusion path $(\hat{\mu}_t)_{t \in [0, 1/\kappa]}$, where the reference distribution $\nu$ is chosen as a Student's t-distribution with tail index $\alpha$, i.e., $\nu \propto (1 + \|x\|^2/\alpha)^{-(\alpha+d)/2}$. This can be implemented using the underdamped slow-fast system defined in (6). However, when the target distribution is heavy-tailed, the use of overdamped Langevin dynamics for the fast component results in slow convergence [Mousavi-Hosseini et al., 2023, Wang, 2006]. This motivates the consideration of alternative diffusion processes for the fast dynamics that mix more efficiently under such conditions. Specifically, we explore a modified Itô diffusion proposed in He et al. [2024a], defined as

$$dY_t = -\frac{1}{\varepsilon}(\alpha + d - 1)\nabla U_{\hat{\rho}_{t,X_t}}(y)dt + \sqrt{\frac{2U_{\hat{\rho}_{t,X_t}}}{\varepsilon}}d\tilde{B}_t, \quad (7)$$

$$\hat{\rho}_{t,X_t}(y) \propto \left( \sqrt{1 + \frac{\|x-y\|^2}{\alpha(1-\lambda_{\kappa t})}} \pi \left( \frac{y}{\sqrt{\lambda_{\kappa t}}} \right)^{-\frac{1}{\alpha+d}} \right)^{-(\alpha+d)} = U_{\hat{\rho}_{t,X_t}}(y)^{-(\alpha+d)}.$$

This modified Itô diffusion for heavy-tailed distributions is shown to have faster convergence guarantees when the target has finite variance [He et al., 2024a]. By replacing the fast process in (6) with the diffusion defined in (7), we obtain samplers better suited for heavy-tailed targets, with improved convergence properties.

### 3.2  MULTCDIFF: Multiscale Controlled Diffusions

Annealed Langevin dynamics (3) offers a convenient approach to sampling from the target distribution for general schedules $\lambda_t$ that interpolate between the base and target distributions in finite time.

---

**Algorithm 1** MULTALMC sampler: accelerated version

---

**Require:** Schedule function $\lambda_t$, value for $\lambda_\delta$ and $\tilde{\lambda}$, friction coefficient $\Gamma$, mass parameter $M$, number of sampling steps $L$, time discretisation $0 = T_0 < \cdots < T_L = 1/\kappa$, step size $h_l = T_{l+1} - T_l$. Constants $\boldsymbol{\mu}, \boldsymbol{\nu}, \boldsymbol{\kappa}$ for SROCK step from (14), (15).

**Initial samples** $X_0 \sim \mathcal{N}(0, I), V_0 \sim \mathcal{N}(0, MI), Y_0 \sim \mathcal{N}(0, I)$, $\lambda'_{\kappa t} = \begin{cases} \lambda_\delta & \text{if } 0 \le \lambda_{\kappa t} < \lambda_\delta \\ \lambda_{\kappa t} & \text{if } \lambda_\delta \le \lambda_{\kappa t} \le 1. \end{cases}$

**for** $l = 0$ **to** $L$ **do**

$\quad \xi_l^{(1)}, \xi_l^{(2)}, \xi_l^{(3)} \sim \mathcal{N}(0, I)$

$\quad$ **if** $0 \le \lambda_{\kappa T_l} < 1 - \lambda_\delta$ **then**
$\qquad$ `Half-step for velocity component`

$$V_l' = \left(1 - \frac{h_l}{2}\Gamma M^{-1}\right) V_l + \sqrt{h_l}\Gamma \xi_l^{(1)}, \qquad V_l'' = \begin{cases} V_l' + \frac{h_l}{2\sqrt{1-\lambda'_{\kappa T_l}}} \nabla \log \nu \left(\frac{X_l - Y_l}{\sqrt{1-\lambda'_{\kappa T_l}}}\right) & \text{if } \lambda'_{\kappa T_l} < \tilde{\lambda} \\ V_l' - \frac{h_l}{2\sqrt{\lambda'_{\kappa T_l}}} \nabla V_\pi \left(\frac{Y_l}{\sqrt{\lambda'_{\kappa T_l}}}\right) & \text{if } \lambda'_{\kappa T_l} \ge \tilde{\lambda} \end{cases}$$

$\qquad$ `Full EM-step for position component` $\quad X_{l+1} = X_l + h_l M^{-1} V_l''$

$\qquad$ `Full SROCK-step for fast component` $\quad Y_{l+1} = f(Y_l, X_{l+1}, \lambda'_{\kappa T_l}, h_l, \varepsilon, \boldsymbol{\mu}, \boldsymbol{\nu}, \boldsymbol{\kappa}) + \sqrt{\frac{2h}{\varepsilon}}\xi_l^{(2)}$

$\qquad$ `Half-step for velocity component`

$$V_l''' = \begin{cases} V_l'' + \frac{h_l}{2\sqrt{1-\lambda'_{\kappa T_l}}} \nabla \log \nu \left(\frac{X_{l+1} - Y_{l+1}}{\sqrt{1-\lambda'_{\kappa T_l}}}\right) & \text{if } \lambda'_{\kappa T_l} < \tilde{\lambda}, \\ V_l'' - \frac{h_l}{2\sqrt{\lambda'_{\kappa T_l}}} \nabla V_\pi \left(\frac{Y_{l+1}}{\sqrt{\lambda'_{\kappa T_l}}}\right) & \text{if } \lambda'_{\kappa T_l} \ge \tilde{\lambda}, \end{cases}, \quad V_{l+1} = \left(1 - \frac{h_l}{2}\Gamma M^{-1}\right) V_l''' + \sqrt{h_l}\Gamma \xi_l^{(3)}$$

$\quad$ **if** $1 - \lambda_\delta \le \lambda_{\kappa T_l} \le 1$ **then**

$\qquad$ `Half-step for velocity component`
$$V_l' = \left(1 - \tfrac{h_l}{2}\Gamma M^{-1}\right) V_l + \sqrt{h_l}\Gamma \xi_l^{(1)}, \qquad V_l'' = V_l' - \tfrac{h_l}{2}\nabla V_\pi(X_l)$$

$\qquad$ `Full EM-step for position component` $\quad X_{l+1} = X_l + h_l M^{-1} V_l''$

$\qquad$ `Half-step for velocity component`
$$V_l''' = V_l'' - \tfrac{h_l}{2}\nabla V_\pi(X_{l+1}), \qquad V_{l+1} = \left(1 - \tfrac{h_l}{2}\Gamma M^{-1}\right) V_l''' + \sqrt{h_l}\Gamma \xi_l^{(3)}$$

**end for**

$\quad$ **return** $(X_L, V_L)$

---

However, a key drawback of annealed Langevin dynamics is that it introduces bias, even if perfectly simulated (see Section 4.2). This bias can be corrected by incorporating control terms. Although such control terms are generally intractable for arbitrary schedules, they can be computed explicitly in the case of an OU noising process. The multiscale approach introduced in the previous section naturally extends to this controlled setting, in both overdamped and underdamped regimes. Given the improved efficiency of underdamped dynamics, we focus our discussion on the underdamped formulation (see App. B.2.1 for details on the overdamped case). When using an OU process as the forward noising model, the underdamped diffusion, initialised at $\overrightarrow{X}_0 \sim \pi$ and $\overrightarrow{V}_0 \sim \mathcal{N}(0, I)$, is defined as

$$\begin{cases} \mathrm{d}\overrightarrow{X}_t = M^{-1}\overrightarrow{V}_t \, \mathrm{d}t \\ \mathrm{d}\overrightarrow{V}_t = \left(-\overrightarrow{X}_t - \Gamma M^{-1}\overrightarrow{V}_t\right) \mathrm{d}t + \sqrt{2\Gamma} \, \mathrm{d}B_t'. \end{cases}$$

In this case, to ensure efficient and stable dynamics, we adopt the critical damping regime, where $\Gamma^2 = 4M$. This choice leads to fast convergence without oscillations [McCall, 2010]. The corresponding time-reversed SDE is

$$\begin{cases} \mathrm{d}\overleftarrow{X}_t = -4\Gamma^{-2}\overleftarrow{V}_t \, \mathrm{d}t \\ \mathrm{d}\overleftarrow{V}_t = \left(\overleftarrow{X}_t + 4\Gamma^{-1}\overleftarrow{V}_t + 2\Gamma\nabla_v \log q_{T-t}(\overleftarrow{X}_t, \overleftarrow{V}_t)\right) \mathrm{d}t + \sqrt{2\Gamma} \, \mathrm{d}B_t, \end{cases} \tag{8}$$

where $q_t$ is the marginal distribution of the forward process which has the following expression

$$q_t(x, v) = \int \pi(y)\varphi(v_0)\rho_t(x, v|y, v_0) \, \mathrm{d}y \, \mathrm{d}v_0,$$

with $\varphi = \mathcal{N}(0, I)$ and conditional distribution $\rho_t(x, v|y, v_0)$ normally distributed with mean $m_t(y, v_0)$ and covariance $\Sigma_t$ given by

$$m_t(y, v_0) = e^{At} \otimes I \begin{pmatrix} y \\ v_0 \end{pmatrix}, \quad \Sigma_t = \int_0^t e^{A(t-s)} \begin{pmatrix} 0 & 0 \\ 0 & 2\Gamma \end{pmatrix} \left[ e^{A(t-s)} \right]^\intercal \, ds \otimes I, \quad A = \begin{pmatrix} 0 & 4\Gamma^{-2} \\ -1 & -4\Gamma^{-1} \end{pmatrix}.$$

We observe that $v_0$ can be analytically integrated out in the expression for $\rho_t(x, v|y, v_0)$. Moreover, the conditional distribution $\rho_t(v|x, y)$ remains Gaussian, with mean $\tilde{m}_t(x, y)$ and covariance $\sigma_t^2 I$, where $\tilde{m}_t(x, y)$ is linear in $y$. This structure allows us to express the score $\nabla_v \log q_t(x, v)$ as

$$\nabla_v \log q_t(x, v) = \nabla_v \log q_t(v|x) = -\frac{v - \mathbb{E}_{Y|x,v}[\tilde{m}_t(x, Y)]}{\sigma_t^2} = -\frac{v - f_t\left(x, \mathbb{E}_{Y|x,v}[Y]\right)}{\sigma_t^2}, \tag{9}$$

where $f_t$ is a linear function of its arguments, see App. B.2 for a detailed derivation. Using the expressions above, we define a multiscale SDE system that enables sampling from the target distribution via the reverse SDE (8), without requiring prior estimation of the denoiser $\mathbb{E}_{Y|x,v}[Y]$ that appears in the expression for $\nabla_v \log q_{T-t}(x, v)$ (9). In this formulation, the fast process is modelled by an overdamped Langevin SDE that targets the conditional distribution $Y \mid (\overleftarrow{X}_t, \overleftarrow{V}_t)$, given by

$$dY_t = \frac{1}{\varepsilon}\left(\nabla \log \pi(Y_t) + \nabla_{Y_t} \log \rho_{T-t}(\overleftarrow{X}_t, \overleftarrow{V}_t|Y_t)\right) dt + \sqrt{\frac{2}{\varepsilon}} d\tilde{B}_t.$$

Combining this fast process with the reverse SDE, we obtain the following stochastic slow-fast system, which we use for sampling from the target distribution

$$\begin{cases} d\overleftarrow{X}_t = -4\Gamma^{-2}\overleftarrow{V}_t \, dt \\ d\overleftarrow{V}_t = \left(\overleftarrow{X}_t + 4\Gamma^{-1}\overleftarrow{V}_t - \frac{2\Gamma}{\sigma_{T-t}^2}\left(\overleftarrow{V}_t - f_{T-t}(\overleftarrow{X}_t, Y_t)\right)\right) dt + \sqrt{2\Gamma} \, dB_t \\ dY_t = \frac{1}{\varepsilon}\left(\nabla \log \pi(Y_t) + \nabla_{Y_t} \log \rho_{T-t}(\overleftarrow{X}_t, \overleftarrow{V}_t|Y_t)\right) dt + \sqrt{\frac{2}{\varepsilon}} d\tilde{B}_t. \end{cases} \tag{10}$$

To implement this sampler, referred to as MULTCDIFF, we construct a novel integrator inspired by Dockhorn et al. [2022b]. Specifically, we leverage the symmetric Trotter splitting and the Baker–Campbell–Hausdorff formula [Strang, 1968, Trotter, 1959, Tuckerman, 2010] to design a stable and efficient symmetric splitting scheme, see App. B.2 for details. We also outline in the appendix the difficulties of extending controlled diffusions to the heavy-tailed scenario.

## 4  Theoretical guarantees

In this section, we identify the different sources of error in the proposed sampling algorithms. For MULTALMC, the total error consists of three components: the discretisation error from numerically solving the slow-fast system, the convergence error of the slow-fast system to its averaged dynamics, and the bias arising from the time-inhomogeneous averaged system, whose marginals differ from those of the diffusion path in (1)—this last component is discussed in 4.2. For MULTCDIFF, the error bound includes an initialisation error present in the error bounds of diffusion models [Benton et al., 2024, Chen et al., 2023], the discretisation error of the slow-fast system, and the convergence of the multiscale system to the averaged dynamics. The discretisation error depends on the choice of numerical integrator, underscoring the importance of an accurate and stable integrator scheme. We do not analyse this error in detail as it follows established methods [Leimkuhler et al., 2024, Monmarché, 2020]. We now study the convergence properties of the slow component $X_t$ to the solution of the corresponding averaged dynamics that evolves along the time-inhomogeneous diffusion path in (1).

### 4.1  Convergence of the slow-fast system to the averaged dynamics

Building on results from stochastic averaging theory [Liu et al., 2020], our goal is to derive sufficient conditions on the coefficients of the multiscale systems—namely, (4), (6), and (10)—that guarantee strong convergence of the slow component to the averaged process as $\varepsilon \to 0$. We focus on the case where the base distribution is Gaussian $\nu \sim \mathcal{N}(0, I)$. Extending the convergence analysis to the case where $\nu$ is a heavy-tailed distribution, such as a Student's t distribution, presents significant technical challenges, which fall outside of the scope of this work. We outline these challenges in App. C.2. To establish convergence guarantees, we assume the following regularity conditions.

**A1.** *The target distribution $\pi$ has density with respect to the Lebesgue measure, which we write $\pi \propto e^{-V_\pi}$. The potential $V_\pi$ has Lipschitz continuous gradients, with Lipschitz constant $L_\pi$. In addition, $V_\pi$ is strongly convex with convexity parameter $M_\pi > 0$.*

**A2.** *The schedule* $\lambda_t : \mathbb{R}^+ \to [0,1]$ *is a non-decreasing function of* $t$, *weakly differentiable and Hölder continuous with exponent* $\gamma_1$ *in* $(0,1]$.

Under these assumptions, we obtain the following result.

**Theorem 4.1.** *Let the base distribution* $\nu \sim \mathcal{N}(0, I)$. *Suppose the target distribution* $\pi$ *and the schedule function* $\lambda_t$ *satisfy assumptions A1 and A2, respectively. Then, for any* $\varepsilon \in (0, \varepsilon_0)$, *where* $\varepsilon_0$ *is specified in App. C.1, and any given initial conditions, there exists unique solutions* $\{(X_t^\varepsilon, Y_t^\varepsilon), t \geq 0\}$, $\{(X_t^\varepsilon, V_t^\varepsilon, Y_t^\varepsilon), t \geq 0\}$ *and* $\{(\overleftarrow{X}_t^\varepsilon, \overleftarrow{V}_t^\varepsilon, Y_t^\varepsilon), t \geq 0\}$ *to the slow-fast stochastic systems* (4), (6) *and* (10), *respectively. Furthermore, for any* $p > 0$, *it holds that*

$$\lim_{\varepsilon \to 0} \mathbb{E} \left( \sup_{t \in [0,T]} \| X_t^\varepsilon - \bar{X}_t \|^p \right) = 0,$$

*where* $\bar{X}_t$ *denotes the solution of the averaged system and* $X_t^\varepsilon$ *is the corresponding slow component of the multiscale systems* (4), (6) *and* (10).

The proof is provided in App. C.1. The theorem applies to both the annealed Langevin dynamics and the controlled diffusion. We note that the strong convexity condition in **A**1—which guarantees exponential convergence of the fast process to its unique invariant measure when $\varepsilon = 1$ is fixed—is a restrictive assumption. However, as shown in Section 5, our proposed algorithms show strong empirical performance on benchmark distributions that do not satisfy this condition. Extending the theoretical analysis to cover non-log-concave targets remains an important future direction.

### 4.2 Bias of the annealed Langevin dynamics

The bias of the overdamped annealed Langevin dynamics (3) has been studied in prior work [Cordero-Encinar et al., 2025, Guo et al., 2025]. Here, we focus on quantifying the bias introduced by the underdamped averaged system (5) relative to the true diffusion path in the augmented state space with the velocity. At fixed time $t$, the invariant distribution of the underdamped averaged system takes the form

$$\hat{p}_t(x, v) \propto \exp \left( -\frac{1}{2} v^\mathsf{T} M^{-1} v + \log \hat{\mu}_t(x) \right). \tag{11}$$

Note that the Hamiltonian is separable, meaning that $x$ and $v$ are independent. It is important to emphasise that, even when simulated exactly, diffusion annealed Langevin dynamics introduces a bias, as the marginal distributions of the solution of (5) do not exactly match the prescribed distributions $(\hat{p}_t)_t$. A key quantity for characterising this bias will be the action of the curve of probability measures $\mu = (\mu_t)_{t \in [0,1]}$ defined in (1) interpolating between the base distribution $\nu$ and the target distribution $\pi$, denoted by $\mathcal{A}(\mu)$. As noted by Guo et al. [2025], the action serves as a measure of the cost of transporting $\nu$ to $\pi$ along the specified path. Formally, for an absolutely continuous curve of probability measures [Lisini, 2007] with finite second-order moment, the action is given by

$$\mathcal{A}(\mu) := \int_0^T \left( \lim_{\delta \to 0} |\delta|^{-1} W_2(\mu_{t+\delta}, \mu_t) \right)^2 \mathrm{d}t.$$

Based on Theorem 1 from Guo et al. [2025], we have the following result.

**Theorem 4.2.** *Let* $\mathbb{Q}_{\text{U-ALD}} = (q_{t,\text{U-ALD}})_{t \in [0, 1/\kappa]}$ *be the path measure of the diffusion annealed Langevin dynamics* (5), *and* $\mathbb{Q} = (\hat{p}_t)_{t \in [0, 1/\kappa]}$ *that of a reference SDE such that the marginals at each time have distribution* $\hat{p}_t$ (11). *If* $q_{0,\text{U-ALD}} = \hat{p}_0$, *the KL divergence between the path measures is given by*

$$\text{KL} \left( \mathbb{Q} \, \| \mathbb{Q}_{\text{U-ALD}} \right) = \kappa \mathcal{A}(\mu)/4.$$

The action $\mathcal{A}(\mu)$ is bounded when the target $\pi$ has bounded second order moment and under mild conditions on the schedule function, see Cordero-Encinar et al. [2025, Lemmas 3.3 and 4.2]. Full details and a complete proof are provided in App. C.3.

## 5 Numerical experiments

We now evaluate the performance of both proposed underdamped samplers, based on the annealed Langevin dynamics (Section 3.1) and the controlled diffusions formulations (Section 3.2) across a range of benchmark distributions. Full details of each benchmark are provided in App. D.

- **Mixture of Gaussians (MoG) and mixture of Student's t (MoS) distributions**.

- **Rings:** Two-dimensional distribution supported on a circular manifold.

- **Funnel:** 10-dimensional distribution defined by $\pi(x) \propto \mathcal{N}(x_1; 0, \eta^2) \prod_{i=2}^{d} \mathcal{N}(x_i; 0, e^{x_1})$ for $x = (x_i)_{i=1}^{10} \in \mathbb{R}^{10}$ with $\eta = 3$ [Neal, 2003].

- **Double well potential (DW):** $d$-dimensional distribution given by $\pi \propto \exp(-\sum_{i=1}^{m}(x_i^2 - \delta)^2 - \sum_{i=m+1}^{d} x_i^2)$ with $m \in \mathbb{N}$ and $\delta \in (0, \infty)$. Ground truth samples are obtained using rejection sampling with a Gaussian mixture proposal distribution [Midgley et al., 2023].

- **Examples from Bayesian statistics:** Posterior distributions arising from Bayesian logistic regression tasks on the Ionosphere (dimension 35) and Sonar (dimension 61) datasets.

- **Statistical physics model:** Sampling metastable states of the stochastic Allen-Cahn equation $\phi^4$ model (dimension 100) [Albergo et al., 2019, Gabrié et al., 2022]. At the chosen temperature, the distribution has two well distinct modes with relative weight controlled by a parameter $h$. Following Grenioux et al. [2024], we estimate the relative weight between the two modes.

While we focus here on the underdamped versions of our algorithms, we have also evaluated their overdamped counterparts. These approximately require an order of magnitude more steps to achieve comparable performance. Detailed results and comparisons are presented in App. E.1. Besides, we have set the base distribution, denoted as $\nu$ in our algorithms, to be a standard Gaussian. In the case of the mixture of Student's t benchmark distributions, we compare the performance of using either a standard Gaussian or a Student's t base distribution. For the latter, we further examine the impact of using a modified Itô diffusion for the fast dynamics (7), as opposed to a standard Langevin diffusion.

We compare our approach against a representative selection of related sampling methods: Sequential Monte Carlo (SMC) [Del Moral et al., 2006, Doucet et al., 2001], Annealed Importance Sampling (AIS) [Neal, 2001], Underdamped Langevin Monte Carlo (ULMC) [Cheng et al., 2018, Neal et al., 2011], Parallel Tempering (PT) [Geyer and Thompson, 1995, Swendsen and Wang, 1986], Diffusive Gibbs Sampling (DiGS) [Chen et al., 2024], Reverse Diffusion Monte Carlo (RDMC) [Huang et al., 2024] and Stochastic Localisation via Iterative Posterior Sampling (SLIPS) [Grenioux et al., 2024]. We ensure a fair comparison by using the same number of energy evaluations and the same initialisation based on a standard Gaussian distribution for all baselines. For completeness, we also report the performance of SLIPS with optimal initialisation (App. D)—which assumes knowledge of the distribution's scalar variance, unavailable in practice. For the $\phi^4$ model, we only compare our results against a Laplace approximation and SLIPS algorithm, as the other methods provide degenerate samples. Since our approach is learning-free, we exclude comparisons with neural-based samplers as it would be difficult to make comparisons at equal computational budget.

Our methods consistently achieve strong performance across all benchmarks, outperforming all baselines. They also exhibit low variance in results (see Tables 1 and 2). Among our samplers, the one based on controlled diffusions, MULTCDIFF, generally achieves slightly better results than the annealed Langevin-based sampler, MULTALMC. Additionally, Figure 1a illustrates the estimation of the relative weight between the two modes for the $\phi^4$ model, where our algorithms demonstrate

Table 1: Metrics for different benchmarks averaged across 30 seeds. The metric for the mixture of Gaussian (MoG) and Rings is the entropy regularised Wasserstein-2 distance (with regularisation parameter 0.05) and the metric for the Funnel is the sliced Kolmogorov-Smirnov distance.

| Algorithm | 8-MoG ($\downarrow$) $(d=2)$ | 40-MoG ($\downarrow$) $(d=2)$ | 40-MoG ($\downarrow$) $(d=50)$ | Rings ($\downarrow$) $(d=2)$ | Funnel ($\downarrow$) $(d=10)$ |
|---|---|---|---|---|---|
| SMC | $5.26 \pm 0.19$ | $4.79 \pm 0.16$ | $52.17 \pm 1.32$ | $0.29 \pm 0.05$ | $0.034 \pm 0.005$ |
| AIS | $5.53 \pm 0.23$ | $5.01 \pm 0.20$ | $65.83 \pm 1.66$ | $0.20 \pm 0.03$ | $0.040 \pm 0.006$ |
| ULMC | $6.90 \pm 0.28$ | $8.17 \pm 0.72$ | $109.26 \pm 2.90$ | $0.35 \pm 0.06$ | $0.123 \pm 0.011$ |
| PT | $3.91 \pm 0.10$ | $0.92 \pm 0.08$ | $21.04 \pm 0.55$ | $0.20 \pm 0.04$ | $0.033 \pm 0.004$ |
| DiGS | $1.22 \pm 0.09$ | $0.95 \pm 0.03$ | $21.87 \pm 0.97$ | $0.20 \pm 0.02$ | $0.037 \pm 0.005$ |
| RDMC | $1.01 \pm 0.21$ | $0.98 \pm 0.07$ | $36.10 \pm 0.62$ | $0.33 \pm 0.01$ | $0.080 \pm 0.007$ |
| SLIPS | $1.15 \pm 0.12$ | $1.04 \pm 0.06$ | $23.71 \pm 0.65$ | $0.26 \pm 0.01$ | $0.039 \pm 0.006$ |
| MULTALMC | $0.65 \pm 0.07$ | $\mathbf{0.91 \pm 0.04}$ | $20.58 \pm 0.71$ | $\mathbf{0.18 \pm 0.02}$ | $0.032 \pm 0.005$ |
| MULTCDIFF | $\mathbf{0.62 \pm 0.10}$ | $0.93 \pm 0.06$ | $\mathbf{20.13 \pm 0.59}$ | $0.19 \pm 0.03$ | $\mathbf{0.031 \pm 0.005}$ |

Table 2: Performance metrics for different sampling benchmarks averaged across 30 seeds. The metric for the double well potential (DW) is the entropy regularised Wasserstein-2 distance (with regularisation parameter 0.05) and the metric for the Bayesian logistic regression on Ionosphere and Sonar datasets is the average predictive posterior log-likelihood on a test dataset.

| Algorithm | 5-dim DW ($\downarrow$) $(m = 5, \delta = 4)$ | 10-dim DW ($\downarrow$) $(m = 5, \delta = 3)$ | 50-dim DW ($\downarrow$) $(m = 5, \delta = 2)$ | Ionosphere ($\uparrow$) $(d = 35)$ | Sonar ($\uparrow$) $(d = 61)$ |
|---|---|---|---|---|---|
| SMC | $4.06 \pm 0.13$ | $11.86 \pm 0.70$ | $28.92 \pm 0.99$ | $-87.74 \pm 0.10$ | $-111.00 \pm 0.11$ |
| AIS | $4.11 \pm 0.17$ | $12.30 \pm 0.61$ | $39.20 \pm 1.07$ | $-88.11 \pm 0.13$ | $-111.15 \pm 0.17$ |
| ULMC | $7.87 \pm 0.51$ | $25.21 \pm 1.00$ | $62.74 \pm 1.52$ | $-116.37 \pm 1.85$ | $-173.61 \pm 2.48$ |
| PT | $2.39 \pm 0.20$ | $4.97 \pm 0.43$ | $16.13 \pm 0.89$ | $-87.91 \pm 0.09$ | $-112.99 \pm 0.10$ |
| DiGS | $2.51 \pm 0.18$ | $5.02 \pm 0.42$ | $17.04 \pm 0.95$ | $-88.02 \pm 0.12$ | $-110.53 \pm 0.25$ |
| RDMC | $3.68 \pm 0.32$ | $9.57 \pm 0.75$ | $25.21 \pm 0.73$ | $-108.44 \pm 1.13$ | $-130.20 \pm 1.36$ |
| SLIPS | $2.46 \pm 0.21$ | $5.09 \pm 0.39$ | $18.15 \pm 0.50$ | $-87.34 \pm 0.11$ | $-110.14 \pm 0.10$ |
| MULTALMC | $2.08 \pm 0.16$ | $4.48 \pm 0.40$ | $14.03 \pm 0.63$ | $-86.85 \pm 0.09$ | $\mathbf{-109.05 \pm 0.13}$ |
| MULTCDIFF | $\mathbf{1.95 \pm 0.24}$ | $\mathbf{4.23 \pm 0.37}$ | $\mathbf{13.98 \pm 0.56}$ | $\mathbf{-86.33 \pm 0.10}$ | $-109.60 \pm 0.21$ |

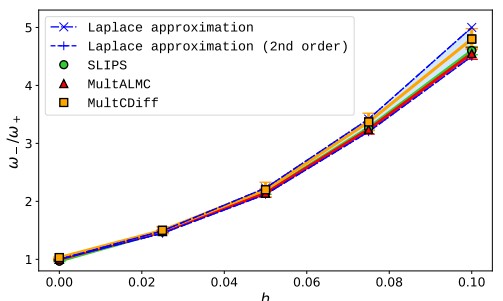

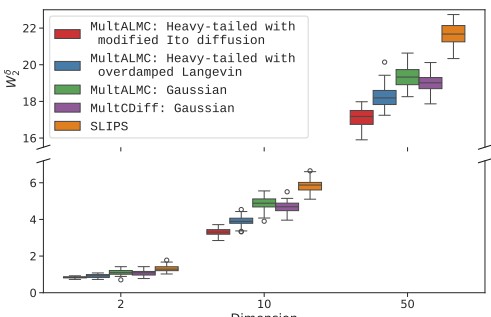

(a) Estimated mode weight ratio of $\phi^4$ for different $h$.  (b) Regularised $W_2$ for MoS in different dimensions.

Figure 1: Results for sampling benchmarks.

competitive performance. For the mixture of Student's t-distributions, Figure 1b shows that using the modified Itô diffusion for the fast dynamics, combined with a Student's t base distribution $\nu$ (red boxplots) outperforms both our other proposed algorithms and the baselines. Further experimental details and additional results are provided in Appendices D and E.

## 6 Discussion

In this work, we introduce a framework based on multiscale diffusions for sampling from an unnormalised target density. In particular, we propose two samplers, MULTALMC and MULTCDIFF depending on the dynamics used for the slow process: annealed Langevin dynamics or controlled diffusions, respectively. We establish theoretical guarantees for the convergence of these sampling algorithms and illustrate their performance on a range of high-dimensional benchmark distributions.

Our approach has certain limitations. Notably, the theoretical guarantees rely on stringent assumptions, which we aim to relax in future work. Additionally, the current method requires manual tuning of hyperparameters such as step size $\delta$, scale separation parameters $\varepsilon$ and friction coefficient $\Gamma$. Automating this tuning process remains an important direction for future research. Further research could explore extending the controlled diffusion framework to heavy-tailed target distributions, which pose additional challenges or developing more efficient numerical schemes for implementing the proposed multiscale samplers.

## Acknowledgements

PCE gratefully acknowledges support from the EPSRC through the Centre for Doctoral Training in Modern Statistics and Statistical Machine Learning (StatML), grant no. EP/S023151/1. SR's work was partially funded by the Deutsche Forschungsgemeinschaft (DFG) under Project-ID 318763901 – SFB1294.

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

# A   Preliminaries

**Multiscale methods and stochastic averaging.**   Before presenting a formal definition, we first provide the intuition behind multiscale methods, following Schuh and Souttar [2024]. In many areas of science and engineering [Bertram and Rubin, 2017, Weinan et al., 2005], one often encounters systems that are difficult to analyse directly. However, there may exist a related system that is more tractable, though it does not produce exactly the same behaviour. In such cases, the simpler system can be viewed as an approximation of the more complex one. In our setting, the simpler system is a slow-fast system, while the complex system corresponds to the averaged dynamics. We develop these concepts in the following.

To conduct a theoretical analysis of the convergence of the proposed sampling algorithms, we examine the general stochastic slow-fast system studied in Liu et al. [2020]

$$\begin{cases} \mathrm{d}X_t^\varepsilon = b(t, X_t^\varepsilon, Y_t^\varepsilon)\mathrm{d}t + \sigma(t, X_t^\varepsilon)\mathrm{d}B_t & X_0^\varepsilon = x \in \mathbb{R}^n \\ \mathrm{d}Y_t^\varepsilon = \frac{1}{\varepsilon}f(t, X_t^\varepsilon, Y_t^\varepsilon)\mathrm{d}t + \frac{1}{\sqrt{\varepsilon}}g(t, X_t^\varepsilon, Y_t^\varepsilon)\mathrm{d}\tilde{B}_t & Y_0^\varepsilon = y \in \mathbb{R}^m, \end{cases} \tag{12}$$

where $\varepsilon$ is a small positive parameter describing the ratio of time scales between the slow component $X_t^\varepsilon$ and the fast component $Y_t^\varepsilon$, and $B_t$ and $\tilde{B}_t$ are mutually independent standard Brownian motions on a complete probability space $(\Omega, \mathcal{F}, \mathbb{P})$ and $\{\mathcal{F}_t, t \geq 0\}$ is the natural filtration generated by $B_t$ and $\tilde{B}_t$. Let $\bar{X}_t$ denote the solution of the following averaged equation

$$\begin{cases} \mathrm{d}\bar{X}_t = \bar{b}(t, \bar{X}_t)\mathrm{d}t + \sigma(t, \bar{X}_t)\mathrm{d}B_t, \\ \bar{X}_0 = x, \end{cases}$$

where $\bar{b}(t, x) = \int_{\mathbb{R}^m} b(t, x, y)\rho_{t,x}(\mathrm{d}y)$ and $\rho_{t,x}$ denotes the unique invariant measure for the transition semigroup of the corresponding frozen process, which can be informally defined as the fast process with $\varepsilon = 1$ and fixed slow dynamics $X_t^\varepsilon = x$,

$$\begin{cases} \mathrm{d}Y_s = f(t, x, Y_s)\mathrm{d}s + g(t, x, Y_s)\mathrm{d}\tilde{B}_s' \\ Y_0 = y, \end{cases}$$

where $\tilde{B}_s'$ is a Brownian motion on another complete probability space.

We begin by presenting the assumptions underlying the analysis in Liu et al. [2020] and stating their main results, which will form the basis of our study in Appendix C.1.

The assumptions on the coefficients of the stochastic slow-fast system (12) are the following.

**A3** (Coefficients of the slow process). *(i) There exists $\theta_1 \geq 0$ such that for any $t, R \geq 0$, $x_i \in \mathbb{R}^n$, $y \in \mathbb{R}^m$ with $\|x_i\| \leq R$,*

$$2\|b(t, x_1, y) - b(t, x_2, y)\|\|x_1 - x_2\| + \|\sigma(t, x_1, y) - \sigma(t, x_2, y)\|^2 \leq K_t(R)(1 + \|y\|^{\theta_1})\|x_1 - x_2\|^2,$$

*where $K_t(R)$ is an $\mathbb{R}_+$-valued $\mathcal{F}_t$-adapted process satisfying for all $R, T, p \in [0, \infty)$,*

$$\alpha_T(R) := \int_0^T K_t(R)\mathrm{d}t < \infty, \quad on\ \Omega, \quad \mathbb{E}\left[e^{p\alpha_T(1)}\right] < \infty, \quad \sup_{t \in [0,T]} \mathbb{E}\|K_t(1)\|^4 < \infty.$$

*Furthermore, there exists $R_0 > 0$, such that for any $R \geq R_0$, $T \geq 0$,*

$$\mathbb{E}\int_0^T [K_t(R)]^4\, \mathrm{d}t < \infty.$$

*(ii) There exist constants $\theta_2, \theta_3 \geq 1$ and $\gamma_1 \in (0, 1]$ such that for any $x \in \mathbb{R}^n$, $y, y_1, y_2 \in \mathbb{R}^m$ and $T > 0$ with $t, s \in [0, T]$,*

$$\|b(t, x, y_1) - b(t, x, y_2)\| \leq C_T\|y_1 - y_2\| \left[\|y_1\|^{\theta_2} + \|y_2\|^{\theta_2} + K_t(1) + \|x\|^{\theta_3}\right]$$

*and*

$$\|b(t, x, y) - b(s, x, y)\| \leq C_T\|t - s\|^{\gamma_1} \left[\|y\|^{\theta_2} + \|x\|^{\theta_3} + Z_T\right], \quad on\ \Omega,$$

*where $C_T > 0$ and $Z_t$ is some random variable satisfying $\mathbb{E}Z_T^2 < \infty$.*

*(iii) There exist $\lambda_1 \geq 0, C > 0, \theta_4 \geq 2$ and $\theta_5, \theta_6 \geq 1$ such that for any $t > 0$, $x \in \mathbb{R}^n$, $y \in \mathbb{R}^m$,*

$$2\langle x, b(t,x,y)\rangle \leq K_t(1)(1 + \|x\|^2) + \lambda_1 \|y\|^{\theta_4},$$

*and*

$$\|b(t,x,y)\| \leq K_t(1) + C(\|x\|^{\theta_5} + \|y\|^{\theta_6}), \quad \|\sigma(t,x)\|^2 \leq K_t(1) + C\|x\|^2.$$

**A4** (Coefficients of the fast process). *(i) There exists $\beta \geq 0$ such that for any $t \geq 0$, $x \in \mathbb{R}^n$, $y_1, y_2 \in \mathbb{R}^m$,*

$$2\langle f(t,x,y_1) - f(t,x,y_2), y_1 - y_2\rangle + \|g(t,x,y_1) - g(t,x,y_2)\|^2 \leq -\beta\|y_1 - y_2\|^2.$$

*(ii) For any $T > 0$, there exists $\gamma_2 \in (0,1]$, $C_T > 0$, $\alpha_i \geq 1, i = 1,2,3,4$ such that for any $t, s \in [0, T]$ and $x_i \in \mathbb{R}^n$, $y_i \in \mathbb{R}^m$, $i = 1,2$,*

$$\|f(t,x_1,y_1) - f(s,x_2,y_1)\| \leq C_T(|t - s|^{\gamma_2} + \|x_1 - x_2\|)(1 + \|x_1\|^{\alpha_1} + \|x_2\|^{\alpha_1} + \|y_1\|^{\alpha_2}),$$

$$\|g(t,x_1,y_1) - g(s,x_2,y_2)\| \leq C_T(|t - s|^{\gamma_2} + \|x_1 - x_2\| + \|y_1 - y_2\|),$$

$$\|f(t,x_1,y_1)\| \leq C_T(1 + \|x_1\|^{\alpha_3} + \|y_1\|^{\alpha_4}),$$

$$\|g(t,x_1,y_1)\| \leq C_T(1 + \|x_1\| + \|y_1\|).$$

*(iii) For some fixed $k \geq 2$ and any $T > 0$, there exist $C_{T,k}, \beta_k > 0$ such that for any $t \in [0,T]$, $x \in \mathbb{R}^n$, $y \in \mathbb{R}^m$,*

$$2\langle y, f(t,x,y)\rangle + (k-1)\|g(t,x,y)\|^2 \leq -\beta_k\|y\|^2 - \lambda_2\|y\|^{\theta_4} + C_{T,k}\left(\|x\|^{\frac{4}{\theta_4}} + 1\right),$$

*where $\lambda_2 = 0$ if $\lambda_1 = 0$, and $\lambda_2 > 0$ otherwise.*

Note that assumption **A**4 on the coefficients of the fast process guarantees that the frozen process has a unique stationary measure and that the fast process converges to it exponentially fast. The following theorems establish the existence and uniqueness of solutions to the system in (12), as well as its convergence to the corresponding averaged dynamics.

**Theorem A.1** (Theorem 2.2 [Liu et al., 2020]). *If assumptions **A**3 and **A**4 hold with $\lambda_1 > 0$, let $\varepsilon_0 = \frac{\lambda_2}{\lambda_1}$. Then for any $\varepsilon \in (0, \varepsilon_0)$ and any given initial values $x \in \mathbb{R}^n$, $y \in \mathbb{R}^m$, there exists a unique solution $\{(X_t^\varepsilon, Y_t^\varepsilon), t \geq 0\}$ to the system (12) and for all $T > 0$, we have $(X_t^\varepsilon, Y_t^\varepsilon) \in C([0,T]; \mathbb{R}^n) \times C([0,T]; \mathbb{R}^m)$, $\mathbb{P}$-almost surely. Furthermore, for all $t \in [0,T]$, the solution $(X_t^\varepsilon, Y_t^\varepsilon)$ is given by*

$$\begin{cases} X_t^\varepsilon = x + \int_0^t b(s, X_s^\varepsilon, Y_s^\varepsilon)\mathrm{d}s + \int_0^t \sigma(s, X_s^\varepsilon)\mathrm{d}B_s \\ Y_t^\varepsilon = y + \frac{1}{\varepsilon}\int_0^t f(s, X_s^\varepsilon, Y_s^\varepsilon)\mathrm{d}s + \frac{1}{\sqrt{\varepsilon}}\int_0^t g(s, X_s^\varepsilon, Y_s^\varepsilon)\mathrm{d}\tilde{B}_s. \end{cases}$$

**Theorem A.2** (Theorem 2.3 [Liu et al., 2020]). *If assumptions **A**3 and **A**4 hold with $\lambda_1 > 0$ and $k > \tilde{\theta}_2$ where $\tilde{\theta}_2 = \max\{4\theta_1, 2\theta_2 + 2, 2\theta_6, 4\alpha_2, \theta_4\theta_5, 2\alpha_1\theta_4\}$. Then, for any $0 < p < \frac{2k}{\theta_4}$ we have*

$$\lim_{\varepsilon \to 0} \mathbb{E}\left(\sup_{t \in [0,T]} \|X_t^\varepsilon - \bar{X}_t\|^2\right) = 0.$$

**Optimal transport.** Let $v = (v_t : \mathbb{R}^d \to \mathbb{R}^d)$ be a vector field and $\mu = (\mu_t)_{t \in [a,b]}$ be a curve of probability measures on $\mathbb{R}^d$ with finite second-order moments. $\mu$ is generated by the vector field $v$ if the continuity equation

$$\partial_t \mu_t + \nabla \cdot (\mu_t v_t) = 0,$$

holds for all $t \in [a, b]$. The metric derivative of $\mu$ at $t \in [a, b]$ is then defined as

$$|\dot{\mu}|_t := \lim_{\delta \to 0} \frac{W_2(\mu_{t+\delta}, \mu_t)}{|\delta|}.$$

If $|\dot{\mu}|_t$ exists and is finite for all $t \in [a, b]$, we say that $\mu$ is an absolutely continuous curve of probability measures. Ambrosio and Kirchheim [2000] establish weak conditions under which a curve of probability measures with finite second-order moments is absolutely continuous.

By Ambrosio et al. [2008, Theorem 8.3.1] we have that among all velocity fields $v_t$ which produce the same flow $\mu$, there is a unique optimal one with smallest $L^p(\mu_t; X)$-norm. This is summarised in the following lemma.

**Lemma A.3** (Lemma 2 from Guo et al. [2025]). *For an absolutely continuous curve of probability measures $\mu = (\mu_t)_{t \in [a,b]}$, any vector field $(v_t)_{t \in [a,b]}$ that generates $\mu$ satisfies $|\dot{\mu}|_t \leq \|v_t\|_{L^2(\mu_t)}$ for almost every $t \in [a,b]$. Moreover, there exists a unique vector field $v_t^\star$ generating $\mu$ such that $|\dot{\mu}|_t = \|v_t^\star\|_{L^2(\mu_t)}$ almost everywhere.*

We also introduce the action of the absolutely continuous curve $(\mu_t)_{t \in [a,b]}$ since it will play a key role in our convergence results. In particular, we define the action $\mathcal{A}(\mu)$ as

$$\mathcal{A}(\mu) := \int_a^b |\dot{\mu}|_t^2 \, \mathrm{d}t.$$

**Girsanov's theorem.** Consider the SDE
$$\mathrm{d}X_t = b(X_t, t)\mathrm{d}t + \sigma(X_t, t)\mathrm{d}B_t,$$
for $t \in [0, T]$, where $(B_t)_{t \in [0,T]}$ is a standard Brownian motion in $\mathbb{R}^d$. Denote by $\mathbb{P}^X$ the *path measure* of the solution $X = (X_t)_{t \in [0,T]}$ of the SDE, which characterises the distribution of $X$ over the sample space $\Omega$.

The KL divergence between two path measures can be characterised as a consequence of Girsanov's theorem [Karatzas and Shreve, 1991]. In particular, the following result will be central in our analysis.

**Lemma A.4.** *Consider the following two SDEs defined on a common probability space $(\Omega, \mathcal{F}, \mathbb{P})$*
$$\mathrm{d}X_t = a_t(X)\mathrm{d}t + \sqrt{2}\mathrm{d}B_t, \qquad \mathrm{d}Y_t = b_t(Y)\mathrm{d}t + \sqrt{2}\mathrm{d}B_t, \qquad t \in [0, T]$$
*with the same initial conditions $X_0, Y_0 \sim \mu_0$. Denote by $\mathbb{P}^X$ and $\mathbb{P}^Y$ the path measures of the processes $X$ and $Y$, respectively. It follows that*

$$\mathrm{KL}(\mathbb{P}^X \| \mathbb{P}^Y) = \frac{1}{4} \mathbb{E}_{X \sim \mathbb{P}^X} \left[ \int_0^T \|a_t(X) - b_t(X)\|^2 \mathrm{d}t \right].$$

# B  Sampling using multiscale dynamics

In this section, we provide a detailed description of the implementation of our proposed algorithms including the numerical discretisation schemes used.

## B.1  MULTALMC: Multiscale Annealed Langevin Monte Carlo

Here, we present further details of the overdamped and underdamped versions of the sampler.

### B.1.1  Overdamped system

In the overdamped setting, we propose a new sampling strategy based on the following stochastic slow-fast system

$$\begin{cases} \mathrm{d}X_t = \begin{cases} \frac{1}{\sqrt{1-\lambda_{\kappa t}}} \nabla \log \nu \left( \frac{X_t - Y_t}{\sqrt{1-\lambda_{\kappa t}}} \right) \mathrm{d}t + \sqrt{2}\mathrm{d}B_t & \text{if } \lambda_{\kappa t} < \tilde{\lambda} \\ -\frac{1}{\sqrt{\lambda_{\kappa t}}} \nabla V_\pi \left( \frac{Y_t}{\sqrt{\lambda_{\kappa t}}} \right) \mathrm{d}t + \sqrt{2}\mathrm{d}B_t & \text{if } \lambda_{\kappa t} \geq \tilde{\lambda} \end{cases} \\ \mathrm{d}Y_t = \frac{1}{\varepsilon} \left( -\frac{1}{\sqrt{\lambda_{\kappa t}}} \nabla V_\pi \left( \frac{Y_t}{\sqrt{\lambda_{\kappa t}}} \right) + \frac{1}{\sqrt{1-\lambda_{\kappa t}}} \nabla \log \nu \left( \frac{Y_t - X_t}{\sqrt{1-\lambda_{\kappa t}}} \right) \right) \mathrm{d}t + \sqrt{\frac{2}{\varepsilon}}\mathrm{d}\tilde{B}_t. \end{cases}$$

As discussed in the main text, the conditional distribution $\rho_{t,x}$ converges to a Dirac delta centred at 0 when $\lambda_{\kappa t} = 0$, and to a Dirac at $x$ when $\lambda_{\kappa t} = 1$. These degenerate limits can cause numerical instabilities at the endpoints of the diffusion schedule. To mitigate this, we define $0 < \lambda_\delta \ll 1$ and consider the modified dynamics

$$\begin{cases} \mathrm{d}X_t = \begin{cases} \frac{1}{\sqrt{1-\lambda_\delta}} \nabla \log \nu \left( \frac{X_t - Y_t}{\sqrt{1-\lambda_\delta}} \right) \mathrm{d}t + \sqrt{2}\mathrm{d}B_t & \text{if } 0 \leq \lambda_{\kappa t} < \lambda_\delta \\ \frac{1}{\sqrt{1-\lambda_{\kappa t}}} \nabla \log \nu \left( \frac{X_t - Y_t}{\sqrt{1-\lambda_{\kappa t}}} \right) \mathrm{d}t + \sqrt{2}\mathrm{d}B_t & \text{if } \lambda_\delta \leq \lambda_{\kappa t} < \tilde{\lambda} \\ -\frac{1}{\sqrt{\lambda_{\kappa t}}} \nabla V_\pi \left( \frac{Y_t}{\sqrt{\lambda_{\kappa t}}} \right) \mathrm{d}t + \sqrt{2}\mathrm{d}B_t & \text{if } \tilde{\lambda} \leq \lambda_{\kappa t} < 1 - \lambda_\delta \\ -\nabla V_\pi \left( X_t \right) \mathrm{d}t + \sqrt{2}\mathrm{d}B_t & \text{if } 1 - \lambda_\delta \leq \lambda_{\kappa t} \leq 1 \end{cases} \\ \\ \mathrm{d}Y_t = \begin{cases} \frac{1}{\varepsilon} \left( -\frac{1}{\sqrt{\lambda_\delta}} \nabla V_\pi \left( \frac{Y_t}{\sqrt{\lambda_\delta}} \right) + \frac{1}{\sqrt{1-\lambda_\delta}} \nabla \log \nu \left( \frac{Y_t - X_t}{\sqrt{1-\lambda_\delta}} \right) \right) \mathrm{d}t + \sqrt{\frac{2}{\varepsilon}}\mathrm{d}\tilde{B}_t & \text{if } 0 \leq \lambda_{\kappa t} < \lambda_\delta \\ \frac{1}{\varepsilon} \left( -\frac{1}{\sqrt{\lambda_{\kappa t}}} \nabla V_\pi \left( \frac{Y_t}{\sqrt{\lambda_{\kappa t}}} \right) + \frac{1}{\sqrt{1-\lambda_{\kappa t}}} \nabla \log \nu \left( \frac{Y_t - X_t}{\sqrt{1-\lambda_{\kappa t}}} \right) \right) \mathrm{d}t + \sqrt{\frac{2}{\varepsilon}}\mathrm{d}\tilde{B}_t & \text{if } \lambda_\delta \leq \lambda_{\kappa t} \leq 1. \end{cases} \end{cases}$$
$$(13)$$

Note that if the slow-fast system converges to its corresponding averaged dynamics, then running the modified multiscale system up to time $t_\delta$, where $\lambda_{\kappa t_\delta} = 1 - \lambda_\delta$, we can approximate the distribution $\hat{\mu}_{t_\delta}$ that is close to the target $\pi$, provided $\lambda_\delta$ is sufficiently small. For $t \geq t_\delta$, the dynamics in (13) reduce to standard overdamped Langevin dynamics targetting $\pi$, initialised at $\hat{\mu}_{t_\delta}$. This warm start is known to significantly improve convergence rates to the target distribution [Chewi et al., 2021, Lee et al., 2021a, Wu et al., 2022].

We now analyse different numerical schemes for implementing the sampler in practice. As we have just mentioned, when $\lambda_{\kappa t} \geq 1 - \lambda_\delta$, the dynamics in (13) reduce to standard overdamped Langevin dynamics, which can be discretised using any preferred numerical method. Therefore, we focus on the regime $0 \leq \lambda_{\kappa t} < 1 - \lambda_\delta$.

For general base distributions $\nu$, we propose to use the following numerical scheme that combines Euler-Maruyama for the slow dynamics with the SROCK method [Abdulle et al., 2018] for the fast dynamics. Before presenting the complete algorithm in Algorithm 2, we introduce the necessary notation and coefficients for the SROCK update.

Denote by $T_s$ the Chebychev polynomials of first kind and define the coefficients in the SROCK update as follows

$$\omega_0 = 1 + \frac{\eta}{s^2}, \quad \omega_1 = \frac{T_s(\omega_0)}{T'_s(\omega_0)}, \quad \mu_1 = \frac{\omega_1}{\omega_0}, \tag{14}$$

and for all $i = 2, \ldots, s$,

$$\mu_i = \frac{2\omega_1 T_{i-1}(\omega_0)}{T_i(\omega_0)}, \quad \nu_i = \frac{2\omega_0 T_{i-1}(\omega_0)}{T_i(\omega_0)}, \quad \kappa_i = 1 - \nu_i. \tag{15}$$

Additionally, when $\nu \sim \mathcal{N}(0, I)$, we can further exploit the linear structure of the score function for a Gaussian distribution to design an efficient exponential integrator scheme. Below, we derive the corresponding update rules, distinguishing between the regimes $0 \leq \lambda_{\kappa t} < \tilde{\lambda}$ and $\tilde{\lambda} \leq \lambda_{\kappa t} \leq 1$.

**Regime 1:** $0 \leq \lambda_{\kappa t} < \tilde{\lambda}$ . Define the modified schedule

$$\lambda'_{\kappa t} = \begin{cases} \lambda_\delta & \text{if } 0 \leq \lambda_{\kappa t} < \lambda_\delta \\ \lambda_{\kappa t} & \text{if } \lambda_\delta \leq \lambda_{\kappa t} \leq 1. \end{cases}$$

The slow-fast system (13) simplifies to

$$\begin{cases} \mathrm{d}X_t = -\frac{X_t - Y_t}{1 - \lambda'_{\kappa t}} \mathrm{d}t + \sqrt{2} \mathrm{d}B_t \\ \mathrm{d}Y_t = \frac{1}{\varepsilon}\left(-\frac{1}{\sqrt{\lambda'_{\kappa t}}}\nabla V_\pi\left(\frac{Y_t}{\sqrt{\lambda'_{\kappa t}}}\right) - \frac{Y_t - X_t}{1 - \lambda'_{\kappa t}}\right)\mathrm{d}t + \sqrt{\frac{2}{\varepsilon}}\mathrm{d}\tilde{B}_t. \end{cases}$$

The exponential integrator scheme is then expressed as

$$\mathrm{d}\begin{pmatrix} X_t \\ Y_t \end{pmatrix} = -\frac{1}{1 - \lambda'_{\kappa t}}\begin{pmatrix} 1 & -1 \\ -\varepsilon^{-1} & \varepsilon^{-1} \end{pmatrix} \otimes I_d \begin{pmatrix} X_t \\ Y_t \end{pmatrix}\mathrm{d}t - \frac{1}{\varepsilon\sqrt{\lambda'_{\kappa t}}}\begin{pmatrix} \mathbf{0}_d \\ \nabla V_\pi\left(\frac{Y_{t_-}}{\sqrt{\lambda'_{t_-}}}\right) \end{pmatrix}\mathrm{d}t + \sqrt{2}\begin{pmatrix} 1 & 0 \\ 0 & \sqrt{\varepsilon^{-1}} \end{pmatrix} \otimes I_d \begin{pmatrix} \mathrm{d}B_t \\ \mathrm{d}\tilde{B}_t \end{pmatrix},$$

where given a time discretisation $0 \leq T_0 < \cdots < T_M$ of the corresponding time interval, we define $t_- = T_{l-1}$ when $t \in [T_{l-1}, T_l)$. The explicit update rule is then

$$\begin{pmatrix} X_{l+1} \\ Y_{l+1} \end{pmatrix} = A_0(T_l, T_{l+1}) \otimes I_d \begin{pmatrix} X_l \\ Y_l \end{pmatrix} - \frac{A_1(T_l, T_{l+1})}{\varepsilon} \otimes I_d \begin{pmatrix} \mathbf{0}_d \\ \nabla V_\pi\left(\frac{Y_l}{\sqrt{\lambda'_{T_l}}}\right) \end{pmatrix} + A_2(T_l, T_{l+1}) \otimes I_d\, \xi_l,$$

where $\xi_l \sim \mathcal{N}(0, I_{2d})$ and

$$A_0(a, b) = \exp\left(-\begin{pmatrix} 1 & -1 \\ -\varepsilon^{-1} & \varepsilon^{-1} \end{pmatrix}\int_a^b (1 - \lambda'_{\kappa u})^{-1}\mathrm{d}u\right)$$

$$A_1(a, b) = \int_a^b \frac{1}{\sqrt{\lambda'_{\kappa u}}}A_0(u, b)\mathrm{d}u$$

$$A_2(a, b) = \sqrt{2\int_a^b A_0(u, b)\begin{pmatrix} 1 & 0 \\ 0 & \varepsilon^{-1} \end{pmatrix}A_0(u, b)^\intercal \mathrm{d}u}.$$

**Algorithm 2** MULTALMC sampler: overdamped version

---

**Require:** Schedule function $\lambda_t$, value for $\lambda_\delta$ and $\tilde{\lambda}$, number of sampling steps $L$, time discretisation $0 = T_0 < \cdots < T_L = 1/\kappa$, step size $h_l = T_{l+1} - T_l$. Constants for SROCK step from (14), (15).

**Initial samples** $X_0 \sim \mathcal{N}(0, I), Y_0 \sim \mathcal{N}(0, I)$. Define the schedule

$$\lambda'_{\kappa t} = \begin{cases} \lambda_\delta & \text{if } 0 \le \lambda_{\kappa t} < \lambda_\delta \\ \lambda_{\kappa t} & \text{if } \lambda_\delta \le \lambda_{\kappa t} \le 1. \end{cases}$$

**for** $l = 0$ **to** $L$ **do**

$\xi_l^{(1)}, \xi_l^{(2)} \sim \mathcal{N}(0, I)$

**if** $0 \le \lambda_{\kappa T_l} < 1 - \lambda_\delta$ **then**

```
EM for slow dynamics
```

$$X_{l+1} = \begin{cases} X_l + \frac{1}{\sqrt{1-\lambda'_{\kappa T_l}}} \nabla \log \nu \left( \frac{X_l - Y_l}{\sqrt{1-\lambda'_{\kappa T_l}}} \right) h_l + \sqrt{2h_l}\xi_l^{(1)} & \text{if } \lambda'_{\kappa T_l} < \tilde{\lambda} \\ X_l - \frac{1}{\sqrt{\lambda'_{\kappa T_l}}} \nabla V_\pi \left( \frac{Y_l}{\sqrt{\lambda'_{\kappa T_l}}} \right) h_l + \sqrt{2h_l}\xi_l^{(1)} & \text{if } \lambda'_{\kappa T_l} \ge \tilde{\lambda} \end{cases}$$

```
SROCK for fast dynamics
```

$$\begin{cases} K_{l,0} = Y_l \\ K_{l,1} = K_{l,0} + \mu_1 \frac{h_l}{\varepsilon} \left( -\frac{1}{\sqrt{\lambda'_{\kappa T_l}}} \nabla V_\pi \left( \frac{K_{l,0}}{\sqrt{\lambda'_{\kappa T_l}}} \right) + \frac{1}{\sqrt{1-\lambda'_{\kappa T_l}}} \nabla \log \nu \left( \frac{K_{l,0} - X_{l+1}}{\sqrt{1-\lambda'_{\kappa T_l}}} \right) \right) \\ \quad \vdots \\ K_{l,i} = \mu_i \frac{h_l}{\varepsilon} \left( -\frac{1}{\sqrt{\lambda'_{\kappa T_l}}} \nabla V_\pi \left( \frac{K_{l,i-1}}{\sqrt{\lambda'_{\kappa T_l}}} \right) + \frac{1}{\sqrt{1-\lambda'_{\kappa T_l}}} \nabla \log \nu \left( \frac{K_{l,i-1} - X_{l+1}}{\sqrt{1-\lambda'_{\kappa T_l}}} \right) \right) + \nu_i K_{l,i-1} + \kappa_i K_{l,i-2} \\ \quad \vdots \\ Y_{l+1} = K_s + \sqrt{\frac{2h}{\varepsilon}}\xi_l^{(2)} \end{cases}$$

**if** $1 - \lambda_\delta \le \lambda_{\kappa T_l} \le 1$ **then**

$\quad X_{l+1} = X_l - h_l \nabla V_\pi (X_l) + \sqrt{2h_l}\xi_l^{(1)}$

**end for**

$\quad$ **return** $X_L$

---

**Regime 2:** $\tilde{\lambda} \le \lambda_{\kappa t} < 1 - \lambda_\delta$ . Here, the dynamics become

$$\begin{cases} dX_t = -\frac{1}{\sqrt{\lambda_{\kappa t}}} \nabla V_\pi \left( \frac{Y_t}{\sqrt{\lambda_{\kappa t}}} \right) dt + \sqrt{2}dB_t \\ dY_t = \frac{1}{\varepsilon} \left( -\frac{1}{\sqrt{\lambda_{\kappa t}}} \nabla V_\pi \left( \frac{Y_t}{\sqrt{\lambda_{\kappa t}}} \right) - \frac{Y_t - X_t}{1 - \lambda_{\kappa t}} \right) dt + \sqrt{\frac{2}{\varepsilon}}d\tilde{B}_t. \end{cases}$$

In this case, the exponential integrator scheme can be formulated as follows

$$d \begin{pmatrix} X_t \\ Y_t \end{pmatrix} = -\frac{1}{1-\lambda_{\kappa t}} \begin{pmatrix} 0 & 0 \\ -\varepsilon^{-1} & \varepsilon^{-1} \end{pmatrix} \otimes I_d \begin{pmatrix} X_t \\ Y_t \end{pmatrix} dt - \frac{1}{\sqrt{\lambda_{\kappa t}}} \begin{pmatrix} \nabla V_\pi \left( \frac{Y_{t_-}}{\sqrt{\lambda_{t_-}}} \right) \\ \frac{1}{\varepsilon} \nabla V_\pi \left( \frac{Y_{t_-}}{\sqrt{\lambda_{t_-}}} \right) \end{pmatrix} dt + \sqrt{2} \begin{pmatrix} 1 & 0 \\ 0 & \sqrt{\varepsilon^{-1}} \end{pmatrix} \otimes I_d \begin{pmatrix} dB_t \\ d\tilde{B}_t \end{pmatrix},$$

The corresponding update rule is

$$\begin{pmatrix} X_{l+1} \\ Y_{l+1} \end{pmatrix} = \tilde{A}_0(T_l, T_{l+1}) \otimes I_d \begin{pmatrix} X_l \\ Y_l \end{pmatrix} - \tilde{A}_1(T_l, T_{l+1}) \otimes I_d \begin{pmatrix} \nabla V_\pi \left( \frac{Y_l}{\sqrt{\lambda_{T_l}}} \right) \\ \frac{1}{\varepsilon} \nabla V_\pi \left( \frac{Y_l}{\sqrt{\lambda_{T_l}}} \right) \end{pmatrix} + \tilde{A}_2(T_l, T_{l+1}) \otimes I_d\, \xi_l,$$

where $\xi_l \sim \mathcal{N}(0, I_{2d})$ and

$$\tilde{A}_0(a, b) = \exp \left( - \begin{pmatrix} 0 & 0 \\ -\varepsilon^{-1} & \varepsilon^{-1} \end{pmatrix} \int_a^b (1 - \lambda_{\kappa u})^{-1} du \right)$$

$$\tilde{A}_1(a,b) = \int_a^b \frac{1}{\sqrt{\lambda_{\kappa u}}} \tilde{A}_0(u,b)\mathrm{d}u$$

$$\tilde{A}_2(a,b) = \sqrt{2\int_a^b \tilde{A}_0(u,b)\begin{pmatrix} 1 & 0 \\ 0 & \varepsilon^{-1} \end{pmatrix}\tilde{A}_0(u,b)^{\mathsf{T}}\mathrm{d}u}.$$

### B.1.2 Underdamped system

Similarly in the overdamped case, to mitigate numerical instabilities arising from the degeneracy of $\rho_{t,x}$ when $\lambda_{\kappa t} = 0$ and $\lambda_{\kappa t} = 1$, we consider the following modified dynamics.

$$\begin{cases}
\mathrm{d}X_t &= M^{-1}V_t\mathrm{d}t \\[4pt]
\mathrm{d}V_t &= \begin{cases}
\left(\frac{1}{\sqrt{1-\lambda_\delta}}\nabla\log\nu\left(\frac{X_t-Y_t}{\sqrt{1-\lambda_\delta}}\right) - \Gamma M^{-1}V_t\right)\mathrm{d}t + \sqrt{2\Gamma}\mathrm{d}B_t & \text{if } 0 \leq \lambda_{\kappa t} < \lambda_\delta \\
\left(\frac{1}{\sqrt{1-\lambda_{\kappa t}}}\nabla\log\nu\left(\frac{X_t-Y_t}{\sqrt{1-\lambda_{\kappa t}}}\right) - \Gamma M^{-1}V_t\right)\mathrm{d}t + \sqrt{2\Gamma}\mathrm{d}B_t & \text{if } \lambda_\delta \leq \lambda_{\kappa t} < \tilde{\lambda} \\
\left(-\frac{1}{\sqrt{\lambda_{\kappa t}}}\nabla V_\pi\left(\frac{Y_t}{\sqrt{\lambda_{\kappa t}}}\right) - \Gamma M^{-1}V_t\right)\mathrm{d}t + \sqrt{2\Gamma}\mathrm{d}B_t & \text{if } \tilde{\lambda} \leq \lambda_{\kappa t} < 1-\lambda_\delta \\
\left(-\nabla V_\pi(X_t) - \Gamma M^{-1}V_t\right)\mathrm{d}t + \sqrt{2\Gamma}\mathrm{d}B_t & \text{if } 1-\lambda_\delta \leq \lambda_{\kappa t} \leq 1
\end{cases} \\[4pt]
\mathrm{d}Y_t &= \begin{cases}
\frac{1}{\varepsilon}\left(-\frac{1}{\sqrt{\lambda_\delta}}\nabla V_\pi\left(\frac{Y_t}{\sqrt{\lambda_\delta}}\right) + \frac{1}{\sqrt{1-\lambda_\delta}}\nabla\log\nu\left(\frac{Y_t-X_t}{\sqrt{1-\lambda_\delta}}\right)\right)\mathrm{d}t + \sqrt{\frac{2}{\varepsilon}}\mathrm{d}\tilde{B}_t & \text{if } 0 \leq \lambda_{\kappa t} < \lambda_\delta \\
\frac{1}{\varepsilon}\left(-\frac{1}{\sqrt{\lambda_{\kappa t}}}\nabla V_\pi\left(\frac{Y_t}{\sqrt{\lambda_{\kappa t}}}\right) + \frac{1}{\sqrt{1-\lambda_{\kappa t}}}\nabla\log\nu\left(\frac{Y_t-X_t}{\sqrt{1-\lambda_{\kappa t}}}\right)\right)\mathrm{d}t + \sqrt{\frac{2}{\varepsilon}}\mathrm{d}\tilde{B}_t & \text{if } \lambda_\delta \leq \lambda_{\kappa t} \leq 1.
\end{cases}
\end{cases}$$
$$\tag{16}$$

Note that when $\lambda_{\kappa t} \geq 1 - \lambda_\delta$, the dynamics in (16) reduce to standard underdamped Langevin dynamics with a warm start, which can be discretised using any preferred numerical scheme. Therefore, we focus on the regime $0 \leq \lambda_{\kappa t} < 1 - \lambda_\delta$. We propose to use a hybrid discretisation method that combines the OBABO splitting scheme [Monmarché, 2020] with an SROCK update [Abdulle et al., 2018] for the fast-dynamics. The algorithm is explictly defined in Algorithm 3, where $\sqrt{\Gamma}$ denotes the matrix square root of the friction coefficient, which is well defined since $\Gamma$ is a symmetric positive definite matrix. That is, $\sqrt{\Gamma}$ is any matrix satisfying $\sqrt{\Gamma}\sqrt{\Gamma}^{\mathsf{T}} = \Gamma$.

It is important to mention that for implementation, we set the mass parameter to $M = I$ and consider a time-dependent friction coefficient $\Gamma_t$ to control the degree of acceleration. Additionally, the same discretisation scheme applies to the heavy-tailed diffusion, with a modified fast process defined by Eq. (7).

### B.2 MULTCDIFF: Multiscale Controlled Diffusions

#### B.2.1 Overdamped system

Diffusion models typically consider a stochastic process $(\overrightarrow{X}_t)_{t\in[0,T]}$ constructed by initialising $\overrightarrow{X}_0$ at the target distribution $\pi$ and then evolving according to the OU SDE

$$\mathrm{d}\overrightarrow{X}_t = -\overrightarrow{X}_t\,\mathrm{d}t + \sqrt{2}\,\mathrm{d}B_t'. \tag{17}$$

Under mild regularity conditions [Anderson, 1982], the dynamics of the reverse process $(\overleftarrow{X}_t)_{t\in[0,T]}$ are described by the following SDE

$$\mathrm{d}\overleftarrow{X}_t = \left(\overleftarrow{X}_t + 2\nabla\log q_{T-t}(\overleftarrow{X}_t)\right)\mathrm{d}t + \sqrt{2}\,\mathrm{d}B_t, \tag{18}$$

where $q_t$ denotes the marginal distribution of the solution of the forward process (17), given by

$$q_t(x) = \int \pi(y)\rho_t(x|y)\mathrm{d}y,$$

with $\rho_t(x|y)$ being the Gaussian transition density $\mathcal{N}(e^{-t}y, \sigma_t^2 I)$, where $\sigma_t^2 = 1 - e^{-2t}$. Using this, the score function $\nabla\log q_{T-t}(\overleftarrow{X}_t)$ can be expressed as

$$\nabla\log q_{T-t}(\overleftarrow{X}_t) = -\frac{\overleftarrow{X}_t - e^{-(T-t)}\mathbb{E}_{Y|\overleftarrow{X}_t}[Y]}{\sigma_{T-t}^2}.$$

Since the score term involves an intractable expectation, we approximate the averaged dynamics in (18) using a multiscale SDE system. The fast process of the system is modelled by an overdamped

---

**Algorithm 3** MULTALMC sampler: accelerated version

---

**Require:** Schedule function $\lambda_t$, value for $\lambda_\delta$ and $\tilde{\lambda}$, friction coefficient $\Gamma$ (scalar), mass parameter $M$ (scalar), number of sampling steps $L$, time discretisation $0 = T_0 < \cdots < T_L = 1/\kappa$, step size $h_l = T_{l+1} - T_l$. Constants for SROCK step from (14), (15).

**Initial samples** $X_0 \sim \mathcal{N}(0, I), V_0 \sim \mathcal{N}(0, MI), Y_0 \sim \mathcal{N}(0, I)$. Define the schedule

$$\lambda'_{\kappa t} = \begin{cases} \lambda_\delta & \text{if } 0 \leq \lambda_{\kappa t} < \lambda_\delta \\ \lambda_{\kappa t} & \text{if } \lambda_\delta \leq \lambda_{\kappa t} \leq 1. \end{cases}$$

**for** $l = 0$ **to** $L$ **do**

    $\xi_l^{(1)}, \xi_l^{(2)}, \xi_l^{(3)} \sim \mathcal{N}(0, I)$

    **if** $0 \leq \lambda_{\kappa T_l} < 1 - \lambda_\delta$ **then**

        `Half-step for velocity component`

$$V_l' = \left(1 - \frac{h_l}{2}\Gamma M^{-1}\right) V_l + \sqrt{h_l \Gamma} \xi_l^{(1)}$$

$$V_l'' = \begin{cases} V_l' + \frac{h_l}{2\sqrt{1 - \lambda'_{\kappa T_l}}} \nabla \log \nu \left(\frac{X_l - Y_l}{\sqrt{1 - \lambda'_{\kappa T_l}}}\right) & \text{if } \lambda'_{\kappa T_l} < \tilde{\lambda} \\ V_l' - \frac{h_l}{2\sqrt{\lambda'_{\kappa T_l}}} \nabla V_\pi \left(\frac{Y_l}{\sqrt{\lambda'_{\kappa T_l}}}\right) & \text{if } \lambda'_{\kappa T_l} \geq \tilde{\lambda} \end{cases}$$

        `Full EM-step for position component`

$$X_{l+1} = X_l + h_l M^{-1} V_l''$$

        `Full SROCK-step for fast component`

$$\begin{cases} K_{l,0} & = Y_l \\ K_{l,1} & = K_{l,0} + \mu_1 \frac{h_l}{\varepsilon}\left(-\frac{1}{\sqrt{\lambda'_{\kappa T_l}}} \nabla V_\pi \left(\frac{K_{l,0}}{\sqrt{\lambda'_{\kappa T_l}}}\right) + \frac{1}{\sqrt{1 - \lambda'_{\kappa T_l}}} \nabla \log \nu \left(\frac{K_{l,0} - X_{l+1}}{\sqrt{1 - \lambda'_{\kappa T_l}}}\right)\right) \\ & \vdots \\ K_{l,i} & = \mu_i \frac{h_l}{\varepsilon}\left(-\frac{1}{\sqrt{\lambda'_{\kappa T_l}}} \nabla V_\pi \left(\frac{K_{l,i-1}}{\sqrt{\lambda'_{\kappa T_l}}}\right) + \frac{1}{\sqrt{1 - \lambda'_{\kappa T_l}}} \nabla \log \nu \left(\frac{K_{l,i-1} - X_{l+1}}{\sqrt{1 - \lambda'_{\kappa T_l}}}\right)\right) + \nu_i K_{l,i-1} + \kappa_i K_{l,i-2} \\ & \vdots \\ Y_{l+1} & = K_s + \sqrt{\frac{2h}{\varepsilon}} \xi_l^{(2)} \end{cases}$$

        `Half-step for velocity component`

$$V_l''' = \begin{cases} V_l'' + \frac{h_l}{2\sqrt{1 - \lambda'_{\kappa T_l}}} \nabla \log \nu \left(\frac{X_{l+1} - Y_{l+1}}{\sqrt{1 - \lambda'_{\kappa T_l}}}\right) & \text{if } \lambda'_{\kappa T_l} < \tilde{\lambda} \\ V_l'' - \frac{h_l}{2\sqrt{\lambda'_{\kappa T_l}}} \nabla V_\pi \left(\frac{Y_{l+1}}{\sqrt{\lambda'_{\kappa T_l}}}\right) & \text{if } \lambda'_{\kappa T_l} \geq \tilde{\lambda} \end{cases}$$

$$V_{l+1} = \left(1 - \frac{h_l}{2}\Gamma M^{-1}\right) V_l''' + \sqrt{h_l \Gamma} \xi_l^{(3)}$$

    **if** $1 - \lambda_\delta \leq \lambda_{\kappa T_l} \leq 1$ **then**

        `Half-step for velocity component`
        $V_l' = \left(1 - \frac{h_l}{2}\Gamma M^{-1}\right) V_l + \sqrt{h_l \Gamma} \xi_l^{(1)}$
        $V_l'' = V_l' - \frac{h_l}{2} \nabla V_\pi (X_l)$

        `Full EM-step for position component`
        $X_{l+1} = X_l + h_l M^{-1} V_l''$

        `Half-step for velocity component`
        $V_l''' = V_l'' - \frac{h_l}{2} \nabla V_\pi (X_{l+1})$
        $V_{l+1} = \left(1 - \frac{h_l}{2}\Gamma M^{-1}\right) V_l''' + \sqrt{h_l \Gamma} \xi_l^{(3)}$

**end for**

    **return** $(X_L, V_L)$

---

Langevin SDE targetting the conditional distribution $Y|\overleftarrow{X}_t$, which takes the form

$$dY_t = \frac{1}{\varepsilon}\left(\nabla \log \pi(Y_t) + \nabla_{Y_t} \log \rho_{T-t}(\overleftarrow{X}_t|Y_t)\right) dt + \sqrt{\frac{2}{\varepsilon}} d\tilde{B}_t.$$

where

$$\nabla_{Y_t} \log \rho_{T-t}(\overleftarrow{X}_t, \overleftarrow{V}_t|Y_t) = -\frac{Y_t - e^{T-t}\overleftarrow{X}_t}{\sigma_{T-t}^2} e^{-2(T-t)}.$$

Putting everything together, we obtain the following multiscale system of SDEs

$$\begin{cases} d\overleftarrow{X}_t = \left(\overleftarrow{X}_t - \frac{2}{\sigma_{T-t}^2}\left(\overleftarrow{X}_t - e^{-(T-t)}Y_t\right)\right) dt + \sqrt{2}\, dB_t \\ dY_t = \frac{1}{\varepsilon}\left(\nabla \log \pi(Y_t) - \frac{e^{-2(T-t)}}{\sigma_{T-t}^2}\left(Y_t - e^{T-t}\overleftarrow{X}_t\right)\right) dt + \sqrt{\frac{2}{\varepsilon}}d\tilde{B}_t. \end{cases} \tag{19}$$

This system of SDEs can be discretised using either an exponential integrator scheme or a numerical scheme that combines Euler-Maruyama for the slow dynamics with the SROCK method [Abdulle et al., 2018] for the fast component, similarly to the discretisation of MULTALMC in the overdamped case (see Appendix B.1.1).

### B.2.2 Underdamped system

When using an OU noising process, the forward underdamped diffusion has the following form

$$\begin{cases} d\overrightarrow{X}_t = M^{-1}\overrightarrow{V}_t\, dt \\ d\overrightarrow{V}_t = \left(-\overrightarrow{X}_t - \Gamma M^{-1}\overrightarrow{V}_t\right) dt + \sqrt{2\Gamma}\, dB'_t, \end{cases}$$

which can be written compactly as

$$d\overrightarrow{Z}_t = A \otimes I_d\, \overrightarrow{Z}_t\, dt + B \otimes I_d\, dB''_t,$$

where $\overrightarrow{Z}_t = (\overrightarrow{X}_t, \overrightarrow{V}_t)$ and

$$A = \begin{pmatrix} 0 & M^{-1} \\ -1 & -\Gamma M^{-1} \end{pmatrix}, \qquad B = \begin{pmatrix} 0 & 0 \\ 0 & \sqrt{2\Gamma} \end{pmatrix},$$

where, as in Appendix B.1.2, $\sqrt{\Gamma}$ denotes the matrix square root of the friction coefficient $\Gamma$, which is a symmetric and positive definite. The solution of this SDE is given by [Karatzas and Shreve, 1991]

$$\overrightarrow{Z}_t = e^{At} \otimes I_d\, \overrightarrow{Z}_0 + \int_0^t \left(e^{A(t-s)}B\right) \otimes I_d\, dB''_t.$$

As discussed in the main text, there is a crucial balance between the mass $M$ and friction coefficient $\Gamma$. We focus on the critically damping regime by setting $\Gamma^2 = 4M$. This provides an ideal balance between the Hamiltonian and the Ohnstein-Uhlenbeck components of the dynamics, which leads to faster convergence without oscillations [McCall, 2010]. Under this setting, the matrix $A$ can be written as

$$A = \begin{pmatrix} 0 & 4\Gamma^{-2} \\ -1 & -4\Gamma^{-1} \end{pmatrix}.$$

Then, the matrix exponential $e^{At}$ has the following form

$$e^{At} = \begin{pmatrix} 1 + 2\Gamma^{-1}t & 4\Gamma^{-2}t \\ -t & 1 - 2\Gamma^{-1}t \end{pmatrix} e^{-2\Gamma^{-1}t} = \begin{pmatrix} m_{1,t} & m_{2,t} \\ m_{3,t} & m_{4,t} \end{pmatrix}.$$

On the other hand, the reverse SDE is given by

$$\begin{cases} d\overleftarrow{X}_t = -4\Gamma^{-2}\overleftarrow{V}_t\, dt \\ d\overleftarrow{V}_t = \left(\overleftarrow{X}_t + 4\Gamma^{-1}\overleftarrow{V}_t + 2\Gamma\nabla_{V_t}\log q_{T-t}(\overleftarrow{X}_t\overleftarrow{V}_t)\right) dt + \sqrt{2\Gamma}\, dB_t, \end{cases}$$

where $q_t$ is the marginal distribution of the solution of the forward noising process which has the following expression

$$q_t(x, v) = \int \pi(y)\varphi(v_0)\rho_t(x, v|y, v_0)\, dy\, dv_0, \tag{20}$$

with $\varphi = \mathcal{N}(0, I)$ and conditional distribution $\rho_t(x, v|y, v_0)$ given by a Normal distribution with mean

$$m_t(y, v_0) = e^{At} \otimes I_d \begin{pmatrix} y \\ v_0 \end{pmatrix} = \begin{pmatrix} m_{1,t}y + m_{2,t}v_0 \\ m_{3,t}y + m_{4,t}v_0 \end{pmatrix}$$

and covariance matrix

$$\Sigma_t \otimes I_d = \begin{pmatrix} \Sigma_{11,t} & \Sigma_{12,t} \\ \Sigma_{21,t} & \Sigma_{22,t} \end{pmatrix} \otimes I_d = \left( \int_0^t e^{A(t-s)} BB^\intercal \left[ e^{A(t-s)} \right]^\intercal ds \right) \otimes I_d,$$

which results into

$$\Sigma_{11,t} = \left( e^{4t\Gamma^{-1}} - 1 - 4t\Gamma^{-1} - 8t^2\Gamma^{-2} \right) e^{-4t\Gamma^{-1}}$$

$$\Sigma_{12,t} = \Sigma_{21,t} = 4t^2\Gamma^{-1}e^{-4t\Gamma^{-1}}$$

$$\Sigma_{22,t} = \left( \frac{\Gamma^2}{4} \left( e^{4t\Gamma^{-1}} - 1 \right) + t\Gamma - 2t^2 \right) e^{-4t\Gamma^{-1}}.$$

We note that $v_0$ in Eq. (20) can be integrated out analytically. Since both $\rho_t(x, v|y, v_0)$ and $\varphi(v_0)$ are Gaussian, it follows that $\rho_t(x, v|y)$ is also Gaussian with mean

$$\hat{m}(y) = \begin{pmatrix} m_{1,t}y \\ m_{3,t}y \end{pmatrix}$$

and covariance matrix

$$\tilde{\Sigma}_t \otimes I_d = \begin{pmatrix} \tilde{\Sigma}_{11,t} & \tilde{\Sigma}_{12,t} \\ \tilde{\Sigma}_{21,t} & \tilde{\Sigma}_{22,t} \end{pmatrix} \otimes I_d = \left[ \begin{pmatrix} \Sigma_{11,t} & \Sigma_{12,t} \\ \Sigma_{21,t} & \Sigma_{22,t} \end{pmatrix} + \begin{pmatrix} m_{2,t}^2 & m_{2,t}m_{4,t} \\ m_{2,t}m_{4,t} & m_{4,t}^2 \end{pmatrix} \right] \otimes I_d.$$

We can rewrite Eq. (20) in terms of $\rho_t(x, v|y)$ as

$$q_t(x, v) = \int \pi(y)\rho_t(x, v|y) \, dy,$$

We note that the conditional distribution $\rho_t(v|x, y)$ remains Gaussian, with mean

$$\tilde{m}_t(x, y) = m_{3,t}y + \tilde{\Sigma}_{12,t}\tilde{\Sigma}_{11,t}^{-1} (x - m_{1,t}y),$$

which is linear in $y$ and covariance $\sigma_t^2 I$, where

$$\sigma_t^2 = \tilde{\Sigma}_{22,t} - \tilde{\Sigma}_{12,t}\tilde{\Sigma}_{11,t}^{-1}\tilde{\Sigma}_{12,t}.$$

This provides the following expression for the score

$$\nabla_v \log q_t(x, v) = \nabla_v \log q_t(v|x) = -\frac{v - \mathbb{E}_{Y|x,v}[\tilde{m}_t(x, Y)]}{\sigma_t^2} = -\frac{v - f_t\left(x, \mathbb{E}_{Y|x,v}[Y]\right)}{\sigma_t^2},$$

where

$$f_t\left(x, \mathbb{E}_{Y|x,v}[Y]\right) = m_{3,t}\mathbb{E}_{Y|x,v}[Y] + \tilde{\Sigma}_{12,t}\tilde{\Sigma}_{11,t}^{-1}\left(x - m_{1,t}\mathbb{E}_{Y|x,v}[Y]\right).$$

Using the derivations above, we define a multiscale SDE system that enables sampling from the target distribution via the reverse SDE, without requiring prior estimation of the denoiser $\mathbb{E}_{Y|x,v}[Y]$. We consider a fast process governed by an overdamped Langevin SDE targeting the conditional distribution of $Y$ given the current state $(\overleftarrow{X}_t, \overleftarrow{V}_t)$, which takes the form

$$dY_t = \frac{1}{\varepsilon}\left(\nabla \log \pi(Y_t) + \nabla_{Y_t} \log \rho_{T-t}(\overleftarrow{X}_t, \overleftarrow{V}_t|Y_t)\right) dt + \sqrt{\frac{2}{\varepsilon}}d\tilde{B}_t.$$

where

$$\nabla_{Y_t} \log \rho_{T-t}(\overleftarrow{X}_t, \overleftarrow{V}_t|Y_t) = -\left[(m_{1,T-t}Y_t, m_{3,T-t}Y_t) - (\overleftarrow{X}_t, \overleftarrow{V}_t)\right]\tilde{\Sigma}_{T-t}^{-1}\begin{pmatrix} m_{1,T-t} \\ m_{3,T-t} \end{pmatrix}.$$

Combining altogether, we arrive at the following multiscale system of SDEs

$$\begin{cases} d\overleftarrow{X}_t = -4\Gamma^{-2}\overleftarrow{V}_t \, dt \\ d\overleftarrow{V}_t = \left(\overleftarrow{X}_t + 4\Gamma^{-1}\overleftarrow{V}_t - \frac{2\Gamma}{\sigma_{T-t}^2}\left(\overleftarrow{V}_t - \left(m_{3,T-t}Y_t + \tilde{\Sigma}_{12,T-t}\tilde{\Sigma}_{11,T-t}^{-1}(\overleftarrow{X}_t - m_{1,T-t}Y_t)\right)\right)\right) dt + \sqrt{2\Gamma} \, dB_t \\ dY_t = \frac{1}{\varepsilon}\left(\nabla \log \pi(Y_t) + \nabla_{Y_t} \log \rho_{T-t}(\overleftarrow{X}_t, \overleftarrow{V}_t|Y_t)\right) dt + \sqrt{\frac{2}{\varepsilon}}d\tilde{B}_t. \end{cases}$$

$$(21)$$

Inspired by Dockhorn et al. [2022b], we propose a novel discretisation scheme which leverages symmetric splitting techniques [Leimkuhler and Matthews, 2013] justified by the symmetric Trotter splitting and the Baker–Campbell–Hausdorff formula [Strang, 1968, Trotter, 1959, Tuckerman, 2010]. The $l$-th iteration of the proposed numerical scheme consists of the following steps, with $h_l$ denoting the step size.

1. Evolve the following system of SDEs exactly for half-step $h_l/2$

$$\begin{cases} d\overleftarrow{X}_t = -4\Gamma^{-2}\overleftarrow{V}_t \, dt, \\ d\overleftarrow{V}_t = \left(\overleftarrow{X}_t + 4\Gamma^{-1}\overleftarrow{V}_t\right) dt + \sqrt{2\Gamma} \, dB_t. \end{cases} \tag{22}$$

2. Evolve the velocity component (full step) under the following ODE using Euler's method

$$d\overleftarrow{V}_t = -\frac{2\Gamma}{\sigma_{T-t}^2}\left(\overleftarrow{V}_t - \left(m_{3,T-t}Y_t + \tilde{\Sigma}_{12,T-t}\tilde{\Sigma}_{11,T-t}^{-1}\left(\overleftarrow{X}_t - m_{1,T-t}Y_t\right)\right)\right) dt.$$

3. Evolve the fast dynamics (full step) using SROCK method

$$dY_t = \frac{1}{\varepsilon}\left(\nabla\log\pi(Y_t) + \nabla_{Y_t}\log\rho_{T-t}(\overleftarrow{X}_t, \overleftarrow{V}_t|Y_t)\right) dt + \sqrt{\frac{2}{\varepsilon}}d\tilde{B}_t.$$

4. Evolve the system in Eq. (22) exactly for half-step $h_l/2$.

We summarise the steps above in the form of a concise algorithm. To do so, we first derive the exact solution of the SDE system (22) used in Step 1. Define the matrices

$$\hat{A} = \begin{pmatrix} 0 & -4\Gamma^{-2} \\ 1 & 4\Gamma^{-1} \end{pmatrix}, \qquad \hat{B} = \begin{pmatrix} 0 & 0 \\ 0 & \sqrt{2\Gamma} \end{pmatrix}.$$

The matrix exponential $e^{\hat{A}t}$ takes the form

$$e^{\hat{A}t} = \begin{pmatrix} 1 - 2\Gamma^{-1}t & -4\Gamma^{-2}t \\ t & 1 + 2\Gamma^{-1}t \end{pmatrix} e^{2\Gamma^{-1}t}.$$

The solution to the system (22) at time $t$, given an initial condition $(X_0, V_0)$, can be expressed as

$$(X_t, V_t) \sim \mathcal{N}(\hat{m}_t(X_0, V_0), \hat{\Sigma}_t \otimes I_d)$$

where the mean and covariance matrix are defined by

$$\hat{m}_t(X_0, V_0) = e^{\hat{A}t} \otimes I_d \begin{pmatrix} X_0 \\ V_0 \end{pmatrix}, \qquad \hat{\Sigma}_t = \begin{pmatrix} \hat{\Sigma}_{11,t} & \hat{\Sigma}_{12,t} \\ \hat{\Sigma}_{21,t} & \hat{\Sigma}_{22,t} \end{pmatrix} = \int_0^t e^{\hat{A}(t-s)}\hat{B}\hat{B}^\mathsf{T}\left[e^{\hat{A}(t-s)}\right]^\mathsf{T} \, ds. \tag{23}$$

The explicit expressions for the entries of the covariance matrix $\hat{\Sigma}_t$ are

$$\hat{\Sigma}_{11,t} = \left(-e^{4t\Gamma^{-1}} + 1 - 4t\Gamma^{-1} + 8t^2\Gamma^{-2}\right) e^{4t\Gamma^{-1}}$$

$$\hat{\Sigma}_{12,t} = \hat{\Sigma}_{21,t} = 4t^2\Gamma^{-1}e^{4t\Gamma^{-1}}$$

$$\hat{\Sigma}_{22,t} = \left(\frac{\Gamma^2}{4}\left(1 - e^{4t\Gamma^{-1}}\right) + t\Gamma + 2t^2\right) e^{4t\Gamma^{-1}}.$$

The full algorithm is presented in Algorithm 4.

**Challenges for heavy-tailed controlled diffusions.** When using a heavy-tailed noising process given by the convolution with a Student's t distribution with tail index $\alpha$, the forward underdamped diffusion takes the form

$$\begin{cases} d\overrightarrow{X}_t = M^{-1}\overrightarrow{V}_t \, dt \\ d\overrightarrow{V}_t = \left(-\frac{\alpha+d}{\alpha}\frac{\overrightarrow{X}_t}{1 + \frac{\|\overrightarrow{X}_t\|^2}{\alpha}} - \Gamma M^{-1}\overrightarrow{V}_t\right) dt + \sqrt{2\Gamma} \, dB_t. \end{cases}$$

The non-linear drift term in the velocity dynamics prevents us from obtaining analytic solutions via standard techniques for linear SDEs. As a result, we cannot directly reproduce the computations used in the case of an OU noising process. A detailed investigation of this regime is left for future work.

## C Theoretical guarantees

In this section, we provide detailed proofs of the theoretical results presented in the paper and discuss the additional challenges posed by heavy-tailed diffusions.

---

**Algorithm 4** MULTCDIFF sampler

---

**Require:** Friction coefficient $\Gamma$ (scalar), $M = \Gamma^2/4$, number of sampling steps $L$, time discretisation $0 = T_0 < \cdots < T_L = T$, step size $h_l = T_{l+1} - T_l$. Constants for SROCK step from (14), (15). Functions $\hat{m}_t(\cdot, \cdot)$ and $\hat{\Sigma}_t$ from (23).

**Initial samples** $X_0 \sim \mathcal{N}(0, I), V_0 \sim \mathcal{N}(0, MI), Y_0 \sim \mathcal{N}(0, I)$.

**for** $l = 0$ **to** $L$ **do**

$(X_{l+1/2}, V_{l+1/2}) \sim \mathcal{N}(\hat{m}_{h_l/2}(X_l, V_l), \hat{\Sigma}_{h_l/2})$

$V'_{l+1/2} = V_{l+1/2} - \frac{2\Gamma}{\sigma^2_{T-T_l}} \left( V_{l+1/2} - \left( m_{3,T-T_l} Y_l + \tilde{\Sigma}_{12,T-T_l} \tilde{\Sigma}^{-1}_{11,T-T_l} \left( X_{l+1/2} - m_{1,T-T_l} Y_l \right) \right) \right) \frac{h_l}{2}$

$$\begin{cases} K_{l,0} &= Y_l \\ K_{l,1} &= K_{l,0} + \mu_1 \frac{h_l}{\varepsilon} \left( -\nabla V_\pi(K_{l,0}) + \nabla_Y \log \rho_{T-T_l}(X_{l+1/2}, V'_{l+1/2}|K_{l,0}) \right) \\ &\vdots \\ K_{l,i} &= \mu_i \frac{h_l}{\varepsilon} \left( -\nabla V_\pi(K_{l,i-1}) + \nabla_Y \log \rho_{T-T_l}(X_{l+1/2}, V'_{l+1/2}|K_{l,i-1}) \right) + \nu_i K_{l,i-1} + \kappa_i K_{l,i-2} \\ &\vdots \\ Y_{l+1} &= K_s + \sqrt{\frac{2h}{\varepsilon}} \xi_l, \quad \xi_l \sim \mathcal{N}(0, I) \end{cases}$$

$(X_{l+1/2}, V_{l+1/2}) \sim \mathcal{N}(\hat{m}_{h_l/2}(X_{l+1}, V'_{l+1/2}), \hat{\Sigma}_{h_l/2})$

**end for**
**return** $(X_L, V_L)$

---

## C.1 Convergence of the slow-fast system to the averaged dynamics

Based on the stochastic averaging results from [Liu et al., 2020] presented in Appendix A, we restate our convergence result, Theorem 4.1, and provide the proof.

**Theorem C.1.** *[Theorem 4.1 restated] Let the base distribution $\nu \sim \mathcal{N}(0, I)$. Suppose the target distribution $\pi$ and the schedule function $\lambda_t$ satisfy assumptions **A1** and **A2**, respectively. Then, for any $\varepsilon \in (0, \varepsilon_0)$, where $\varepsilon_0$ is specified in the proof, and any given initial conditions, there exists unique solutions $\{(X^\varepsilon_t, Y^\varepsilon_t), t \geq 0\}$, $\{(X^\varepsilon_t, V^\varepsilon_t, Y^\varepsilon_t), t \geq 0\}$ and $\{(\overleftarrow{X}^\varepsilon_t, \overleftarrow{V}^\varepsilon_t, Y^\varepsilon_t), t \geq 0\}$ to the slow-fast stochastic systems (4), (6) and (10), respectively. Furthermore, for any $p > 0$, it holds that*

$$\lim_{\varepsilon \to 0} \mathbb{E}\left( \sup_{t \in [0,T]} \|X^\varepsilon_t - \bar{X}_t\|^p \right) = 0,$$

*where $\bar{X}_t$ denotes the solution of the averaged system and $X^\varepsilon_t$ is the corresponding slow component of the multiscale systems (4), (6) and (10).*

*Proof.* If the coefficients of the proposed slow-fast systems (4), (6) and (10) satisfy assumptions **A** 3 and **A4**, then the result follows directly from Theorems A.1 and A.2. Thus, we verify below that these assumptions hold in all cases.

It is important to mention that although the theorem specifically addresses the case where $\nu \sim \mathcal{N}(0, I)$, assumption **A**3 only requires that $\nabla \log \nu$ is Lipschitz continuous with constant $L_\nu$ and has a maximiser $x^\star$. Without loss of generality, we assume $x^\star = 0$, i.e., $\nabla \log \nu(0) = 0$. Therefore, we show that assumption **A**3 holds for general base distributions $\nu$ satisfying these properties.

Besides, for the systems based on annealed Langevin diffusion (4) and (6), we instead consider their modified versions (13) and (16), respectively, as these reflect the dynamics used in practice during implementation. We observed in Appendix B.1 that these systems reduced to

- A multiscale system with time-independent coefficients for $0 \leq \lambda_{\kappa t} < \lambda_\delta$.

- A multiscale system with time-dependent coefficients for $\lambda_\delta \leq \lambda_{\kappa t} < 1 - \lambda_\delta$. Here, we distinguish between two different dynamics depending on whether $\lambda_{\kappa t} < \tilde{\lambda}$ or $\lambda_{\kappa t} \geq \tilde{\lambda}$.

- Standard Langevin dynamics for $1 - \lambda_\delta \leq \lambda_{\kappa t} \leq 1$.

We analyse the first two cases, as standard Langevin dynamics is known to converge to its invariant distribution when initialised with a warm start [Chewi et al., 2021, Lee et al., 2021a, Wu et al., 2022]. See Appendix B.1 for further discussion.

We denote by $m^\star$ a minimiser of the target potential $V_\pi$.

**Overdamped MULTALMC** (4) **or** (13)  The coefficients of the slow-fast system for $\lambda_{\kappa t} \in [0, 1 - \lambda_\delta]$ have the following expressions

$$
b(t, x, y) = \begin{cases} \frac{1}{\sqrt{1-\lambda_\delta}} \nabla \log \nu \left( \frac{x-y}{\sqrt{1-\lambda_\delta}} \right) & \text{if } 0 \leq \lambda_{\kappa t} < \lambda_\delta \\ \frac{1}{\sqrt{1-\lambda_{\kappa t}}} \nabla \log \nu \left( \frac{x-y}{\sqrt{1-\lambda_{\kappa t}}} \right) & \text{if } \lambda_\delta \leq \lambda_{\kappa t} < \tilde{\lambda} \\ -\frac{1}{\sqrt{\lambda_{\kappa t}}} \nabla V_\pi \left( \frac{y}{\sqrt{\lambda_{\kappa t}}} \right) & \text{if } \tilde{\lambda} \leq \lambda_{\kappa t} < 1 - \lambda_\delta, \end{cases}
$$

$$
\sigma(t, x) = \sqrt{2},
$$

$$
f(t, x, y) = \begin{cases} \frac{1}{\sqrt{1-\lambda_\delta}} \nabla \log \nu \left( \frac{y-x}{\sqrt{1-\lambda_\delta}} \right) - \frac{1}{\sqrt{\lambda_\delta}} \nabla V_\pi \left( \frac{y}{\sqrt{\lambda_\delta}} \right) & \text{if } 0 \leq \lambda_{\kappa t} < \lambda_\delta \\ \frac{1}{\sqrt{1-\lambda_{\kappa t}}} \nabla \log \nu \left( \frac{y-x}{\sqrt{1-\lambda_{\kappa t}}} \right) - \frac{1}{\sqrt{\lambda_{\kappa t}}} \nabla V_\pi \left( \frac{y}{\sqrt{\lambda_{\kappa t}}} \right) & \text{if } \lambda_\delta \leq \lambda_{\kappa t} < 1 - \lambda_\delta, \end{cases}
\tag{24}
$$

$$
g(t, x, y) = \sqrt{2}.
$$

We consider two regimes based on the value of $\lambda_{\kappa t}$: Regime 1 $\lambda_{\kappa t} \in [0, \tilde{\lambda}]$ and Regime 2 $\lambda_{\kappa t} \in [\tilde{\lambda}, 1 - \lambda_\delta]$. We analyse the convergence within each regime separately.

Substituting the expressions of the coefficients, it follows that assumption **A3** holds with $\theta_1 = 0$, $\theta_2 = \theta_3 = \theta_5 = \theta_6 = 1$, $\theta_4 = 2$ and $Z_T = \|m^\star\| \sqrt{\tilde{\lambda}}/2$ is a constant random variable in both regimes, as shown below.

$$
2\|b(t, x_1, y) - b(t, x_2, y)\| \|x_1 - x_2\| + \|\sigma(t, x_1) - \sigma(t, x_2)\|^2
$$
$$
\leq \begin{cases} \mathbb{1}_{\{0 \leq \lambda_{\kappa t} < \lambda_\delta\}} \frac{2L_\nu}{1-\lambda_\delta} \|x_1 - x_2\|^2 + \mathbb{1}_{\{\lambda_\delta \leq \lambda_{\kappa t} < \tilde{\lambda}\}} \frac{2L_\nu}{1-\lambda_{\kappa t}} \|x_1 - x_2\|^2 & \text{Regime 1} \\ 0 & \text{Regime 2} \end{cases}
$$
$$
\leq \frac{2L_\nu}{1 - \tilde{\lambda}} \|x_1 - x_2\|^2,
$$

$$
\|b(t, x, y_1) - b(t, x, y_2)\| \leq \begin{cases} \mathbb{1}_{\{0 \leq \lambda_{\kappa t} < \lambda_\delta\}} \frac{L_\nu}{1-\lambda_\delta} \|y_1 - y_2\| + \mathbb{1}_{\{\lambda_\delta \leq \lambda_{\kappa t} < \tilde{\lambda}\}} \frac{L_\nu}{1-\lambda_{\kappa t}} \|y_1 - y_2\| & \text{Regime 1} \\ \mathbb{1}_{\{\tilde{\lambda} \leq \lambda_{\kappa t} < 1 - \lambda_\delta\}} \frac{L_\pi}{\lambda_{\kappa t}} \|y_1 - y_2\| & \text{Regime 2} \end{cases}
$$
$$
\leq \max \left\{ \frac{L_\nu}{1 - \tilde{\lambda}}, \frac{L_\pi}{\tilde{\lambda}} \right\} \|y_1 - y_2\|.
$$

For the following, we can assume without loss of generality that $s \leq t$

$\|b(t, x, y) - b(s, x, y)\|$
$$
\leq \begin{cases} \mathbb{1}_{\{0 \leq \lambda_{\kappa s} < \lambda_\delta \leq \lambda_{\kappa t} < \tilde{\lambda}\}} \left| \frac{1}{\sqrt{1-\lambda_{\kappa t}}} - \frac{1}{\sqrt{1-\lambda_\delta}} \right| \left( L_\nu \frac{\|x\|+\|y\|}{\sqrt{1-\tilde{\lambda}}} + \left\| \nabla \log \nu \left( \frac{x-y}{\sqrt{1-\lambda_{\kappa t}}} \right) \right\| \right) \\ \quad + \mathbb{1}_{\{\lambda_\delta \leq \lambda_{\kappa t}, \lambda_{\kappa s} < \tilde{\lambda}\}} \left| \frac{1}{\sqrt{1-\lambda_{\kappa t}}} - \frac{1}{\sqrt{1-\lambda_{\kappa s}}} \right| \left( L_\nu \frac{\|x\|+\|y\|}{\sqrt{1-\tilde{\lambda}}} + \left\| \nabla \log \nu \left( \frac{x-y}{\sqrt{1-\lambda_{\kappa t}}} \right) \right\| \right) & \text{Regime 1} \\ \mathbb{1}_{\{\tilde{\lambda} \leq \lambda_{\kappa t}, \lambda_{\kappa s} < 1 - \lambda_\delta\}} \left| \frac{1}{\sqrt{\lambda_{\kappa t}}} - \frac{1}{\sqrt{\lambda_{\kappa s}}} \right| \left( \frac{L_\pi \|y\|}{\sqrt{\tilde{\lambda}}} + \left\| \nabla V_\pi \left( \frac{y}{\sqrt{\lambda_{\kappa t}}} \right) \right\| \right) & \text{Regime 2} \end{cases}
$$
$$
\leq \begin{cases} \frac{C|t-s|^{\gamma_1}}{(1-\tilde{\lambda})^2} L_\nu (\|x\| + \|y\|) & \text{Regime 1} \\ \frac{C|t-s|^{\gamma_1}}{\tilde{\lambda}^2} L_\pi \left( \|y\| + \frac{\|m^\star\| \sqrt{\tilde{\lambda}}}{2} \right) & \text{Regime 2} \end{cases}
$$
$$
\leq C \max \left\{ \frac{L_\nu}{(1 - \tilde{\lambda})^2}, \frac{L_\pi}{\tilde{\lambda}^2} \right\} |t - s|^{\gamma_1} \left( \|y\| + \|x\| + \frac{\|m^\star\| \sqrt{\tilde{\lambda}}}{2} \right),
$$

$2\langle x, b(t, x, y) \rangle$

$$\leq \begin{cases} \mathbb{1}_{\{0 \leq \lambda_t < \lambda_\delta\}} \frac{2\|x\|}{\sqrt{1-\lambda_\delta}} \left\|\nabla \log \nu \left(\frac{x-y}{\sqrt{1-\lambda_\delta}}\right)\right\| + \mathbb{1}_{\{\lambda_\delta \leq \lambda_{\kappa t} < \tilde{\lambda}\}} \frac{2\|x\|}{\sqrt{1-\lambda_{\kappa t}}} \left\|\nabla \log \nu \left(\frac{x-y}{\sqrt{1-\lambda_{\kappa t}}}\right)\right\| & \text{Regime 1} \\ \mathbb{1}_{\{\tilde{\lambda} \leq \lambda_{\kappa t} < 1-\lambda_\delta\}} \frac{2\|x\|}{\sqrt{\lambda_{\kappa t}}} \left\|\nabla V_\pi \left(\frac{y}{\sqrt{\lambda_{\kappa t}}}\right)\right\| & \text{Regime 2} \end{cases}$$

$$\leq \begin{cases} \frac{L_\nu}{1-\tilde{\lambda}} \left(2\|x\|^2 + \|y\|^2\right) & \text{Regime 1} \\ \frac{L_\pi}{\tilde{\lambda}} \left(2\|x\|^2 + \|y\|^2 + \tilde{\lambda}\|m^\star\|^2\right) & \text{Regime 2} \end{cases}$$

$$\leq 2 \max\left\{\frac{L_\nu}{1-\tilde{\lambda}}, \frac{L_\pi}{\tilde{\lambda}}\right\} \max\{1, \tilde{\lambda}\|m^\star\|^2\} \left(1 + \|x\|^2\right) + \max\left\{\frac{L_\nu}{1-\tilde{\lambda}}, \frac{L_\pi}{\tilde{\lambda}}\right\} \|y\|^2,$$

which leads to the constant $\lambda_1$ in assumption **A3** being given by $\lambda_1 = \max\left\{\frac{L_\nu}{1-\tilde{\lambda}}, \frac{L_\pi}{\tilde{\lambda}}\right\}$.

$$\|b(t,x,y)\| \leq \begin{cases} \frac{L_\nu(\|x\|+\|y\|)}{1-\tilde{\lambda}} & \text{Regime 1} \\ \frac{L_\pi}{\tilde{\lambda}} \left(\|y\| + \sqrt{\tilde{\lambda}}\|m^\star\|\right) & \text{Regime 2} \end{cases}$$

$$\leq \frac{L_\pi}{\sqrt{\tilde{\lambda}}} \|m^\star\| + \max\left\{\frac{L_\nu}{1-\tilde{\lambda}}, \frac{L_\pi}{\tilde{\lambda}}\right\} (\|x\| + \|y\|),$$

$$\|\sigma(t,x)\|^2 = 2.$$

On the other hand, to verify that assumption **A4** holds, we recall that for a standard Normal distribution $\nabla \log \nu(x) = -x$. Besides, note that we can express the drift term of the fast process, $f(t,x,y)$, in a compact form by rewriting (24) as

$$f(t,x,y) = \frac{1}{\sqrt{1-\lambda'_{\kappa t}}} \nabla \log \nu \left(\frac{y-x}{\sqrt{1-\lambda'_{\kappa t}}}\right) - \frac{1}{\sqrt{\lambda'_{\kappa t}}} \nabla V_\pi \left(\frac{y}{\sqrt{\lambda'_{\kappa t}}}\right),$$

where the schedule $\lambda'_{\kappa t}$ is defined by

$$\lambda'_{\kappa t} = \begin{cases} \lambda_\delta & \text{if } 0 \leq \lambda_{\kappa t} < \lambda_\delta \\ \lambda_{\kappa t} & \text{if } \lambda_\delta \leq \lambda_{\kappa t} < 1-\lambda_\delta. \end{cases}$$

Therefore, we adopt this compact form in the following analysis and omit explicit distinctions between regimes to simplify notation. Using this, we have

$$2\langle f(t,x,y_1) - f(t,x,y_2), y_1 - y_2\rangle + \|g(t,x,y_1) - g(t,x,y_2)\|^2$$

$$\leq -\frac{2}{1-\lambda'_{\kappa t}} \|y_1 - y_2\|^2 - \frac{2}{\sqrt{\lambda'_{\kappa t}}} \left\langle \nabla V_\pi \left(\frac{y_1}{\sqrt{\lambda'_{\kappa t}}}\right) - \nabla V_\pi \left(\frac{y_2}{\sqrt{\lambda'_{\kappa t}}}\right), y_1 - y_2\right\rangle$$

$$\leq -\left(\frac{2}{1-\lambda'_{\kappa t}} + \frac{2M_\pi}{\lambda'_{\kappa t}}\right) \|y_1 - y_2\|^2 \leq -\beta_\delta \|y_1 - y_2\|^2,$$

$$\|f(t,x_1,y) - f(s,x_2,y)\| \leq \left|\frac{1}{1-\lambda'_{\kappa t}} - \frac{1}{1-\lambda'_{\kappa s}}\right| (\|y\| + \|x_2\|) + \frac{\|x_1 - x_2\|}{1-\lambda'_{\kappa t}}$$

$$+ \left|\frac{1}{\sqrt{\lambda'_{\kappa t}}} - \frac{1}{\sqrt{\lambda'_{\kappa s}}}\right| \left(\frac{L_\pi\|y\|}{\sqrt{\lambda'_{\kappa s}}} + \left\|\nabla V_\pi \left(\frac{y}{\sqrt{\lambda'_{\kappa t}}}\right)\right\|\right)$$

$$\leq C_{1,\delta}|t-s|^{\gamma_1} (\|y\| + \|x_2\|) + C_{2,\delta}\|x_1 - x_2\| + C_{3,\delta}|t-s|^{\gamma_1}\|y\|$$

$$\leq C_\delta(|t-s|^{\gamma_1} + \|x_1 - x_2\|)(1 + \|x_2\| + \|y\|),$$

$$\|f(t,x,y)\| \leq \max\left\{\frac{L_\pi}{\lambda'_{\kappa t}(1-\lambda'_{\kappa t})}, \frac{1}{1-\lambda'_{\kappa t}}, \frac{L_\pi\|m^\star\|}{\lambda'_{\kappa t}}\right\} (1 + \|x\| + \|y\|) \leq C_\delta (1 + \|x\| + \|y\|),$$

$$\|g(t,x_1,y_1) - g(s,x_2,y_2)\| = 0, \quad \|g(t,x,y)\| = \sqrt{2}.$$

Since the potential $V_\pi$ is strongly convex with constant $M_\pi$, then $V_\pi$ satisfies the dissipativity inequality $\langle \nabla V_\pi(x), x\rangle \geq a_\pi\|x\|^2 - b_\pi$. Therefore, we have that for any $k \geq 2$

$$2\langle y, f(t,x,y)\rangle + (k-1)\|g(t,x,y)\|^2 \leq -2\left\langle y, \frac{y-x}{1-\lambda'_{\kappa t}} + \frac{1}{\sqrt{\lambda'_{\kappa t}}} \nabla V_\pi \left(\frac{y}{\sqrt{\lambda'_{\kappa t}}}\right)\right\rangle + 2(k-1)$$

$$\leq -\frac{2}{1-\lambda'_{\kappa t}} \left(\|y\|^2 - \langle y, x\rangle\right) - \frac{2a_\pi}{\lambda'_{\kappa t}}\|y\|^2 + b_\pi + 2(k-1)$$

$$\leq -\left(\frac{1}{1-\lambda'_{\kappa t}} + \frac{2a_\pi}{\lambda'_{\kappa t}}\right)\|y\|^2 + \frac{1}{1-\lambda'_{\kappa t}}\|x\|^2 + b_\pi + 2(k-1)$$

$$\leq -\lambda_2\|y\|^2 + C_{k,\delta}(\|x\|^2 + 1),$$

where $\lambda_2$ is given by

$$\lambda_2 = \frac{1}{1-\lambda_\delta} + \frac{2a_\pi}{\lambda_\delta}.$$

This concludes the proof for the overdamped MULTALMC, where the constant $\varepsilon_0$ in the theorem statement is explicitly given by

$$\varepsilon_0 = \frac{\lambda_2}{\lambda_1} = \frac{\frac{1}{1-\lambda_\delta} + \frac{2a_\pi}{\lambda_\delta}}{\max\left\{\frac{L_\nu}{1-\tilde\lambda}, \frac{L_\pi}{\tilde\lambda}\right\}}.$$

**Underdamped MULTALMC** (6) **or** (16)    In this case, since we introduce an auxiliary velocity variable, the slow component consists of both position and velocity, $(x, v)$. Accordingly, the coefficients of the slow-fast system are given by the following expressions

$$b(t, (x,v), y) = \begin{cases} \begin{pmatrix} M^{-1}v \\ \frac{1}{\sqrt{1-\lambda_\delta}}\nabla\log\nu\left(\frac{x-y}{\sqrt{1-\lambda_\delta}}\right) - \Gamma M^{-1}v \end{pmatrix} & \text{if } 0 \leq \lambda_{\kappa t} < \lambda_\delta \\[2ex] \begin{pmatrix} M^{-1}v \\ \frac{1}{\sqrt{1-\lambda_{\kappa t}}}\nabla\log\nu\left(\frac{x-y}{\sqrt{1-\lambda_{\kappa t}}}\right) - \Gamma M^{-1}v \end{pmatrix} & \text{if } \lambda_\delta \leq \lambda_{\kappa t} < \tilde\lambda \\[2ex] \begin{pmatrix} M^{-1}v \\ -\frac{1}{\sqrt{\lambda_{\kappa t}}}\nabla V_\pi\left(\frac{y}{\sqrt{\lambda_{\kappa t}}}\right) - \Gamma M^{-1}v \end{pmatrix} & \text{if } \tilde\lambda \leq \lambda_{\kappa t} < 1 - \lambda_\delta, \end{cases}$$

$$\sigma(t, (x,v)) = \begin{pmatrix} 0 & 0 \\ 0 & \sqrt{2\Gamma} \end{pmatrix} \otimes I_d,$$

$$f(t, (x,v), y) = \begin{cases} \frac{1}{\sqrt{1-\lambda_\delta}}\nabla\log\nu\left(\frac{y-x}{\sqrt{1-\lambda_\delta}}\right) - \frac{1}{\sqrt{\lambda_\delta}}\nabla V_\pi\left(\frac{y}{\sqrt{\lambda_\delta}}\right) & \text{if } 0 \leq \lambda_{\kappa t} < \lambda_\delta \\[2ex] \frac{1}{\sqrt{1-\lambda_{\kappa t}}}\nabla\log\nu\left(\frac{y-x}{\sqrt{1-\lambda_{\kappa t}}}\right) - \frac{1}{\sqrt{\lambda_{\kappa t}}}\nabla V_\pi\left(\frac{y}{\sqrt{\lambda_{\kappa t}}}\right) & \text{if } \lambda_\delta \leq \lambda_{\kappa t} < 1 - \lambda_\delta, \end{cases}$$

$$g(t, (x,v), y) = \sqrt{2}.$$

Since the coefficients of the fast process are the same as in the overdamped setting, then assumption **A4** also holds in the underdamped case. To verify assumption **A3**, we adopt the same two-regime framework defined in the overdamped case and analyse convergence within each regime separately.

$$2\|b(t, (x_1, v_1), y) - b(t, (x_2, v_2), y)\|\|(x_1, v_1) - (x_2, v_2)\| + \|\sigma(t, x_1) - \sigma(t, x_2)\|^2$$

$$\leq \begin{cases} 2\left\|\begin{pmatrix} M^{-1}(v_1 - v_2) \\ \mathbb{1}_{\{0\leq\lambda_{\kappa t}<\lambda_\delta\}}\frac{1}{\sqrt{1-\lambda_\delta}}\left(\nabla\log\nu\left(\frac{x_1-y}{\sqrt{1-\lambda_\delta}}\right) - \nabla\log\nu\left(\frac{x_2-y}{\sqrt{1-\lambda_\delta}}\right)\right) - \Gamma M^{-1}(v_1 - v_2) \end{pmatrix} \right. \\ \left. + \begin{pmatrix} 0 \\ \mathbb{1}_{\{\lambda_\delta\leq\lambda_{\kappa t}<\tilde\lambda\}}\frac{1}{\sqrt{1-\lambda_{\kappa t}}}\left(\nabla\log\nu\left(\frac{x_1-y}{\sqrt{1-\lambda_{\kappa t}}}\right) - \nabla\log\nu\left(\frac{x_2-y}{\sqrt{1-\lambda_{\kappa t}}}\right)\right) \end{pmatrix}\right\|\|(x_1, v_1) - (x_2, v_2)\| \quad \text{Regime 1} \\ 0 \qquad\qquad\qquad\qquad\qquad\qquad\qquad\qquad\qquad\qquad\qquad\qquad\qquad\qquad\qquad\qquad\qquad \text{Regime 2} \end{cases}$$

$$\leq 2\sqrt{M^{-2}(1 + 2\Gamma^2)\|v_1 - v_2\|^2 + \frac{2L_\nu^2}{(1-\tilde\lambda)^2}\|x_1 - x_2\|^2}\,\|(x_1, v_1) - (x_2, v_2)\|$$

$$\leq 2\max\left\{M^{-1}\sqrt{1 + 2\Gamma^2}, \frac{\sqrt{2}L_\nu}{1-\tilde\lambda}\right\}\|(x_1, v_1) - (x_2, v_2)\|^2$$

$$\|b(t, (x,v), y_1) - b(t, (x,v), y_2)\|$$

$$\leq \begin{cases} \left\|\begin{pmatrix} 0 \\ \frac{\mathbb{1}_{\{0\leq\lambda_{\kappa t}<\lambda_\delta\}}}{\sqrt{1-\lambda_\delta}}\left(\nabla\log\nu\left(\frac{x-y_1}{\sqrt{1-\lambda_\delta}}\right) - \nabla\log\nu\left(\frac{x-y_2}{\sqrt{1-\lambda_\delta}}\right)\right) \end{pmatrix}\right. \\ \left. + \begin{pmatrix} 0 \\ \frac{\mathbb{1}_{\{\lambda_\delta\leq\lambda_{\kappa t}<\tilde\lambda\}}}{\sqrt{1-\lambda_{\kappa t}}}\left(\nabla\log\nu\left(\frac{x-y_1}{\sqrt{1-\lambda_{\kappa t}}}\right) - \nabla\log\nu\left(\frac{x-y_2}{\sqrt{1-\lambda_{\kappa t}}}\right)\right) \end{pmatrix}\right\| \quad \text{Regime 1} \\ \left\|\begin{pmatrix} 0 \\ \frac{1}{\sqrt{\lambda_{\kappa t}}}\left(\nabla V_\pi\left(\frac{y_1}{\sqrt{\lambda_{\kappa t}}}\right) - \nabla V_\pi\left(\frac{y_2}{\sqrt{\lambda_{\kappa t}}}\right)\right) \end{pmatrix}\right\| \qquad\qquad \text{Regime 2} \end{cases}$$

$$\leq \max\left\{\frac{L_\nu}{1-\tilde\lambda}, \frac{L_\pi}{\tilde\lambda}\right\} \|y_1 - y_2\|.$$

For the following, we can assume without loss of generality that $s \leq t$

$\|b(t,(x,v),y) - b(s,(x,v),y)\|$

$$\leq \begin{cases} \mathbb{1}_{\{0\leq\lambda_{\kappa s}<\lambda_\delta\leq\lambda_{\kappa t}<\tilde\lambda\}}\left|\frac{1}{\sqrt{1-\lambda_{\kappa t}}} - \frac{1}{\sqrt{1-\lambda_\delta}}\right|\left(L_\nu\frac{\|x\|+\|y\|}{\sqrt{1-\tilde\lambda}} + \left\|\nabla\log\nu\left(\frac{x-y}{\sqrt{1-\lambda_{\kappa t}}}\right)\right\|\right) \\ \quad +\mathbb{1}_{\{\lambda_\delta\leq\lambda_{\kappa t},\lambda_{\kappa s}<\tilde\lambda\}}\left|\frac{1}{\sqrt{1-\lambda_{\kappa t}}} - \frac{1}{\sqrt{1-\lambda_{\kappa s}}}\right|\left(L_\nu\frac{\|x\|+\|y\|}{\sqrt{1-\tilde\lambda}} + \left\|\nabla\log\nu\left(\frac{x-y}{\sqrt{1-\lambda_{\kappa t}}}\right)\right\|\right) & \text{Regime 1} \\ \mathbb{1}_{\{\tilde\lambda\leq\lambda_{\kappa t},\lambda_{\kappa s}<1-\lambda_\delta\}}\left|\frac{1}{\sqrt{\lambda_{\kappa t}}} - \frac{1}{\sqrt{\lambda_{\kappa s}}}\right|\left(\frac{L_\pi\|y\|}{\sqrt{\tilde\lambda}} + \left\|\nabla V_\pi\left(\frac{y}{\sqrt{\lambda_{\kappa t}}}\right)\right\|\right) & \text{Regime 2} \end{cases}$$

$$\leq \begin{cases} \frac{C|t-s|^{\gamma_1}}{(1-\tilde\lambda)^2}L_\nu(\|x\|+\|y\|) & \text{Regime 1} \\ \frac{C|t-s|^{\gamma_1}}{\tilde\lambda^2}L_\pi\left(\|y\| + \frac{\|m^\star\|\sqrt{\tilde\lambda}}{2}\right) & \text{Regime 2} \end{cases}$$

$$\leq C\max\left\{\frac{L_\nu}{(1-\tilde\lambda)^2}, \frac{L_\pi}{\tilde\lambda^2}\right\}|t-s|^{\gamma_1}\left(\|y\| + \|x\| + \frac{\|m^\star\|\sqrt{\tilde\lambda}}{2}\right),$$

$2\langle(x,v), b(t,(x,v),y)\rangle$

$$\leq \begin{cases} 2M^{-1}\langle x,v\rangle - 2\Gamma M^{-1}\|v\|^2 + \mathbb{1}_{\{0\leq\lambda_{\kappa t}<\lambda_\delta\}}\frac{2\|v\|}{\sqrt{1-\lambda_\delta}}\left\|\nabla\log\nu\left(\frac{x-y}{\sqrt{1-\lambda_\delta}}\right)\right\| \\ \quad +\mathbb{1}_{\{\lambda_\delta\leq\lambda_{\kappa t}<\tilde\lambda\}}\frac{2\|v\|}{\sqrt{1-\lambda_{\kappa t}}}\left\|\nabla\log\nu\left(\frac{x-y}{\sqrt{1-\lambda_{\kappa t}}}\right)\right\| & \text{Regime 1} \\ 2M^{-1}\langle x,v\rangle - 2\Gamma M^{-1}\|v\|^2 + \frac{2\|v\|}{\sqrt{\lambda_{\kappa t}}}\left\|\nabla V_\pi\left(\frac{y}{\sqrt{\lambda_{\kappa t}}}\right)\right\| & \text{Regime 2} \end{cases}$$

$$\leq \begin{cases} M^{-1}(\|x\|^2+\|v\|^2) - 2\Gamma M^{-1}\|v\|^2 + \frac{L_\nu}{1-\tilde\lambda}\left(2\|v\|^2 + \|x\|^2 + \|y\|^2\right) & \text{Regime 1} \\ M^{-1}(\|x\|^2+\|v\|^2) - 2\Gamma M^{-1}\|v\|^2 + \frac{L_\pi}{\tilde\lambda}\left(2\|v\|^2 + \|y\|^2 + \tilde\lambda\|m^\star\|^2\right) & \text{Regime 2} \end{cases}$$

$$\leq \max\left\{M^{-1}, \frac{2L_\nu}{1-\tilde\lambda}, \frac{2L_\pi}{\tilde\lambda}\right\}\max\{1,\tilde\lambda\|m^\star\|^2\}\left(1 + \|x\|^2 + \|v\|^2\right) + \max\left\{\frac{L_\nu}{1-\tilde\lambda}, \frac{L_\pi}{\tilde\lambda}\right\}\|y\|^2,$$

$\|b(t,(x,v),y)\|$

$$\leq \begin{cases} \sqrt{M^{-2}(1+2\Gamma^2)\|v\|^2 + \mathbb{1}_{\{0\leq\lambda_{\kappa t}<\lambda_\delta\}}\frac{2}{1-\tilde\lambda}\left\|\nabla\log\nu\left(\frac{x-y}{\sqrt{1-\lambda_\delta}}\right)\right\|^2 + \mathbb{1}_{\{\lambda_\delta\leq\lambda_{\kappa t}<\tilde\lambda\}}\frac{2}{1-\tilde\lambda}\left\|\nabla\log\nu\left(\frac{x-y}{\sqrt{1-\lambda_{\kappa t}}}\right)\right\|^2} & \text{Regime 1} \\ \sqrt{M^{-2}(1+2\Gamma^2)\|v\|^2 + \frac{2}{\tilde\lambda}\left\|\nabla V_\pi\left(\frac{y}{\sqrt{\lambda_{\kappa t}}}\right)\right\|^2} & \text{Regime 2} \end{cases}$$

$$\leq \max\left\{M^{-1}\sqrt{1+2\Gamma^2}, \frac{2L_\nu}{1-\tilde\lambda}, \frac{2L_\pi}{\tilde\lambda}\right\}\|(x,v)\| + \frac{2L_\pi\|m^\star\|}{\sqrt{\tilde\lambda}} + 2\max\left\{\frac{L_\nu}{1-\tilde\lambda}, \frac{L_\pi}{\tilde\lambda}\right\}\|y\|$$

$$\leq \frac{2L_\pi}{\sqrt{\tilde\lambda}}\|m^\star\| + \max\left\{M^{-1}\sqrt{1+2\Gamma^2}, \frac{2L_\nu}{1-\tilde\lambda}, \frac{2L_\pi}{\tilde\lambda}\right\}(\|(x,v)\| + \|y\|),$$

$$\|\sigma(t,x)\|^2 = 2\Gamma.$$

This concludes that assumption **A3** holds with $\theta_1 = 0$, $\theta_2 = \theta_3 = \theta_5 = \theta_6 = 1$, $\theta_4 = 2$ and $Z_T = \|m^\star\|\sqrt{\tilde\lambda}/2$ is a constant random variable in both regimes. Besides, the constant $\varepsilon_0$ in the theorem statement is given by

$$\varepsilon_0 = \frac{\lambda_2}{\lambda_1} = \frac{\frac{1}{1-\lambda_\delta} + \frac{2a_\pi}{\lambda_\delta}}{\max\left\{\frac{L_\nu}{1-\tilde\lambda}, \frac{L_\pi}{\tilde\lambda}\right\}}.$$

**Underdamped MULTCDIFF** (10) **or** (21) From Appendix B.2, the coefficients of the slow-fast system are given by the following expressions

$$b(t,(x,v),y) = \begin{pmatrix} -4\Gamma^{-2}v \\ x + 4\Gamma^{-1}v - \frac{2\Gamma}{\sigma_{T-t}^2}\left(v - \left(m_{3,T-t}y + \tilde\Sigma_{12,T-t}\tilde\Sigma_{11,T-t}^{-1}(x - m_{1,T-t}y)\right)\right) \end{pmatrix}$$

$$\sigma(t,(x,v)) = \begin{pmatrix} 0 & 0 \\ 0 & \sqrt{2\Gamma} \end{pmatrix} \otimes I_d,$$

$$f(t, (x, v), y) = \nabla \log \pi(y) - [(m_{1,T-t}y, m_{3,T-t}y) - (x, v)] \tilde{\Sigma}_{T-t}^{-1} \begin{pmatrix} m_{1,T-t} \\ m_{3,T-t} \end{pmatrix},$$

$$g(t, (x, v), y) = \sqrt{2}.$$

where $m_{1,T-t}, m_{3,T-t}, \tilde{\Sigma}_{T-t}$ are defined in Appendix B.2. Substituting these coefficients, it follows that assumption **A3** holds with $\theta_1 = 0$, $\theta_2 = \theta_3 = \theta_5 = \theta_6 = 1$, $\theta_4 = 2$ and $Z_T$ is a constant random variable, as shown below.

$2\|b(t, (x_1, v_1), y) - b(t, (x_2, v_2), y)\| \|(x_1, v_1) - (x_2, v_2)\| + \|\sigma(t, (x_1, v_1)) - \sigma(t, (x_2, v_2))\|^2$

$$\leq 2 \left\| \left( (x_1 - x_2) \left( 1 + \frac{2\Gamma}{\sigma_{T-t}^2} \tilde{\Sigma}_{12,T-t} \tilde{\Sigma}_{11,T-t}^{-1} \right) + (v_1 - v_2) \left( 4\Gamma^{-1} - \frac{2\Gamma}{\sigma_{T-t}^2} \right) \right) \right\| \|(x_1, v_1) - (x_2, v_2)\|$$

$$\leq 2 \sqrt{2 \left( 1 + \frac{2\Gamma}{\sigma_{T-t}^2} \tilde{\Sigma}_{12,T-t} \tilde{\Sigma}_{11,T-t}^{-1} \right)^2 \|x_1 - x_2\|^2 + \left( 16\Gamma^{-4} + 2 \left( 4\Gamma^{-1} - \frac{2\Gamma}{\sigma_{T-t}^2} \right)^2 \right) \|v_1 - v_2\|^2} \ \|(x_1, v_1) - (x_2, v_2)\|$$

$$\leq C_t \|(x_1, v_1) - (x_2, v_2)\|^2,$$

$\|b(t, (x, v), y_1) - b(t, (x, v), y_2)\|$

$$\leq \left\| \left( \begin{matrix} 0 \\ \frac{2\Gamma}{\sigma_{T-t}^2} \left( m_{3,T-t} + \tilde{\Sigma}_{12,T-t} \tilde{\Sigma}_{11,T-t}^{-1} m_{1,T-t} \right) (y_1 - y_2) \end{matrix} \right) \right\| \leq C_t \|y_1 - y_2\|,$$

$\|b(t, (x, v), y) - b(s, (x, v), y)\|$

$$\leq 2\Gamma \left[ \left| \frac{1}{\sigma_{T-t}^2} - \frac{1}{\sigma_{T-s}^2} \right| \|v\| + \left| \frac{\tilde{\Sigma}_{12,T-t} \tilde{\Sigma}_{11,T-t}^{-1}}{\sigma_{T-t}^2} - \frac{\tilde{\Sigma}_{12,T-s} \tilde{\Sigma}_{11,T-s}^{-1}}{\sigma_{T-s}^2} \right| \|x\| \right.$$

$$\left. + \left| \frac{m_{3,T-t} - \tilde{\Sigma}_{12,T-t} \tilde{\Sigma}_{11,T-t}^{-1} m_{1,T-t}}{\sigma_{T-t}^2} - \frac{m_{3,T-s} - \tilde{\Sigma}_{12,T-s} \tilde{\Sigma}_{11,T-s}^{-1} m_{1,T-s}}{\sigma_{T-s}^2} \right| \|y\| \right]$$

$$\leq C_T |t - s|^{\tilde{\gamma}_1} (\|y\| + \|(x, v)\|),$$

where we have used the expressions in Appendix B.2, which imply the existence of a constant $\tilde{\gamma}_1 \in (0, 1]$ such that

$$\left| \frac{1}{\sigma_{T-t}^2} - \frac{1}{\sigma_{T-s}^2} \right|, \left| \frac{\tilde{\Sigma}_{12,T-t} \tilde{\Sigma}_{11,T-t}^{-1}}{\sigma_{T-t}^2} - \frac{\tilde{\Sigma}_{12,T-s} \tilde{\Sigma}_{11,T-s}^{-1}}{\sigma_{T-s}^2} \right|,$$

$$\left| \frac{m_{3,T-t} - \tilde{\Sigma}_{12,T-t} \tilde{\Sigma}_{11,T-t}^{-1} m_{1,T-t}}{\sigma_{T-t}^2} - \frac{m_{3,T-s} - \tilde{\Sigma}_{12,T-s} \tilde{\Sigma}_{11,T-s}^{-1} m_{1,T-s}}{\sigma_{T-s}^2} \right| \leq C_T |t - s|^{\tilde{\gamma}_1}.$$

$2\langle (x, v), b(t, (x, v), y) \rangle$

$$\leq 2 \left( 1 - 4\Gamma^{-2} + \frac{2\Gamma}{\sigma_{T-t}^2} \tilde{\Sigma}_{12,T-t} \tilde{\Sigma}_{11,T-t}^{-1} \right) \langle x, v \rangle + 2 \left( 4\Gamma^{-1} - \frac{2\Gamma}{\sigma_{T-t}^2} \right) \|v\|^2$$

$$+ \frac{4\Gamma}{\sigma_{T-t}^2} \left( m_{3,T-t} - \tilde{\Sigma}_{12,T-t} \tilde{\Sigma}_{11,T-t}^{-1} m_{1,T-t} \right) \langle y, v \rangle$$

$$\leq \max \left\{ \left( 1 - 4\Gamma^{-2} + \frac{2\Gamma}{\sigma_{T-t}^2} \tilde{\Sigma}_{12,T-t} \tilde{\Sigma}_{11,T-t}^{-1} \right), 2 \left( 4\Gamma^{-1} - \frac{2\Gamma}{\sigma_{T-t}^2} \right), \frac{2\Gamma}{\sigma_{T-t}^2} \left( m_{3,T-t} - \tilde{\Sigma}_{12,T-t} \tilde{\Sigma}_{11,T-t}^{-1} m_{1,T-t} \right) \right\} \|(x, v)\|^2$$

$$+ \frac{2\Gamma}{\sigma_{T-t}^2} \left( m_{3,T-t} - \tilde{\Sigma}_{12,T-t} \tilde{\Sigma}_{11,T-t}^{-1} m_{1,T-t} \right) \|y\|^2 \leq C_t \|(x, v)\|^2 + \lambda_1 \|y\|^2,$$

$\|b(t, (x, v), y)\|$

$$\leq \sqrt{3 \left( 1 + \frac{2\Gamma}{\sigma_{T-t}^2} \tilde{\Sigma}_{12,T-t} \tilde{\Sigma}_{11,T-t}^{-1} \right)^2 \|x\|^2 + \left( 16\Gamma^{-4} + 3 \left( 4\Gamma^{-1} - \frac{2\Gamma}{\sigma_{T-t}^2} \right)^2 \right) \|v\|^2 + \frac{12\Gamma^2}{\sigma_{T-t}^4} \left( m_{3,T-t} - \tilde{\Sigma}_{12,T-t} \tilde{\Sigma}_{11,T-t}^{-1} m_{1,T-t} \right)^2 \|y\|^2}$$

$$\leq \max \left\{ \sqrt{3} \left| 1 + \frac{2\Gamma}{\sigma_{T-t}^2} \tilde{\Sigma}_{12,T-t} \tilde{\Sigma}_{11,T-t}^{-1} \right|, 4\Gamma^{-2} + \sqrt{3} \left| 4\Gamma^{-1} - \frac{2\Gamma}{\sigma_{T-t}^2} \right| \right\} \|(x, v)\| + \frac{2\sqrt{3}\Gamma}{\sigma_{T-t}^2} \left| m_{3,T-t} - \tilde{\Sigma}_{12,T-t} \tilde{\Sigma}_{11,T-t}^{-1} m_{1,T-t} \right| \|y\|$$

$$\leq C_T (\|(x, v)\| + \|y\|),$$

$$\|\sigma(t,x)\|^2 = 2\Gamma.$$

We finally check that assumption **A4** holds.

$$2\langle f(t,(x,v),y_1) - f(t,(x,v),y_2), y_1 - y_2\rangle + \|g(t,(x,v),y_1) - g(t,(x,v),y_2)\|^2$$

$$\leq -2\langle \nabla V_\pi(y_1) - \nabla V_\pi(y_2), y_1 - y_2\rangle - 2(m_{1,T-t}, m_{3,T-t})\tilde{\Sigma}_{T-t}^{-1}\begin{pmatrix} m_{1,T-t} \\ m_{3,T-t}\end{pmatrix}\|y_1 - y_2\|^2$$

$$\leq -2\left(M_\pi + \left(m_{1,T-t}^2 + m_{3,T-t}^2\right)\lambda_{\min}(\tilde{\Sigma}_{T-t}^{-1})\right)\|y_1 - y_2\|^2 \leq -\beta\|y_1 - y_2\|^2,$$

where we have used that, for all $t$, the precision matrix $\tilde{\Sigma}_{T-t}^{-1}$ is positive definite and $\lambda_{\min}(\tilde{\Sigma}_{T-t}^{-1})$ denotes its smallest eigenvalue.

$$\|f(t,(x_1,v_1),y) - f(s,(x_2,v_2),y)\| \leq \left|(m_{1,T-t}, m_{3,T-t})\tilde{\Sigma}_{T-t}^{-1}\begin{pmatrix} m_{1,T-t} \\ m_{3,T-t}\end{pmatrix} - (m_{1,T-s}, m_{3,T-s})\tilde{\Sigma}_{T-s}^{-1}\begin{pmatrix} m_{1,T-s} \\ m_{3,T-s}\end{pmatrix}\right|\|y\|$$

$$+ \|(x_1,v_1) - (x_2,v_2)\|\left\|\tilde{\Sigma}_{T-t}^{-1}\begin{pmatrix} m_{1,T-t} \\ m_{3,T-t}\end{pmatrix}\right\| + \|(x_2,v_2)\|\left\|\tilde{\Sigma}_{T-t}^{-1}\begin{pmatrix} m_{1,T-t} \\ m_{3,T-t}\end{pmatrix} - \tilde{\Sigma}_{T-s}^{-1}\begin{pmatrix} m_{1,T-s} \\ m_{3,T-s}\end{pmatrix}\right\|$$

$$\leq C_T|t-s|^{\tilde{\gamma}_2} + C_T\|(x_1,v_1) - (x_2,v_2)\| + C_T\|(x_2,v_2)\||t-s|^{\tilde{\gamma}_2}$$

$$\leq C_T\left(|t-s|^{\tilde{\gamma}_2} + \|(x_1,v_1) - (x_2,v_2)\|\right)(1 + \|(x_2,v_2)\|),$$

where the existence of $\tilde{\gamma}_2 \in (0,1]$ follows from the expressions for $m_{1,T-t}, m_{3,T-t}, \tilde{\Sigma}_{T-t}$ provided in Appendix B.2.

$$\|f(t,(x,v),y)\| \leq \|\nabla\log\pi(y)\| + (m_{1,T-t}, m_{3,T-t})\tilde{\Sigma}_{T-t}^{-1}\begin{pmatrix} m_{1,T-t} \\ m_{3,T-t}\end{pmatrix}\|y\| + \left\|(x,v)\tilde{\Sigma}_{T-t}^{-1}\begin{pmatrix} m_{1,T-t} \\ m_{3,T-t}\end{pmatrix}\right\|$$

$$\leq L_\pi(\|y\| + \|m^\star\|) + \sqrt{m_{1,T-t}^2 + m_{3,T-t}^2}\,\lambda_{\max}(\tilde{\Sigma}_{T-t}^{-1})\left(\sqrt{m_{1,T-t}^2 + m_{3,T-t}^2}\|y\| + \|(x,v)\|\right)$$

$$\leq C_T(1 + \|y\| + \|(x,v)\|),$$

$$\|g(t,(x_1,v_1),y_1) - g(s,(x_2,v_2),y_2)\| = 0, \quad \|g(t,(x,v),y)\| = \sqrt{2}.$$

Recalling that the strong convexity of $V_\pi$ implies a dissipativity inequality, we have that for any $k \geq 2$

$$2\langle y, f(t,(x,v),y)\rangle + (k-1)\|g(t,(x,v),y)\|^2 = -2\langle y, \nabla V_\pi(y)\rangle - 2(m_{1,T-t}, m_{3,T-t})\tilde{\Sigma}_{T-t}^{-1}\begin{pmatrix} m_{1,T-t} \\ m_{3,T-t}\end{pmatrix}\|y\|^2$$

$$+ 2\left\langle (x,v)\tilde{\Sigma}_{T-t}^{-1}\begin{pmatrix} m_{1,T-t} \\ m_{3,T-t}\end{pmatrix}, y\right\rangle + 2(k-1)$$

$$\leq -2(a_\pi\|y\|^2 - b_\pi) - (m_{1,T-t}, m_{3,T-t})\tilde{\Sigma}_{T-t}^{-1}\begin{pmatrix} m_{1,T-t} \\ m_{3,T-t}\end{pmatrix}\|y\|^2 + \|(x,v)\|^2\left\|\tilde{\Sigma}_{T-t}^{-1/2}\right\|^2 + 2(k-1)$$

$$\leq -\lambda_2\|y\|^2 + C_{T,k}(\|(x,v)\|^2 + 1),$$

which concludes the proof. $\qquad\square$

## C.2 Challenges of extension to heavy-tailed diffusions

When considering heavy-tailed diffusions, that is, when the base distribution $\nu$ is a Student's t distribution, the stochastic slow-fast systems (4) and (6) could still be used to derive sampling algorithms. However, Theorem 4.1 showing convergence to the averaged dynamics does not apply in this setting, as it relies on exponential ergodicity of the frozen process. In the case where the fast process follows overdamped Langevin dynamics targeting a heavy-tailed distribution, convergence is sub-exponential or polynomial rather than exponential [Wang, 2006, Chapter 4], and thus the frozen process fails to meet the required condition.

As discussed in Section 3.1, instead of using overdamped Langevin dynamics for the fast process, an alternative is to consider different diffusion processes that target the conditional distribution $\rho_{t,x}$ more efficiently. Motivated by the work of He et al. [2024a], we propose employing a natural Itô diffusion that arises in the context of the weighted Poincaré inequality, which has the following expression

$$\mathrm{d}Y_t = -\frac{1}{\varepsilon}(\alpha + d - 1)\nabla U_{\hat{\rho}_{t,X_t}}(y)\mathrm{d}t + \sqrt{\frac{2U_{\hat{\rho}_{t,X_t}}}{\varepsilon}}\mathrm{d}\tilde{B}_t, \tag{25}$$

$$\hat{\rho}_{t,X_t}(y) \propto \left( \sqrt{1 + \frac{\|x-y\|^2}{\alpha(1-\lambda_{\kappa t})}} \pi \left( \frac{y}{\sqrt{\lambda_{\kappa t}}} \right)^{-\frac{1}{\alpha+d}} \right)^{-(\alpha+d)} = U_{\hat{\rho}_{t,X_t}}(y)^{-(\alpha+d)},$$

where $\alpha$ denotes the tail index of the noising distribution $\nu$. This results into the following slow-fast system in the underdamped setting

$$
\begin{cases}
dX_t = & M^{-1}V_t dt \\
dV_t = & \begin{cases} \left( \frac{1}{\sqrt{1-\lambda_{\kappa t}}} \nabla \log \nu \left( \frac{X_t - Y_t}{\sqrt{1-\lambda_{\kappa t}}} \right) - \Gamma M^{-1}V_t \right) dt + \sqrt{2\Gamma} dB_t & \text{if } \lambda_{\kappa t} < \tilde{\lambda} \\ \left( -\frac{1}{\sqrt{\lambda_{\kappa t}}} \nabla V_\pi \left( \frac{Y_t}{\sqrt{1-\lambda_{\kappa t}}} \right) - \Gamma M^{-1}V_t \right) dt + \sqrt{2\Gamma} dB_t & \text{if } \lambda_{\kappa t} \geq \tilde{\lambda} \end{cases} \\
dY_t = & -\frac{1}{\varepsilon}(\alpha+d-1) \nabla U_{\hat{\rho}_{t,X_t}}(y) dt + \sqrt{\frac{2U_{\hat{\rho}_{t,X_t}}}{\varepsilon}} d\tilde{B}_t.
\end{cases}
\tag{26}
$$

However, although the modified Itô diffusion (25) offers improved convergence properties [He et al., 2024a], it does not guarantee exponential ergodicity of the frozen process under mild conditions. As a result, assumption **A4** is not satisfied in this setting. We leave the analysis of the convergence of (26) to the averaged dynamics (5) for future work.

### C.3   Bias of the annealed Langevin dynamics

We formally quantify the bias introduced by the underdamped averaged system (5) relative to the true diffusion path which is given by $(\hat{p}_t)_{t \in [0,1/\kappa]}$ with

$$\hat{p}_t(x,v) \propto \exp\left( -\frac{1}{2}v^\intercal M^{-1}v + \log \hat{\mu}_t(x) \right). \tag{27}$$

**Theorem C.2** (Theorem 4.2 restated). *Let $\mathbb{Q}_{\text{U-ALD}} = (q_{t,\text{U-ALD}})_{t \in [0,1/\kappa]}$ be the path measure of the diffusion annealed Langevin dynamics (5), and $\mathbb{Q} = (\hat{p}_t)_{t \in [0,1/\kappa]}$ that of a reference SDE such that the marginals at each time have distribution $\hat{p}_t$ defined in (27). If $q_{0,\text{U-ALD}} = \hat{p}_0$, the KL divergence between the path measures is given by*

$$\mathrm{KL}\left(\mathbb{Q} \,||\, \mathbb{Q}_{\text{U-ALD}}\right) = \frac{\kappa}{4}\mathcal{A}(p) = \frac{\kappa}{4}\mathcal{A}(\mu).$$

*Proof.* Let $\mathbb{Q}$ be the path measure corresponding to the following reference SDE

$$
\begin{cases}
d\bar{Z}_t = \left( M^{-1}\bar{U}_t + \hat{v}_{1,t}(\bar{Z}_t, \bar{U}_t) \right) dt \\
d\bar{U}_t = \left( \nabla \log \hat{\mu}_t(\bar{Z}_t) - \Gamma M^{-1}\bar{U}_t + \hat{v}_{2,t}(\bar{Z}_t, \bar{U}_t) \right) dt + \sqrt{2\Gamma} dB_t,
\end{cases} \quad \text{for } t \in [0,1/\kappa].
$$

The vector field $\hat{v} = ((\hat{v}_{1,t}, \hat{v}_{2,t}))_{t \in [0,1/\kappa]}$ is designed such that $(\bar{Z}_t, \bar{U}_t) \sim \hat{p}_t$ for all $t \in [0,1/\kappa]$. Using the Fokker-Planck equation, we have that

$$
\begin{aligned}
\partial_t \hat{p}_t &= -\nabla_x \cdot \left( \hat{p}_t \left( M^{-1}v + \hat{v}_{1,t} \right) \right) - \nabla_v \cdot \left( \hat{p}_t \left( \nabla \log \hat{\mu}_t - \Gamma M^{-1}v + \hat{v}_{2,t} \right) \right) + \Gamma \Delta_v \hat{p}_t \\
&= -\nabla_x \cdot (\hat{p}_t \hat{v}_{1,t}) - \nabla_v \cdot (\hat{p}_t \hat{v}_{2,t}), \quad t \in [0,1/\kappa].
\end{aligned}
$$

This implies that $\hat{v}_t = (\hat{v}_{1,t}, \hat{v}_{2,t})$ satisfies the continuity equation and hence generates the curve of probability measures $(\hat{p}_t)_t$. Leveraging Lemma A.3, we choose $\hat{v}$ to be the one that minimises the $L^2(\hat{p}_t)$ norm, resulting in $\|\hat{v}_t\|_{L^2(\hat{p}_t)} = \left|\dot{\hat{p}}\right|_t$ being the metric derivative. Using the form of Girsanov's theorem given in Lemma A.4 we have

$$
\begin{aligned}
\mathrm{KL}\left(\mathbb{Q} \,||\, \mathbb{Q}_{\text{U-ALD}}\right) &= \frac{1}{4}\mathbb{E}_\mathbb{Q}\left[ \int_0^{1/\kappa} \|\hat{v}_t(\bar{X}_t, \bar{V}_t)\|^2 dt \right] = \frac{1}{4}\int_0^{1/\kappa} \|\hat{v}_t(\bar{X}_t, \bar{V}_t)\|_{L^2(\hat{p})}^2 dt = \frac{1}{4}\int_0^{1/\kappa} \left|\dot{\hat{p}}\right|_t^2 dt \\
&= \frac{\kappa}{4}\int_0^1 |\dot{p}|_t^2 dt = \frac{\kappa}{4}\mathcal{A}(p),
\end{aligned}
$$

where we have used that $\left|\dot{\hat{p}}\right|_t = \kappa |\dot{p}|_t$ and the change of variable formula.

To conclude, we note from (27) that the position $x$ and the velocity $v$ variable are independent, and that the marginal distribution of the velocity $v$ remains constant along the diffusion path $(\hat{p}_t)_{t \in [0,1]}$. As a result, the metric derivative simplifies to

$$|\dot{p}|_t = \lim_{\delta \to 0} \frac{W_2(p_{t+\delta}, p_t)}{|\delta|} = \lim_{\delta \to 0} \frac{W_2(\mu_{t+\delta}, \mu_t)}{|\delta|} = |\dot{\mu}|_t.$$

Consequently, the action $\mathcal{A}(p)$ satisfies

$$\mathcal{A}(p) = \mathcal{A}(\mu),$$

where $\mu = (\mu_t)_{t \in [0,1]}$ is the diffusion path defined in (1). $\qquad\square$

Finally note that from the data processing inequality, it follows that the KL divergence between the marginals at final time is bounded by $\mathrm{KL}\left(\hat{p}_{1/\kappa} \,||\, q_{1/\kappa,\text{U-DALD}}\right) \leq \mathrm{KL}\left(\mathbb{Q} \,||\, \mathbb{Q}_{\text{U-ALD}}\right)$.

By Cordero-Encinar et al. [2025, Lemmas 3.3 and 4.2], the action $\mathcal{A}(\mu)$ is bounded when the base distribution $\nu$ is either a Gaussian or a Student's t distribution, provided that the target distribution $\pi$ has finite second order moment and the annealing schedule $\lambda_t$ satisfies the following assumption.

**A5.** *Let $\lambda_t : \mathbb{R}^+ \to [0,1]$ be non-decreasing in $t$ and weakly differentiable, such that if $\nu \sim \mathcal{N}(0, I)$ there exists a constant $C_\lambda$ satisfying either of the following conditions*

$$\max_{t \in [0,T]} |\partial_t \log \lambda_t| \leq C_\lambda$$

*or*

$$\max_{t \in [0,T]} \left| \frac{\partial_t \lambda_t}{\sqrt{\lambda_t(1 - \lambda_t)}} \right| \leq C_\lambda, \tag{28}$$

*or if $\nu$ follows a Student's $t$ distribution, then condition (28) holds for some constant $C_\lambda$.*

# D   Experiment details

The code to reproduce our experiments is available at `https://github.com/paulaoak/sampling_by_averaging.git`. All experiments were implemented using JAX.

## D.1   Target distributions

**Mixture of Gaussians (MoG) and mixture of Student's t (MoS) distributions**   The 8 mixture of Gaussians distribution (8-MoG) consists of 8 equally weighted Gaussian distributions with mean $m_i = 10 \times (1 + \cos(2\pi i/8), 1 + \sin(2\pi i/8))$ for $i \in \{9, \ldots, 7\}$ and covariance $0.7 I_2$. We have shifted the distribution instead of considering the usual benchmark centred at 0 to make sampling more challenging. We note that some of the baselines get stuck in modes close to the initial standard Gaussian distribution, unlike our methods.

For the mixture of 40 Gaussians in dimensions 2 and 50, the modes are equally weighted, with means sampled uniformly from a hypercube of side length 40, and all the covariance matrices set to the identity matrix $I$.

The mixture of Student's t distributions consists of 10 standard Student's t distributions, each with 2 degrees of freedom ($t_2$). Following [Blessing et al., 2024, Chen et al., 2025], the mean of each component is sampled uniformly from a hypercube with a side length of 10. We evaluate this benchmark in dimensions 2, 10 and 50.

**Rings**   This distribution is defined by the inverse polar reparameterisation of a distribution $p_z$ which has itself a decomposition into two univariate marginals: $p_r$ and $p_\theta$. The radial component $p_r$ is a mixture of 4 Gaussian distributions $\mathcal{N}(i + 1, 0.15^2)$ for $i \in \{0, \ldots, 3\}$ describing the radial positions. The angular component $p_\theta$ is a uniform distribution over the interval $[0, 2\pi]$.

**Funnel**   The density of this distribution is given by $\pi(x) \propto \mathcal{N}(x_1; 0, \eta^2) \prod_{i=2}^{d} \mathcal{N}(x_i; 0, e^{x_1})$ for $x = (x_i)_{i=1}^{10} \in \mathbb{R}^{10}$ with $\eta = 3$ [Neal, 2003].

**Double well potential (DW)**   The unnormalised density of the $d$-dimensional DW is given by $\pi \propto \exp(-\sum_{i=1}^{m}(x_i^2 - \delta)^2 - \sum_{i=m+1}^{d} x_i^2)$ with $m \in \mathbb{N}$ and a separation parameter $\delta \in (0, \infty)$. This distribution has $2m$ modes, and larger values of $\delta$ make sampling more challenging due to higher energy barriers. Ground truth samples are obtained using rejection sampling with a Gaussian mixture proposal distribution [Midgley et al., 2023].

**Bayesian logistic regression examples**  We consider binary classification problems on two benchmark datasets: Ionosphere (dimension 35) and Sonar (dimension 61). Given a training dataset $\mathcal{D} = \{(x_j, y_j)\}_{j=1}^N$ where $x_j \in \mathbb{R}^d$ and $y_j$, the posterior distribution over the model parameters $(w, b)$, $w \in \mathbb{R}^d$ and $b \in \mathbb{R}$, is defined by

$$p(w, b|\mathcal{D}) \propto p(w, b) \prod_{j=1}^N \text{Bernoulli}(y, \ \sigma(w^\intercal x + b)),$$

where $\sigma$ denotes the sigmoid function. Following [Grenioux et al., 2024], we place independent Gaussian priors on the parameters $p(w, b) = \mathcal{N}(w; 0, I_d)\mathcal{N}(b; 0, 2.5^2)$.

**Statistical physics model: $\phi^4$ distribution**  The goal is to sample metastable states of the stochastic Allen-Cahn equation $\phi^4$ model (dimension 100) [Albergo et al., 2019, Gabrié et al., 2022]. The $\phi^4$ model is a continuous relaxation of the Ising model which is used to study phase transitions in statistical mechanics. Following [Albergo et al., 2019, Gabrié et al., 2022, Grenioux et al., 2024], we consider a version of the model discretised on a 1-dimensional grid of size $d = 100$. Each configuration of the model is represented by a $d$-dimensional vector $(\phi_i)_{i=1}^d$. We clip the field to 0 at the boundaries by defining $\phi_0 = \phi_{d+1} = 0$. The negative log-density of the distribution is defined as

$$\ln \pi_h(\phi) = -\beta \left( \frac{ad}{2} \sum_{i=1}^{d+1} (\phi_i - \phi_{i+1})^2 - \frac{1}{4ad} \sum_{i=1}^d (1 - \phi_i^2)^2 + h\phi_i \right).$$

We fix the parameters $a = 0.1$ and $\beta = 20$ to ensure a bimodal regime, and vary the value of $h$. We denote by $w_+$ the statistical occurrence of configurations with $\phi_{d/2} > 0$ and $w_-$ the statistical occurrence of configurations with $\phi_{d/2} < 0$. At $h = 0$, the distribution is invariant under the symmetry $\phi \to -\phi$, so we expect $w_+ = w_-$. For $h > 0$, the negative mode becomes dominant.

When the dimension $d$ is large, the relative probabilities of the two modes can be estimated by Laplace approximations at 0-th and 2-nd order. Let $\phi_+^h$ and $\phi_-^h$ denote the local maxima of the distribution, these approximations yield respectively

$$\frac{w_-}{w_+} \approx \frac{\pi_h(\phi_-^h)}{\pi_h(\phi_+^h)}, \qquad \frac{w_-}{w_+} \approx \frac{\pi_h(\phi_-^h) \times |\det H_h(\phi_-^h)|^{-1/2}}{\pi_h(\phi_+^h) \times |\det H_h(\phi_+^h)|^{-1/2}},$$

where $H_h$ is the Hessian of the function $\phi \to \ln \pi_h(\phi)$.

For this last benchmark, we only compare our method with SLIPS [Grenioux et al., 2024], as it is the only baseline capable of accurately recovering samples from the target distribution while preserving the correct relative mode weights.

### D.2  Evaluation metrics

To evaluate the quality of the generated samples, we consider the entropy regularised Wasserstein-2 distance [Peyré and Cuturi, 2019], with regularisation parameter $\varepsilon = 0.05$, for distributions where true samples are available. This metric can be efficiently computed in JAX using the OTT library [Cuturi et al., 2022].

For the Funnel benchmark, we assess the sample quality using the Kolmogorov-Smirnov distance. Specifically, we use the sliced version introduced in Grenioux et al. [2023, Appendix D.1].

In the Bayesian logistic regression tasks, performance is measured via the mean predictive log-likelihood, computed as $p(w, b|\mathcal{D}_{\text{test}})$, where $\mathcal{D}_{\text{test}}$ is a held-out test dataset not used during training.

### D.3  Algorithms and hyperparameters

All algorithms are initialised with samples drawn from a standard Gaussian distribution to ensure a fair comparison. This contrasts with the approach in Grenioux et al. [2024], where the initial samples are drawn using prior knowledge of the target distribution via a quantity denoted as $R_\pi$, which is upper bounded by the scalar variance of the target $\pi$. While we observe that initialising from this distribution can improve performance for our methods, such information is typically unavailable in real-world scenarios. Nevertheless, for the $\phi^4$ model, we consider the same implementation for SLIPS as described in Grenioux et al. [2024], given the complexity of this benchmark.

Besides, to ensure consistency, all algorithms use the same number of energy evaluations, as detailed for each benchmark distribution in Table 3.

Table 3: Number of energy evaluations for each benchmark

| | 8-MoG $(d = 2)$ | 40-MoG $(d = 2)$ | 40-MoG $(d = 50)$ | Rings $(d = 2)$ | Funnel $(d = 10)$ |
|---|---|---|---|---|---|
| Energy evaluations | $3 \times 10^5$ | $3 \times 10^5$ | $5 \times 10^6$ | $3 \times 10^5$ | $5 \times 10^5$ |

| | 5-dim DW $(m = 5, \delta = 4)$ | 10-dim DW $(m = 5, \delta = 3)$ | 50-dim DW $(m = 5, \delta = 2)$ | Ionosphere $(d = 35)$ | Sonar $(d = 61)$ |
|---|---|---|---|---|---|
| Energy evaluations | $3 \times 10^5$ | $5 \times 10^5$ | $5 \times 10^6$ | $1 \times 10^6$ | $3 \times 10^6$ |

| | $\phi^4$ model $(d = 100)$ | 10-MoS $(d = 2)$ | 10-MoS $(d = 10)$ | 10-MoS $(d = 50)$ | |
|---|---|---|---|---|---|
| Energy evaluations | $1 \times 10^8$ | $3 \times 10^5$ | $5 \times 10^5$ | $1 \times 10^6$ | |

**Hyperparameter selection**    For each baseline algorithm, we perform a grid search over a predefined set of hyperparameter values. Selection is based on the corresponding performance metric, computed using 4096 samples. The selected hyperparameters for each algorithm are summarised below.

- The SMC and AIS algorithms define a sequence of annealed distributions $\mu_k$ for $k \in [0, K]$. At each intermediate distribution, we perform $n = 64$ MCMC steps. The parameter $n_{\text{MCMC}}$ is selected from the interval $[20, 100]$ using a grid step of 4. The total number of distributions $K$ is chosen such that the product $n \times K$ matches the number of target evaluations specified in Table 3.

- For ULMC algorithm, we tune three hyperparameters: the mass $M$, the friction coefficient $\Gamma$ and the step size $h$. In all benchmarks, we fix the mass to the identity matrix $M = I$. The step size is chosen within the following grid $h \in \{0.001, \dots, 0.009, 0.01, \dots, 0.09, 0.10, 0.11, \dots, 0.20\}$. The selected values for the step sizes $h$ are provided in Table 4. Additionally, we use a non-constant friction coefficient. Specifically, when running the algorithm for $L$ steps, the friction coefficient remains constant at $\Gamma_{\min} = 1$ for the first $L/2$ steps and then increases linearly from $\Gamma_{\min}$ to $\Gamma_{\max} = 5$ during the remaining steps. The values of $\Gamma_{\min}$ and $\Gamma_{\max}$ are chosen within the grid $\{0.001, 0.005, 0.01, , \dots, 0.09, 0.1, \dots, 0.9, 1, 1.5, \dots, 10\}$.

- For the PT algorithm, the number of chains $K$ is selected from the interval $[1, 20]$ with a grid step of 2. The corresponding temperatures are chosen to be equally spaced on a logarithmic scale, with the minimum temperature fixed at 1 and the maximum temperature chosen from the grid $\{10^2, 10^3, 10^4, 10^5, 10^6, 10^7\}$. In our experiments, we use $K = 5$ parallel chains with temperatures $\{1.0, 5.6, 31.6, 177.8, 1000.0\}$. Each chain employs a Hamiltonian Monte Carlo sampler with $n$ leapfrog steps per sample and a step size $h$ specified in Table 4. The grid for the step size is the same as that of ULMC. The number of leapfrog steps $n$ is chosen so that the total number of target evaluations matches the computational budget defined in Table 3.

Table 4: Selected step sizes for ULMC and PT algorithm across experiments.

| | 8-MoG | 40-MoG 2-dim | 40-MoG 50-dim | Rings | Funnel |
|---|---|---|---|---|---|
| ULMC | 0.05 | 0.05 | 0.01 | 0.05 | 0.03 |
| PT | 0.10 | 0.10 | 0.03 | 0.10 | 0.05 |

| | 5-dim DW | 10-dim DW | 50-dim DW | Ionosphere | Sonar |
|---|---|---|---|---|---|
| ULMC | 0.05 | 0.05 | 0.01 | 0.04 | 0.04 |
| PT | 0.10 | 0.05 | 0.05 | 0.05 | 0.03 |

- The DiGS algorithm uses $K$ noise levels ranging from $\alpha_K$ to $\alpha_1$. The number of noise levels $K$ is selected within the interval $[1, 20]$ with a grid step of 2. The maximum and minimum noise levels, $\alpha_1$ and $\alpha_K$, respectively, are chosen from the grid $\{0.05, \dots, 0.09, 0.1, 0.2, \dots, 1\}$. At each noise level, we perform $n_{\text{Gibbs}}$ Gibbs sweeps, each consisting of $n_{\text{MALA}}$ denoising steps using MALA with step size $h$. The number of Gibbs sweeps $n_{\text{Gibbs}}$ is selected from the interval $[50, 500]$ using a grid step of 50, while the grid for the step size is the same as that of ULMC. The final values of $K$,

$\alpha_K$, $\alpha_1$, $n_{\text{Gibbs}}$, and $h$ are provided in Table 5 for each experiment. The number of MALA steps, $n_{\text{MALA}}$, is determined based on the computational budget.

Table 5: Selected hyperparameters for DiGS algorithm across experiments.

|  | 8-MoG | 40-MoG 2-dim | 40-MoG 50-dim | Rings | Funnel |
|---|---|---|---|---|---|
| $K$ | 3 | 1 | 5 | 1 | 3 |
| $\alpha_K$ | 0.1 | 0.1 | 0.1 | 0.1 | 0.1 |
| $\alpha_1$ | 0.5 | 0.1 | 0.9 | 0.1 | 0.5 |
| $n_{\text{Gibbs}}$ | 200 | 300 | 100 | 300 | 200 |
| $h$ | 0.05 | 0.10 | 0.01 | 0.10 | 0.03 |
|  | **5-dim DW** | **10-dim DW** | **50-dim DW** | **Ionosphere** | **Sonar** |
| $K$ | 1 | 3 | 5 | 3 | 3 |
| $\alpha_K$ | 0.1 | 0.1 | 0.1 | 0.1 | 0.1 |
| $\alpha_1$ | 0.1 | 0.5 | 0.8 | 0.5 | 0.5 |
| $n_{\text{Gibbs}}$ | 300 | 200 | 100 | 200 | 200 |
| $h$ | 0.06 | 0.01 | 0.005 | 0.03 | 0.01 |

- The RDMC algorithm has a hyperparameter $T$, corresponding to the final time of the OU process, its value is provided in Table 6. All other parameters follow the implementation detailed in [Grenioux et al., 2024].

Table 6: Selected hyperparameters for RDMC algorithm across experiments.

|  | 8-MoG | 40-MoG 2-dim | 40-MoG 50-dim | Rings | Funnel |
|---|---|---|---|---|---|
| $T$ | $-\log(0.80)$ | $-\log(0.75)$ | $-\log(0.70)$ | $-\log(0.85)$ | $-\log(0.90)$ |
|  | **5-dim DW** | **10-dim DW** | **50-dim DW** | **Ionosphere** | **Sonar** |
| $T$ | $-\log(0.70)$ | $-\log(0.70)$ | $-\log(0.75)$ | $-\log(0.95)$ | $-\log(0.95)$ |

- For the SLIPS algorithm, we adopt the implementation and hyperparameters provided in the original paper [Grenioux et al., 2024]. To ensure a fair comparison with other algorithms initialised from a standard Gaussian distribution, we set the scalar variance parameter $R_\pi = \sqrt{d}$, yielding $\sigma = R_\pi/\sqrt{d} = 1$. While this choice may degrade SLIPS' performance compared to using an optimally tuned $R_\pi$, estimating such a parameter in practical scenarios is often non trivial. Nevertheless, for completeness, we report below the performance of SLIPS with its optimal parameters alongside our algorithms. In this setting, the results are comparable. However, a key advantage of our method is that it does not require estimation of the scalar variance. Moreover, if we initialise our algorithms using the same informed choice $R_\pi$ by setting the base distribution $\nu \sim \mathcal{N}(0, \sigma^2 I)$, with $\sigma = R_\pi/\sqrt{d}$, we observe a performance improvement, particularly in the overdamped regime.

- For all numerical experiments in the main text, we use the underdamped version of our algorithms, as it yields improved performance. For MULTALMC, the required hyperparameters include the schedule function $\lambda_t$, the values of $\lambda_\delta$ and $\tilde{\lambda}$, the mass matrix $M$, $\varepsilon$, the friction coefficient $\Gamma$, the step size $h$, and the number of SROCK steps $s$. The grid for the step size is the same as that of ULMC. The scale separation parameter $\varepsilon$ is chosen within the grid $\{0.001, 0.005, 0.01, \ldots, 0.09, 0.10, 0.11, \ldots, 0.20\}$. The parameter $\tilde{\lambda}$ is chosen from the interval $[0.3, 0.7]$ using a grid step of 0.05. In all experiments, we select $\lambda_\delta = 0.01$, $\tilde{\lambda} = 0.6$, $M = I$, and $s = 5$. Additionally, we consider a time-dependent friction coefficient. Specifically, when running the algorithm for $L$ steps, the friction coefficient remains constant at $\Gamma_{\min}$ for the first $L/2$ steps and then increases linearly from $\Gamma_{\min}$ to $\Gamma_{\max}$ during the remaining steps. The values of $\Gamma_{\min}$ and $\Gamma_{\max}$ are selected within the grid $\{0.001, 0.005, 0.01, \ldots, 0.09, 0.1, \ldots, 0.9, 1, 1.5, \ldots, 10\}$. The mass matrix is set to the identity. It is worth noting that the number of SROCK steps can be reduced, which may result in a small compromise in performance. This effect is further analysed in Appendix E.2. Lastly, the number of iterations is determined based on the computational budget specified in Table 3.

Table 7: Metrics for different benchmarks averaged across 30 seeds. The metric for the mixture of Gaussian (MoG), Rings and the double well potential (DW) is the entropy regularised Wasserstein-2 distance (with regularisation parameter 0.05), the metric for the Funnel is the sliced Kolmogorov-Smirnov distance and the metric for the Bayesian logistic regression on Ionosphere and Sonar datasets is the average predictive posterior log-likelihood on a test dataset. We compare the performance of our algorithms (initialised with a standard Gaussian distribution) with that of SLIPS with an optimal value of the parameter $R_\pi$, refer to as SLIPS $(R_\pi)$.

| Algorithm | 8-MoG ($\downarrow$) $(d=2)$ | 40-MoG ($\downarrow$) $(d=2)$ | 40-MoG ($\downarrow$) $(d=50)$ | Rings ($\downarrow$) $(d=2)$ | Funnel ($\downarrow$) $(d=10)$ |
|---|---|---|---|---|---|
| SLIPS $(R_\pi)$ | $0.66 \pm 0.11$ | $0.98 \pm 0.06$ | $20.85 \pm 0.63$ | $0.19 \pm 0.02$ | $\mathbf{0.029 \pm 0.007}$ |
| MULTALMC | $0.65 \pm 0.07$ | $\mathbf{0.91 \pm 0.04}$ | $20.58 \pm 0.71$ | $\mathbf{0.18 \pm 0.02}$ | $0.032 \pm 0.005$ |
| MULTCDIFF | $\mathbf{0.62 \pm 0.10}$ | $0.93 \pm 0.06$ | $\mathbf{20.13 \pm 0.59}$ | $0.19 \pm 0.03$ | $0.031 \pm 0.005$ |

| Algorithm | 5-dim DW ($\downarrow$) $(m=5, \delta=4)$ | 10-dim DW ($\downarrow$) $(m=5, \delta=3)$ | 50-dim DW ($\downarrow$) $(m=5, \delta=2)$ | Ionosphere ($\uparrow$) $(d=35)$ | Sonar ($\uparrow$) $(d=61)$ |
|---|---|---|---|---|---|
| SLIPS $(R_\pi)$ | $2.05 \pm 0.30$ | $4.45 \pm 0.36$ | $14.99 \pm 0.59$ | $-86.72 \pm 0.10$ | $-109.38 \pm 0.12$ |
| MULTALMC | $2.08 \pm 0.16$ | $4.48 \pm 0.40$ | $14.03 \pm 0.63$ | $-86.85 \pm 0.09$ | $\mathbf{-109.05 \pm 0.13}$ |
| MULTCDIFF | $\mathbf{1.95 \pm 0.24}$ | $\mathbf{4.23 \pm 0.37}$ | $\mathbf{13.98 \pm 0.56}$ | $\mathbf{-86.33 \pm 0.10}$ | $-109.60 \pm 0.21$ |

Table 8: Selected hyperparameters for MULTALMC algorithm across experiments.

| | 8-MoG | 40-MoG 2-dim | 40-MoG 50-dim | Rings | Funnel |
|---|---|---|---|---|---|
| $\lambda_t$ | Linear | Linear | Linear | Linear | Cosine-like |
| $\varepsilon$ | 0.10 | 0.05 | 0.05 | 0.10 | 0.07 |
| $\Gamma_{\min}, \Gamma_{\max}$ | 0.07, 0.5 | 0.01, 0.5 | 0.01, 0.5 | 0.1, 0.5 | 0.01, 0.5 |
| $h$ | 0.005 | 0.005 | 0.001 | 0.005 | 0.003 |

| | 5-dim DW | 10-dim DW | 50-dim DW | Ionosphere | Sonar |
|---|---|---|---|---|---|
| $\lambda_t$ | Cosine-like | Cosine-like | Cosine-like | Cosine-like | Cosine-like |
| $\varepsilon$ | 0.10 | 0.07 | 0.05 | 0.10 | 0.10 |
| $\Gamma_{\min}, \Gamma_{\max}$ | 0.01, 0.5 | 0.01, 0.5 | 0.01, 0.5 | 0.1, 0.5 | 0.1, 0.5 |
| $h$ | 0.005 | 0.001 | 0.001 | 0.005 | 0.005 |

| | $\phi^4$ model | 10-MoS 2-dim | 10-MoS 10-dim | 10-MoS 50-dim | |
|---|---|---|---|---|---|
| $\lambda_t$ | Cosine-like | Linear | Linear | Linear | |
| $\varepsilon$ | 0.05 | 0.10 | 0.10 | 0.05 | |
| $\Gamma_{\min}, \Gamma_{\max}$ | 0.01, 0.5 | 0.1, 0.5 | 0.1, 0.5 | 0.1, 0.5 | |
| $h$ | 0.001 | 0.005 | 0.005 | 0.001 | |

On the other hand, for MULTCDIFF, the following hyperparameters need to be specified: the friction coefficient $\Gamma$, the scale separation parameter $\varepsilon$, the final time $T$ of the OU process, the time discretisation $0 = T_0 < \cdots < T_L = T$, and the number of SROCK steps $s$. The values of $\Gamma$, $\varepsilon$ and $s$ are set to be the same as in the MULTALMC algorithm. The time discretisation is chosen such that the difference $\lambda_{T_{l+1}} - \lambda_{T_l}$ is constant, where $\lambda_t$ denotes the OU schedule. The number of iterations $L$ is determined based on the computational budget for each benchmark.

## D.4 Computation time

All experiments were conducted on a GPU server consisting of eight Nvidia GeForce RTX 3090 Ti GPU cards, 896 GB of memory and 14TB of local on-server data storage. Each GPU has 10496 cores as well as 24 GB of memory.

Recall that to ensure a fair comparison, we fixed the number of energy evaluations across all algorithms. As a result, the runtime of ULMC, DiGS, RDMC, SLIPS, and our proposed methods, MULTALMC and MULTCDIFF, are broadly similar. In contrast, algorithms such as SMC, AIS, and PT exhibit longer runtimes due to their accept/reject steps. This is demonstrated in Figure 2,

which shows average computation times (over 30 random seeds) for each method on the 40-MoG benchmark in 50 dimensions.

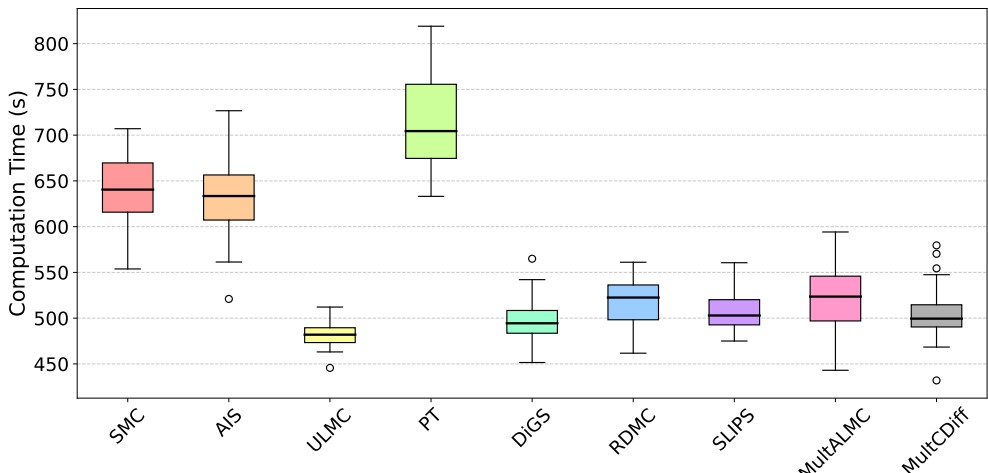

Figure 2: Boxplots of computation times for different algorithms on the 40-MoG benchmark in 50 dimensions, averaged over 30 random seeds.

In addition, we analyse how the runtime of our methods scales with dimensionality. To that end, Table 9 includes performance metrics, number of energy evaluations, and computation times for the 40-MoG benchmark across different dimensions, averaged over 30 random seeds. The performance metric used is the entropy-regularised Wasserstein-2 distance, $W_2^\delta$, with regularisation parameter $\delta = 0.05$.

Table 9: Number of energy evaluations, entropy regularised Wasserstein-2 distance (with regularisation parameter 0.05) and computations times for our sampling methods, MULTALMC (Algorithm 3) and MULTCDIFF (Algorithm 4) evaluated on the 40-MoG benchmark across different dimensions. The results are averaged over 30 random seeds.

|  |  | **d = 2** | **d = 5** | **d = 10** | **d = 20** | **d = 50** |
|---|---|---|---|---|---|---|
| **Energy evaluations** |  | $3 \times 10^5$ | $6 \times 10^5$ | $1 \times 10^6$ | $2 \times 10^6$ | $5 \times 10^6$ |
| $\mathbf{W_2^\delta}$ | MULTALMC | $0.91 \pm 0.04$ | $1.64 \pm 0.12$ | $4.10 \pm 0.27$ | $8.76 \pm 0.34$ | $20.58 \pm 0.71$ |
|  | MULTCDIFF | $0.93 \pm 0.06$ | $1.59 \pm 0.20$ | $3.92 \pm 0.36$ | $8.68 \pm 0.41$ | $20.13 \pm 0.59$ |
| **Time (s)** | MULTALMC | $35 \pm 5$ | $57 \pm 9$ | $111 \pm 13$ | $223 \pm 28$ | $517 \pm 28$ |
|  | MULTCDIFF | $33 \pm 6$ | $58 \pm 7$ | $104 \pm 15$ | $231 \pm 21$ | $502 \pm 30$ |

# E  Additional numerical experiments

## E.1  Comparison of overdamped and underdamped dynamics

As noted in the main text, the overdamped version of MULTALMC requires a small value of $\kappa$, which corresponds to a slowly varying dynamics driven by $\nabla \log \hat{\mu}_t$ to perform well in practice. However, this leads to a large number of discretisation steps since we use a small step size with respect to this slowly changing dynamics resulting in high computational costs. In contrast, the underdamped version achieves better performance without the need for such small step sizes, thanks to the faster convergence properties of underdamped dynamics [Eberle and Lörler, 2024]. To further analyse this, we compare in Table 10 the performance and number of energy evaluations of the overdamped and underdamped versions of MULTALMC, as specified in Algorithms 2 and 3, respectively, on the 40-MoG benchmark across different dimensions. The results show that the overdamped version

requires approximately an order of magnitude more energy evaluations (and hence time steps) to achieve performance comparable to the underdamped version.

Table 10: Performance metrics on the 40-MoG benchmark across varying dimensions, averaged over 30 runs with different random seeds, along with the number of energy evaluations for the overdamped and underdamped versions of MULTALMC, as defined in Algorithms 2 and 3, respectively.

| MULTALMC | | $d = 2$ | $d = 5$ | $d = 10$ | $d = 20$ | $d = 50$ |
|---|---|---|---|---|---|---|
| **Overdamped** | $\mathbf{W}_2^\delta$ | $1.12 \pm 0.10$ | $1.88 \pm 0.22$ | $4.46 \pm 0.39$ | $9.23 \pm 0.67$ | $21.33 \pm 0.98$ |
| | **# evaluations** | $1 \times 10^6$ | $8 \times 10^6$ | $2 \times 10^7$ | $6 \times 10^7$ | $1 \times 10^8$ |
| **Underdamped** | $\mathbf{W}_2^\delta$ | $0.91 \pm 0.04$ | $1.64 \pm 0.12$ | $4.10 \pm 0.27$ | $8.76 \pm 0.34$ | $20.58 \pm 0.71$ |
| | **# evaluations** | $3 \times 10^5$ | $6 \times 10^5$ | $1 \times 10^6$ | $2 \times 10^6$ | $5 \times 10^6$ |

## E.2 Analysis of the impact of the number of SROCK steps in the discretisation

The number of SROCK steps used in our algorithms—MULTALMC and MULTCDIFF—as implemented in Algorithms 2, 3, and 4, plays a critical role in balancing computational efficiency and numerical performance. Increasing the number of inner SROCK steps generally leads to better performance [Abdulle et al., 2018]; however, this comes at the cost of greater computational overhead. Ideally, we aim to identify the minimum number of steps required to maintain strong performance and computational efficiency.

In the numerical experiments presented in Section 5, we use five SROCK steps. We further analyse the sensitivity of our accelerated methods, MULTALMC (Algorithm 3) and MULTCDIFF (Algorithm 4), to the number of SROCK steps. To this end, we report the entropy regularised Wasserstein-2 distance for the 40-MoG benchmark in 50 dimensions (Table 11) when running our methods with varying numbers of SROCK steps while keeping the number of sampling steps $L$ for the slow process fixed.

Table 11: Performance on the 40-MoG benchmark in 50 dimensions using varying numbers of SROCK steps in the discretisation of MULTALMC and MULTCDIFF, as proposed in Algorithms 3 and 4, respectively. The number of sampling steps $L$ for the slow process is held fixed. The standard deviation is computed over 30 runs with different random seeds. The regularisation parameter for the entropy regularised Wasserstein-2 distance $W_2^\delta$ is set to $\delta = 0.05$.

| | SROCK steps | 3 | 5 | 7 | 9 | 11 |
|---|---|---|---|---|---|---|
| $\mathbf{W}_2^\delta$ | MULTALMC | $21.11 \pm 0.74$ | $20.58 \pm 0.71$ | $20.29 \pm 0.68$ | $20.08 \pm 0.50$ | $19.99 \pm 0.45$ |
| | MULTCDIFF | $20.92 \pm 0.63$ | $20.13 \pm 0.59$ | $20.01 \pm 0.57$ | $19.96 \pm 0.48$ | $19.89 \pm 0.40$ |

## E.3 Stability across hyperparameter values

Our algorithms demonstrate robustness and stability across a broad range of hyperparameter values, as shown in Figures 3 and 4. These results suggest that while some tuning is beneficial, our methods do not require highly sensitive or exhaustive hyperparameter optimisation.

# F   Limitations and future work

We elaborate here on the limitations of our method discussed in Section 6.

First, the theoretical guarantees presented in Section 4 rely on stringent assumptions, such as strong convexity of the potential $V_\pi$ and Lipschitz continuity of the score function of the target $\nabla \log \pi$. Relaxing these assumptions, e.g., to strong convexity outside a compact region or to satisfying weak functional inequalities, could broaden the applicability of our analysis. The former would accommodate target distributions that are multimodal within a compact region and Gaussian-like in the tails, while the latter could capture heavy-tailed distributions.

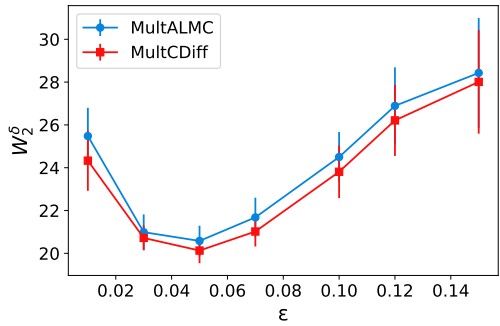
(a) Performance against scale separation parameter $\varepsilon$.

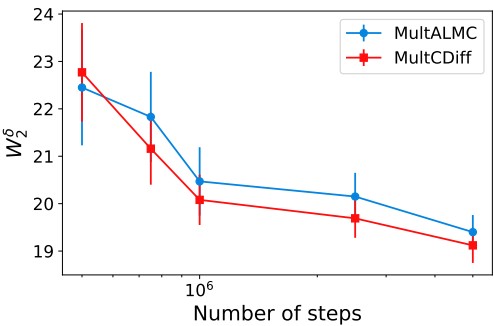
(b) Performance against number of steps.

Figure 3: Ablation results on the 40-component Mixture of Gaussians benchmark in 50 dimensions. Results are averaged over 30 random seeds.

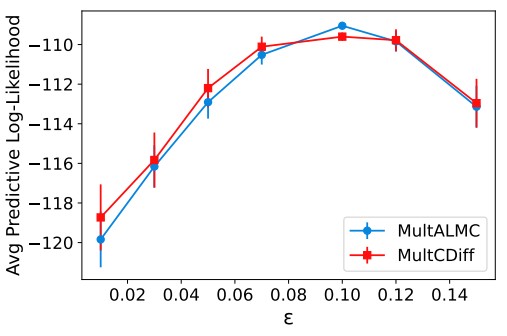
(a) Performance against scale separation parameter $\varepsilon$.

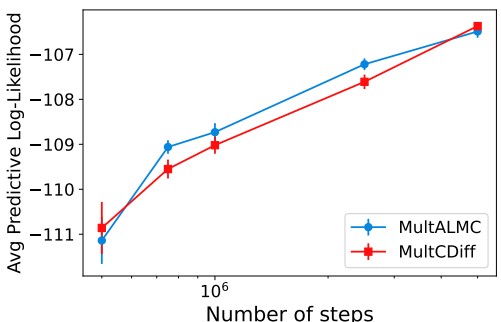
(b) Performance against number of steps.

Figure 4: Ablation results on the Sonar benchmark. Results are averaged over 30 random seeds. The performance metric is the average predictive posterior log-likelihood on a test dataset.

Additionally, the current method requires manual tuning of hyperparameters such as step size $\delta$, scale separation parameters $\varepsilon$ and friction coefficient $\Gamma$. Automating this tuning process, similar to the approach in Blessing et al. [2025], is a valuable direction for improving usability and robustness.

Further research could explore extending the controlled diffusion framework to heavy-tailed target distributions, which pose additional challenges as explained in Appendix B.2.2, or developing more efficient numerical schemes for implementing the proposed multiscale samplers.

# G   Broader impact

This work introduces MULTALMC and MULTCDIFF, two training-free, multiscale diffusion samplers that enable efficient, provably accurate sampling from complex distributions. These methods have the potential to significantly reduce the computational and environmental footprint of generative modelling and Bayesian inference. Potential societal benefits include faster scientific discovery, improved uncertainty quantification, and broader participation in high-impact ML research. As with all general-purpose algorithms, misuse is possible, e.g. lowering the cost of harmful synthetic content generation or accelerating harmful molecule design.

