# OpenReview forum: "Sampling by averaging: A multiscale approach to score estimation"
_NeurIPS.cc/2025/Conference — NeurIPS 2025 poster_

### Official Review · Reviewer_RNkH · 2025-06-23

**Clarity:** 4
**Significance:** 3
**Originality:** 3
**Rating:** 5
**Confidence:** 3

**Summary:**

This work introduces learning-free methods for the sampling of complex unnormalized distributions. The methods aim at estimating the score of the target distribution and are based on the insight that the path of distributions associated to a reverse diffusion process can be expressed as a convolution of the target distribution with a noising distribution. It follows that the score of these distributions is an expectation over a distribution that depends on the noising distribution and the target distribution. The methods rely on the concept of introducing SDEs that represent stochastic processes at different time scales. Simulating these coupled SDEs corresponds to carrying out the aforementioned score estimation (fast process) and the corresponding Langevin dynamics (slow process) which generates samples from the target distribution.

**Questions:**

1. As pointed out in the paper, the multiscale approach avoids the need for a nested algorithm. However, a nested algorithm that does not introduce SDE with different time scales is a natural competitor for the introduced multiscale methods. In such a nested approach, samples ~ $\rho$ in (2) would be drawn in the inner-loop by simulating the corresponding SDE. While some of the methods in the experimental section rely on such a nested approach it is not clear whether any of those corresponds to a variant of the introduced methods where the multiscale SDE simulation is replaced with a nested SDE simulation.

2. How is it ensured that hyperparameter optimization was carried out with comparable computational resources for the compared sampling methods? What were the hyperparameter grids used for each method and benchmark?

**Ethical Concerns:**

["NO or VERY MINOR ethics concerns only"]

**Final Justification:**

The paper represents a valuable contribution to the community. While the presented theoretical contribution may be of limited practical relevance, the strong methodological section and thorough experimental evaluation elevate this work above the bar for publication. It matches all requirements for a publication, and the authors' response to my questions clarified all of my potential concerns.
Concretely, the authors addressed a question concerning a nested competitor method. Their theoretical argumentation and the newly added experimental results provide in total a satisfying response.
Questions about the methodology underlying the hyperparameter tuning and the allocation of computational resources in the experimental evaluation were also answered appropriately. With the additional details provided on the hyperparameter tuning, this work should be well reproducible and it appears to be based on a sound experimental evaluation.

**Limitations:**

yes

**Paper Formatting Concerns:**

-

**Quality:**

3

**Strengths And Weaknesses:**

Strengths:

1. Employing multiscale SDEs for sampling a diffusion path appears to be novel and promising.
2. Well written, clear notation.
3. Experimental evaluation is extensive and the results indicate that the new methods perform well.


Weaknesses:

1. Missing ablations on the heavy-tail distribution in for MULTALMC: the heavy-tail and Gaussian versions of MULTALMC are only compared on the MoS data set but not on any other benchmarks.
2. The theoretical result relies on log-concavity of the energy function which makes it practically hardly relevant. This problem is, however, openly pointed out by the authors.

---

> ### Author Rebuttal · Authors · 2025-07-31
>
> Thank you for your thorough review and helpful comments. We also greatly appreciate your positive feedback.
>
> We reply to your questions below, please let us know if anything is unclear.
>
> > As pointed out in the paper, the multiscale approach avoids the need for a nested algorithm. However, a nested algorithm that does not introduce SDE with different time scales is a natural competitor for the introduced multiscale methods. In such a nested approach, samples ~ $\rho$ in (2) would be drawn in the inner-loop by simulating the corresponding SDE. While some of the methods in the experimental section rely on such a nested approach it is not clear whether any of those corresponds to a variant of the introduced methods where the multiscale SDE simulation is replaced with a nested SDE simulation.
>
> Reverse Diffusion Monte Carlo (RDMC) proposed by Huang et al. (2024) [1] corresponds to the framework that you describe when the averaged process is given by the reverse diffusion process (that is, the setting described in Section 3.2 of our work). We include comparisons of our algorithms with RDMC in Tables 1 and 2.
>
> However, RDMC does not explore the cases where the averaged process is given by the time inhomogeneous Langevin diffusion (i.e., Equation (3) of our work) either for Gaussian diffusion paths, where the base distribution $\nu$ is a Normal distribution, or for heavy-tailed diffusions with a Student’s t base distribution.
>
> The nested inner-outer loop algorithm that you suggest based on annealed (time-inhomogeneous) Langevin diffusion with a Gaussian base distribution will generally lead to a worse performance compared to Reverse Diffusion Monte Carlo. This is due to the bias inherent in annealed Langevin dynamics, as discussed in Section 4.2 of our work.
> Moreover, for completeness we show in the table below, that the nested inner-outer loop algorithm based on annealed Langevin diffusion with a heavy-tailed base distribution underperforms our multiscale approach for the mixture of Student's t distributions benchmarks, when the number of energy evaluations is set to be the same for all methods.
>
> Table 1. Comparison of our MultDALMC method with a nested inner–outer loop algorithm, both using a heavy-tailed base distribution. Results are averaged over 30 random seeds. We evaluate two variants: one based on the modified Itô diffusion process (Equation (7) in our work) and another using an overdamped Langevin diffusion for the fast process and inner loop. Performance is measured by the entropy-regularised Wasserstein-2 distance.
> |        |    $\mathbf{d=2}$   |$\mathbf{d=10}$ | $\mathbf{d=50}$ |
> |----------------|---------|-------------|-------------|
> | **MultALMC w/ modified Itô diffusion**          |   **0.85±0.06** |   **3.42±0.23**   |      **17.21±0.52**    |
> | **Inner-outer   w/ modified Itô diffusion**           |   1.12±0.29   |    5.01±0.57     |      20.39±1.14    |
> | **MultALMC   w/ overdamped Langevin**           |   0.91±0.11   |    3.97±0.28     |      18.56±0.69    |
> | **Inner-outer   w/ overdamped Langevin**          |   1.34±0.20   |    5.62±0.81     |      21.88±1.03    |
>
> > How is it ensured that hyperparameter optimization was carried out with comparable computational resources for the compared sampling methods? What were the hyperparameter grids used for each method and benchmark?
>
> Below, we provide the hyperparameter grids used for our proposed algorithms as well as for all baseline methods. The design of these grids was carefully guided by the original papers introducing each method. We allocated sufficient computational resources to all baselines to ensure that the selected combination of hyperparameters was optimal, some methods required more extensive tuning than others. More specifically, each hyperparameter configuration was evaluated over 30 independent random seeds to account for variability and ensure robustness. When a method exhibited high sensitivity to certain hyperparameters, we expanded the grid around high-performing regions for finer exploration.
>
> Besides, it is important to remark that to ensure a fair comparison we used the same initialisation based on a standard Gaussian distribution for our methods and all baselines in Tables 1 and 2. However, this initialisation is not optimal for the SLIPS algorithm [2], which benefits from a more tailored initialisation. For completeness, we also report the performance of SLIPS with optimal initialisation in Table 7, Appendix D. This optimal initialisation assumes knowledge of the distribution’s scalar variance which is unavailable in practical scenarios.
>
> **Hyperparameter grids for the different algorithms**
> - **SMC and AIS**: The hyperparameter we tune is the number of MCMC steps $n_{\text{MCMC}}$, its value is selected from the interval $[20, 100]$ using a grid step of 4.
> - **ULMC**: The step size is chosen within the following grid $h\in\(0.001,\dots, 0.009, 0.01, \dots, 0.09, 0.10, 0.11, \dots, 0.20\)$. The friction coefficients $\Gamma_{\min}$ and $\Gamma_{\max}$ are chosen within the grid $\(0.001, 0.005, 0.01, , \dots, 0.09, 0.1,\dots, 0.9, 1, 1.5, \dots, 10\)$. The mass matrix is set to the identity.
> - **PT**: The grid for the step size is the same as that of ULMC. The number of chains is selected from the interval $[1, 20]$ with a grid step of 2. The corresponding temperatures are chosen to be equally spaced on a logarithmic scale, with the minimum temperature fixed at 1 and the maximum temperature chosen from the grid $\(10^2, 10^3, 10^4, 10^5, 10^6, 10^7\)$.
> - **DiGS**: The grid for the step size is the same as that of ULMC. The number of noise levels $K$ is selected within the interval $[1, 20]$ with a grid step of 2. The maximum and minimum noise levels, $\alpha_1$ and $\alpha_K$, respectively, are chosen from the grid $\(0.05, \dots, 0.09, 0.1, 0.2, \dots, 1\)$. Finally, the number of Gibbs sweeps $n_{\text{Gibbs}}$ is selected from the interval $[50, 500]$ using a grid step of 50.
> - **RDMC**: It has a single hyperparameter $T$ which we search within the grid $T \in \(0.001, 0.005, 0.01, 0.05, 0.10, 0.15, 0.20, 0.25, 0.30, 0.35, 0.40, 0.45, 0.5\)$ as suggested in [1].
> - **SLIPS**: Following [2], the number of MCMC steps is selected from the interval $[20, 100]$ with a grid step of 2. The value for $t_0$ is chosen by approximate equal log-SNR spacing in $[-5.0, 0.0]$. The value of $\eta$ is selected from the interval $[4,6]$ using a grid step of 0.2.
> - **Ours**: The grid for the step size is the same as that of ULMC. The scale separation parameter $\varepsilon$ is chosen within the grid $\(0.001, 0.005, 0.01, \dots, 0.09, 0.10, 0.11, \dots ,0.20\)$. The friction coefficients $\Gamma_{\min}$ and $\Gamma_{\max}$ are selected within the grid $\(0.001, 0.005, 0.01, , \dots, 0.09, 0.1,\dots, 0.9, 1, 1.5, \dots, 10\)$. The mass matrix is set to the identity. The parameter $\tilde\lambda$ is chosen from the interval $[0.3, 0.7]$ using a grid step of 0.05.
>
> We look forward to engaging further during the rebuttal period to address any remaining concerns you may have. Thanks again!
>
> [1] Huang, X. et al. (2024) Reverse Diffusion Monte Carlo. The Twelfth International Conference on Learning Representations.
>
> [2] Grenioux, L. et al. (2024) Stochastic Localization via Iterative Posterior Sampling. In Proceedings of the 41st International Conference on Machine Learning.

---

> ### Comment · Reviewer_RNkH · 2025-08-01
>
> Thanks for this insightful and detailed response. I would ask the authors to include these detailed grids for the hyperparameter optimization in the appendix of the paper.
> I have no further questions for the authors.

---

> > ### Author Response · Authors · 2025-08-01
> > **Thank you for the prompt response**
> >
> > Thank you for your prompt reply. We will include the detailed hyperparameter grids in Appendix D, as suggested.
> > Thanks again for your feedback and the time you’ve taken to review our work!

---

### Official Review · Reviewer_MkbF · 2025-07-02

**Clarity:** 3
**Significance:** 3
**Originality:** 3
**Rating:** 4
**Confidence:** 1

**Summary:**

This paper proposed a novel multiscale sampling method for score estimation and develop two algorithms based on annealed Langevin dynamics and controlled diffusions, respectively. This paper validate the proposed method on multiple synthetic data with various dimensions. This paper also provide convergence guarantees.

**Questions:**

All of the questions are above.

**Ethical Concerns:**

["NO or VERY MINOR ethics concerns only"]

**Final Justification:**

I'll keep my score.

**Limitations:**

Can this multiscale score estimation method scale up to real-world data.

**Quality:**

4

**Strengths And Weaknesses:**

Strengths:

- Score estimation with multiscale sampling approach is interesting and novel. The proposed method use the multiscale sampling approach to improve the sampleing efficiency and quality.

- This paper provide theoretical guarantees for proposed method.
- The proposed method achieve promising results on low-dimension data.



Weaknesses:

- I'd like to know the results on real-world data, at leaset like mnist or cifar10 because I find the performance on high-dimensional and complex data (table 2) doesn't has consistent improvement.
- Lack of comparison of computational cost.

---

> ### Author Rebuttal · Authors · 2025-07-31
>
> Thank you for your thoughtful review and insightful comments. We also greatly appreciate your positive feedback on our work.
>
> We reply to your concerns below, please let us know if anything is unclear.
>
> > I'd like to know the results on real-world data, at leaset like mnist or cifar10 because I find the performance on high-dimensional and complex data (table 2) doesn't has consistent improvement.
>
> We would like to clarify that our algorithms are designed for **sampling tasks** where we have direct access to a complex unnormalised target distribution and its gradient, and the goal is to generate samples from it. This contrasts with **typical generative modelling tasks** where the objective is to generate new samples from an unknown data distribution, given only access to a dataset of existing samples. To make this distinction clearer, we provide the following table highlighting the key differences between the two tasks.
>
> |                       | **Sampling**                                                                                  | **Generative Modelling**                                                                                               |
> |-----------------------|------------------------------------------------------------------------------------------------|------------------------------------------------------------------------------------------------------------------------|
> | **Available information**     | $\log \pi(x)$ (up to a constant) and $\nabla \log \pi(x) = \nabla V_\pi(x)$                   | Samples/dataset from underlying data distribution $\pi_{\text{data}}(x)$           |
> | **Some typical algorithms**| Markov chain Monte Carlo, sequential Monte Carlo | VAEs, score matching, GANs                      |
>
> Therefore, direct sampling from MNIST or CIFAR10, given access to the dataset of images, does not fit naturally into our framework. To apply our methods to these datasets, one would first need to train an energy-based model [1,2] to learn the unnormalised target distribution corresponding to the dataset and then use our proposed method to draw new samples from the learnt distribution. However, in such a setup, the quality of the generated samples would depend not only on the performance of our sampler but also on how well the energy-based model captures the true data distribution. This would make it difficult to isolate and evaluate the effectiveness of the sampling algorithm itself.
>
> That said, we have evaluated our methods on a wide range of established benchmarks for sampling algorithms [3,4,5] (some of them arising from real-world applications such as Bayesian inference) and included a real-world example involving a statistical physics model which has a dimension of 100 (see Figure 1a).
>
> > Lack of comparison of computational cost.
>
> To **ensure a fair comparison across different algorithms we use the same number of energy evaluations**, as detailed in Table 3 (Appendix D.3). Thus, the computational cost is kept the same for all compared methods.
>
> Additionally, in Appendix D.4, we report the computational resources used, along with the computation times (averaged over 30 random seeds) for each algorithm on the 40-component mixture of Gaussians benchmark in 50 dimensions (see Figure 2).
>
> To further analyse the computational cost, we provide below a comparison in terms of energy evaluations and computation times required by each algorithm on the 40-MoG benchmark (dimension 50) to achieve comparable performance, as measured by the entropy-regularised Wasserstein-2 distance. Results are averaged over 30 random seeds.
>
> Table 1. Comparison of the computational cost across methods on the 40-MoG benchmark in 50-dimensions. Results are averaged over 30 random seeds.
> | Algorithm | $\mathbf{W_2^\delta}$       |   Time (s)   |  Energy evaluations |
> |-----------|------------------|------------------|----------|
> | **SMC** |  25.80±2.03   |  1402±96    |   $1\times10^7$    |
> | **AIS**|  27.14±2.26     | 1327±72     |     $1\times10^7$   |
> | **PT**   | 23.36±0.39      | 593±56    |  $4\times10^6$  |
> | **DiGS**  | 23.68±0.91     | 438±31    |    $4\times10^6$    |
> | **RDMC**  | 24.19±0.82     | 856±40     |  $8\times10^6$  |
> | **SLIPS**  | 23.71±0.65      | 509±27    |  $5\times10^6$  |
> | **MultALMC**  | 23.40±0.45      | 379±35     |  $3\times10^6$  |
> | **MultCDiff**  | **23.31±0.42**     |**363±21**     |  $3\times10^6$  |
>
>
> We note that our proposed algorithms consistently require fewer energy evaluations to achieve comparable performance than alternative methods. We plan to include this discussion in an updated version of the paper.
>
> We hope that this addresses your concerns, and we sincerely request any further feedback that you may be able to provide us during the rebuttal period. Thank you for your continued feedback!
>
> [1] Du, Y. et al. (2021) Improved Contrastive Divergence Training of Energy-Based Model. In Proceedings of the 38th International Conference on Machine Learning.
>
> [2] Song, Y. and Kingma, D. (2021) How to Train Your Energy-Based Models. arXiv:2101.03288.
>
> [3] Grenioux, L. et al. (2024) Stochastic Localization via Iterative Posterior Sampling. In Proceedings of the 41st International Conference on Machine Learning.
>
> [4] Vargas, F. et al. (2024) Transport meets Variational Inference: Controlled Monte Carlo Diffusions. The Twelfth International Conference on Learning Representations.
>
> [5] Richter, L. and Berner, J. (2024) Improved sampling via learned diffusions. The Twelfth International Conference on Learning Representations.

---

> > ### Author Response · Authors · 2025-08-05
> >
> > We hope our response has adequately addressed your concerns. If there are any remaining questions or points you’d like us to clarify, we would be happy to continue the discussion. As before we thank you for your review and valuable comments!

---

> > ### Comment · Reviewer_MkbF · 2025-08-05
> >
> > Thanks for your rebuttal! Thanks for your clarification about classical sampling task and generative modeling task. I'm familiar with mcmc / gibbs / ..., some classic sampling methods and I believe your methods indeed fill the gap in the field of multi-scale sampling methods. And you've solve my concerns at least with my limited knowledge. I'll keep my score.

---

### Official Review · Reviewer_HLYz · 2025-07-03

**Clarity:** 3
**Significance:** 3
**Originality:** 3
**Rating:** 5
**Confidence:** 2

**Summary:**

This paper proposes a multi scale dynamics-based approach to improve the sampling efficiency of score-based samplers.  Existing method typically rely on nested iterative MCMC frameworks to sample from an unnormalized distribution without access to samples.  To alleviate the complexity of these approaches this paper proposed a slow-fast dynamics based on different SDEs to estimate the score function and different points along a diffusion path without resorting to averaging through MCMC-generated samples as in existing approaches.  Results on several simple examples shows the superior performance of the proposed method.

**Questions:**

1. How does the empirical performance change with the dimensionality of the underlying signal? Do the number of steps and scale separation parameter have to be tuned very carefully as the dimensionality changes?
2. Is there any intuition on how to choose the hyper parameters of the proposed algorithm carefully without too much experimental tuning?  What happens when the conditional distribution mixing time is set too slow or too fast?

**Ethical Concerns:**

["NO or VERY MINOR ethics concerns only"]

**Final Justification:**

I believe that the paper is well written and the analysis is thorough.  Despite not being evaluate on very high dimensional datasets, the experiments are fairly extensive and provide evidence for the viability of the proposed method.

**Limitations:**

Yes

**Paper Formatting Concerns:**

Paper is well written and formatted nicely

**Quality:**

4

**Strengths And Weaknesses:**

**Strengths**

1. The goal of the paper and the proposed method is clear.  The background and related work is thorough and well written.
2. The proposed method with parallel slow-fast dynamics is fairly novel and addresses the computational burden of existing approaches.
3. The drawbacks of each proposed scheme are explained and it helps motivate additional improvements with subsequently proposed algorithms
4. The experimental evaluation is strong and show the competitive performance of the proposed scheme.

**Weaknesses**

1. The practical viability of the approach to some real-world settings is not immediately clear given the constraints and assumptions placed within the theoretical analysis.

---

> ### Author Rebuttal · Authors · 2025-07-31
>
> Thanks a lot for your thoughtful review and helpful comments. We also greatly appreciate your positive remarks on our work.
>
> > The practical viability of the approach to some real-world settings is not immediately clear given the constraints and assumptions placed within the theoretical analysis.
>
> Our algorithm is specifically designed to sample from *diffusion paths* with the aim of targeting highly complex probability distributions in practice. In our experiments, we demonstrate that the proposed algorithms exhibit strong empirical performance in general settings beyond strong convexity. This includes a real-world application in statistical physics involving a 100-dimensional system (see Figure 1a), as well as benchmarks from Bayesian inference tasks such as Ionosphere and Sonar.
>
> We acknowledge that the assumptions made in the theoretical section are arguably strong. Our objective in providing this theoretical guarantee was to demonstrate that the method is principled under suitable conditions. Moreover, the analysis serves as a template that can be extended to more general assumptions in future work.
>
> Analysing multiscale systems under weaker assumptions presents significant technical challenges and constitutes a substantial line of research in its own right. For this reason, it falls outside the scope of the present work. Nevertheless, we hope that the connections we establish with the averaging literature will encourage applied mathematicians working on multiscale dynamics to engage with this problem. We are genuinely excited and eager to see what theoretical developments our initial approach may inspire in future work.
>
> > How does the empirical performance change with the dimensionality of the underlying signal? Do the number of steps and scale separation parameter have to be tuned very carefully as the dimensionality changes?
>
> As the dimension of the underlying signal (i.e., the $(Y_t)\_{t\geq 0}$ process), and consequently the dimension of the $(X_t)_{t\geq 0}$ process, increase, convergence to the target distribution becomes slower, requiring a larger number of time steps. However, convergence still occurs. This is illustrated in Table 9 (Appendix D.4), where we report performance metrics (specifically, the entropy-regularised Wasserstein-2 distance) alongside the number of energy evaluations (which is in one-to-one correspondence with the number of steps) and computations times for our sampling methods on the 40-MoG benchmark across different dimensions.
>
> On the other hand, we observed that **our algorithms demonstrate robustness and stability across a broad range of hyperparameter values**, as shown in the table below. We plan to include a plot (extending the results of this table) on a finer hyperparameter grid in an updated version of the paper. These results suggest that while some tuning is beneficial, our methods do not require highly sensitive or exhaustive hyperparameter optimisation.
>
> Table 1. Ablation results on the 40-component Mixture of Gaussians benchmark in 50 dimensions for the scale separation parameter $\varepsilon$ and the number of steps. Results are averaged over 30 random seeds. Notably, the performance remains stable across a wide range of hyperparameter values.
> |        |    $\mathbf{\varepsilon}$   | 0.01   | 0.03 | 0.05 | 0.07   | 0.10      | 0.12|0.15|
> |----------------|---------|-------------|---------------|---------|-------------|-------------|-------------|-------------|
> | **MultALMC**          | $W_2^\delta$  |   25.48±1.32    |    20.99±0.83    |   20.58±0.71|  21.68±0.92|  24.50±1.17| 26.89±1.80| 28.43±2.57|
> | **MultCDiff**             | $W_2^\delta$    |    24.33±1.41     |     20.72±0.58     |    20.13±0.59 |    21.02±0.70     |   23.81±1.23| 26.21±1.66| 28.01±2.42|
>
> |        |    Number of steps  | $\mathbf{5\times 10^5}$   | $\mathbf{7.5\times 10^5}$ | $\mathbf{1\times 10^6}$ | $\mathbf{2.5\times 10^6}$   | $\mathbf{5\times 10^6}$ |
> |----------------|---------|-------------|---------------|---------|-------------|-------------|
> | **MultALMC**          | $W_2^\delta$  |   22.45±1.22    |    21.83±0.95    |   20.47±0.72| 20.15±0.50|  19.40±0.36|
> | **MultCDiff**             | $W_2^\delta$    |    22.77±1.04    |     21.16±0.76     |   20.08±0.53 |    19.69±0.41     |   19.12±0.37|
>
>
> Table 2. Ablation results on the Sonar benchmark for the scale separation parameter $\varepsilon$ and the number of steps. Results are averaged over 30 random seeds. The performance metric is the average predictive posterior log-likelihood on a test dataset. Notably, the performance remains stable across a wide range of hyperparameter values.
> |        |    $\mathbf{\varepsilon}$   | 0.01|0.03 | 0.05 | 0.07   | 0.10      | 0.12| 0.15|
> |----------------|---------|-------------|---------------|---------|-------------|-------------|-------------|-------------|
> | **MultALMC**          | Performance  |   -119.84±1.41    |     -116.16±1.07    |   -112.91±0.83|  -110.52±0.49|  -109.05±0.13| -109.84±0.52| -113.14±1.06|
> | **MultCDiff**             | Performance    |    -118.73±1.67     |     -115.83±1.39     |    -112.20±0.97 |    -110.11±0.51     |   -109.60±0.21| -109.78±0.55| -112.96±1.23|
>
> |        |   Number of steps   |  $\mathbf{5\times 10^5}$   | $\mathbf{7.5\times 10^5}$ | $\mathbf{1\times 10^6}$ | $\mathbf{2.5\times 10^6}$   | $\mathbf{5\times 10^6}$ |
> |----------------|---------|-------------|---------------|---------|-------------|-------------|
> | **MultALMC**          | Performance  |  -111.14±0.52    |     -109.06±0.15    |   -108.73±0.20|  -107.22±0.13|  -106.49±0.14|
> | **MultCDiff**             | Performance    |    -110.86±0.58     |     -109.55±0.21     |   -109.02±0.19 |    -107.61±0.16     |   -106.37±0.09|
>
>
> > Is there any intuition on how to choose the hyper parameters of the proposed algorithm carefully without too much experimental tuning? What happens when the conditional distribution mixing time is set too slow or too fast?
>
> The main hyperparameters in our algorithms are the step size and scale separation parameter ($\varepsilon$). The intuition for choosing the step size is similar to that used in classical Langevin-based algorithms (see [1,2]).In particular, this parameter typically scales inversely with the dimension of the problem. Regarding the scale separation parameter, we empirically observe that values in the range 0.01-0.11 yield good performance across a wide range of problem settings (see Table 8, Appendix D.3, where we report the optimal value of $\varepsilon$ for each benchmark distribution).
>
> If the conditional distribution's mixing time is set too slow, then the obtained samples from the process $Y_t$ will not correspond to that of the correct conditional distribution. This results in the drift term of the slow process ($X_t$) being poorly estimated. That is, the drift term will not accurately reflect the drift term corresponding to the true moving target, which in turn fails to capture the correct dynamics of the underlying moving target. Intuitively, this leads to a lag in tracking the moving target distribution, which may cause some modes of the final target distribution to be missed.
>
> On the other hand, if the conditional distribution mixing time is too fast, the particles from the slow process $(X_t)$ may exhibit overly noisy or "wiggly" trajectories, hindering convergence over time.
>
> We hope that the additional discussions and ablation studies have addressed the reviewer’s questions, and we look forward to receiving further feedback!
>
> [1] Chewi, S. et al. (2024) Analysis of Langevin Monte Carlo from Poincare to Log-Sobolev. Foundations of Computational Mathematics.
>
> [2] Milstein, G. N. (1994) Numerical integration of stochastic differential equations. Springer.

---

> > ### Author Response · Authors · 2025-08-05
> >
> > We hope our response has adequately addressed your concerns. If there are any remaining questions or points you’d like us to clarify, we would be glad to continue the discussion. Once again, thank you for your thoughtful review and valuable feedback.

---

> > > ### Comment · Area_Chair_bPLj · 2025-08-08
> > >
> > > Dear Reviewer HLYz,
> > >
> > > As the author-reviewer discussion period is approaching its end, please review the rebuttal and engage in the discussion promptly. A note confirming your concerns are resolved is critical. Also, non-participating reviewers may face penalties under the Responsible Reviewing Initiative, affecting future invitations.
> > >
> > > Thanks,
> > >
> > > AC

---

> > > ### Comment · Reviewer_HLYz · 2025-08-09
> > > **Thank you for the additional results**
> > >
> > > Dear author(s),
> > >
> > > Thank you for answering my questions.  The answers are detailed and I appreciate the new results that I believe will improve the paper further.  I will maintain my score and recommend acceptance.

---

### Official Review · Reviewer_JLNp · 2025-07-05

**Clarity:** 3
**Significance:** 2
**Originality:** 3
**Rating:** 4
**Confidence:** 4

**Summary:**

This paper considers the problem of sampling using diffusion processes, given oracle access to the gradients of the potential function of the underlying distribution. Their main contribution is a new algorithm based on a multi-scale version of the standard annealed Langevin dynamics. They show that this algorithm provably converges to the true distribution asymptotically, given that the true distribution is strongly log-concave. They complement these theoretical results with some preliminary evaluations both for the Gaussian setting, as well as for the heavy-tailed setting.

**Questions:**

My main question, as described above, is regarding the motivation of the paper. While the authors briefly justify why this method could be preferred to the other works, these concerns do not really hold for the strongly log-concave setting that their theoretical results hold for. If the authors could justify why the results are interesting in comparison to prior work specifically in this strongly log-concave setting, that would be very helpful.

**Ethical Concerns:**

["NO or VERY MINOR ethics concerns only"]

**Final Justification:**

After reading the rebuttal and in discussion with the other reviewers, I am okay with increasing my score.

To clarify, and this was definitely because I was being hasty, I was generally referring to the fact that good, efficient rates are known for sampling from strongly log-concave and smooth distributions under log-grad access, but indeed, the papers I referenced are not quite the right references for this. Regardless, as the authors point out, the novelty of the paper must be for these more complex, empirical settings.

I am not an expert on the state-of-the-art in this regard, but after discussion it seems like these benchmarks are acceptable, albeit not ideal, for this area, which is surprising to me. For this reason, though, I am okay with raising the score.

**Limitations:**

yes

**Quality:**

3

**Strengths And Weaknesses:**

Quality: From a technical point of view, the results in this paper are quite nontrivial and above the bar for acceptance.

Clarity: The results in the paper are presented clearly, at least within the main text of the paper.

Significance: This is my main concern for the paper. Namely, it is by now fairly well understood how to sample from strongly log-concave distributions given this sort of oracle access, e.g. the papers that the authors cite of Chen-Chewi-Salim-Wibisono and Lee-Shen-Tian. Moreover, these papers yield non-asymptotic rates, as opposed to the asymptotic convergence demonstrated here. Despite the paper’s claims, I think it is unlikely that this approach will work for complex real-world settings (as it doesn’t learn a score function). Indeed, the evaluations in the paper are on simple, synthetic settings. Thus, I think the main motivation must be theoretical, but the results are then largely superceded by these previous works.

Originality: The method is original, to my understanding.

---

> ### Author Rebuttal · Authors · 2025-07-31
>
> Thank you for your review and insightful comments. We also sincerely appreciate your positive remarks on the quality and clarity of our work.
>
> We reply to your concerns and questions below, please let us know if anything is unclear.
>
> > This is my main concern for the paper. Namely, it is by now fairly well understood how to sample from strongly log-concave distributions given this sort of oracle access, e.g. the papers that the authors cite of Chen-Chewi-Salim-Wibisono and Lee-Shen-Tian. Moreover, these papers yield non-asymptotic rates, as opposed to the asymptotic convergence demonstrated here.
>
> We are not entirely sure whether, in your comment, oracle refers to access to the gradients of the target potential, i.e. $\nabla \log \pi = \nabla V_\pi$, or instead to what Chen–Chewi–Salim–Wibisono and Lee–Shen–Tian define as a restricted Gaussian oracle (RGO). For completeness, we address both interpretations below.
>
> **1. Oracle = access to $\nabla\log\pi=\nabla V_\pi$**: Our algorithms are designed for sampling tasks, where the goal is to obtain samples from a complex target distribution $\pi$. In this setting, it is assumed in the literature that we have access to $\log \pi(x)$ (up to an additive constant) and its gradient $\nabla \log \pi(x) = \nabla V_\pi(x)$. This contrasts with typical generative modelling tasks where the objective is to generate new samples from an unknown data distribution, given only access to a dataset of existing samples. To make this distinction clearer, we provide the following table highlighting the key differences between the two tasks.
>
> |                       | **Sampling**                                                                                  | **Generative Modelling**                                                                                               |
> |-----------------------|------------------------------------------------------------------------------------------------|------------------------------------------------------------------------------------------------------------------------|
> | **Available information**     | $\log \pi(x)$ (up to a constant) and $\nabla \log \pi(x) = \nabla V_\pi(x)$                   | Samples/dataset from underlying data distribution $\pi_{\text{data}}(x)$           |
> | **Some typical algorithms**| Markov chain Monte Carlo, sequential Monte Carlo | VAEs, score matching, GANs
>
> Having said this, when $\pi$ is strongly log-concave, standard Langevin Monte Carlo (LMC) algorithms can be readily applied, with known convergence guarantees in this setting [1], without requiring the use of the more sophisticated frameworks of Chen–Chewi–Salim–Wibisono or Lee–Shen–Tian, which rely on access to an RGO. However, we would like to emphasise that, when developing our algorithms, our main focus is on complex multimodal distributions, where plain LMC and related methods typically struggle. To address this challenge, we employ methods based on annealed Langevin diffusions, which are better suited for these harder sampling problems.
>
> **2. Oracle as in Chen-Chewi-Salim-Wibisono and Lee-Shen-Tian**: We would like to clarify that our setting does not assume access to an oracle in this sense. The works of Chen-Chewi-Salim-Wibisono and Lee-Shen-Tian rely on access to what they refer to as a restricted Gaussian oracle (RGO), which corresponds in our notation to exact sampling from the conditional distribution $Y\sim\rho_{t,X}(y)$ (or $\pi^{X|Y}$ in their notation). Some of their theoretical results explicitly assume this oracle access and therefore do not account for the complexity of sampling from $Y\sim\rho_{t,X}(y)$, which in our case corresponds to the fast process. Our theoretical analysis does include the complexity of this sampling step.
>
> Besides, while both aforementioned works also provide theoretical results when implementing the oracle approximately, via a rejection sampling scheme (Corollary 6 in Chen-Chewi-Salim-Wibisono and Corollary 1 in Lee-Shen-Tian), these approaches assume access to the minimiser of the target potential—an assumption that can be impractical in higher-dimensional settings.
>
> > Despite the paper’s claims, I think it is unlikely that this approach will work for complex real-world settings (as it doesn’t learn a score function). Indeed, the evaluations in the paper are on simple, synthetic settings. Thus, I think the main motivation must be theoretical, but the results are then largely superceded by these previous works.
>
> We would like to highlight that we do estimate the drift term of the averaged process $\bar X_t$, Equation (3), (which corresponds to the score function of the true moving target distribution $\mu_t$). This estimation is made possible via the auxiliary fast process $Y_t$, which enables us to approximate the drift of the marginal $X_t$ without direct score learning.
>
> Besides, our algorithms demonstrate strong performance across a variety of established benchmarks in the sampling literature that go beyond the strongly log-concave setting (some of them arising from real-world applications such as Bayesian inference). In addition, we include a real-world example given by a statistical physics model which has a dimension of 100 (see Figure 1a). Importantly, this example is not only high-dimensional but also inherently challenging due to the free-energy barrier between the modes [2].
>
> > While the authors briefly justify why this method could be preferred to the other works, these concerns do not really hold for the strongly log-concave setting that their theoretical results hold for. If the authors could justify why the results are interesting in comparison to prior work specifically in this strongly log-concave setting, that would be very helpful.
>
> As we acknowlegde in the discussion section, the strongly-convex assumption required for deriving theoretical guarantees is indeed restrictive. However, the motivation for providing partial theoretical guarantees was to demonstrate that our method is principled and to provide an initial template for more general theory. This does not mean that the proposed methods are restricted to the assumption used in the theoretical analysis. Developing a full theoretical framework for multiscale systems under weaker assumptions remains technically challenging and constitutes a substantial line of research in its own right. For this reason, it falls outside the scope of the present work. Nevertheless, we hope that the connections we establish with the averaging literature will encourage applied mathematicians working on multiscale dynamics to engage with this problem. We are genuinely excited and eager to see what theoretical developments our initial approach may inspire in future work.
>
> **Conclusion**
>
> We sincerely appreciate your recognition that our work is “quite nontrivial and above the bar for acceptance” (as noted in your Quality comment). We also fully acknowledge your concerns regarding the scope of our theoretical results, which are currently confined to the strongly log-concave and asymptotic settings.
>
> That said, we would like to emphasise that while our proof-of-concept theoretical guarantees focus on the strongly log-concave case, our method is explicitly designed for — and evaluated on — real-world scenarios that are non log-concave and high-dimensional. To this end, we **have benchmarked our algorithms against a broad set of state-of-the-art methods on challenging, high-dimensional sampling problems** demonstrating both practicality and competitive performance.
>
> In light of this, we kindly ask you to consider reevaluating your score. We believe that current results in our manuscript provide a comprehensive evaluation of our samplers, including a practical (and highly challenging) statistical physics model.
>
> We look forward to engaging with you further during the rebuttal period to address any remaining concerns and welcome your feedback on our responses. Thanks again!
>
> [1] Durmus, A. et al. (2019) Analysis of Langevin Monte Carlo via Convex Optimization. Journal of Machine Learning Research, 20 (73).
>
> [2] Gabrié, M. et al. (2022) Proc. Natl. Acad. Sci. U.S.A. 119 (10).

---

> > ### Author Response · Authors · 2025-08-05
> >
> > We hope our response has adequately addressed your concerns. If there are any remaining questions or points you’d like us to clarify, we would be happy to continue the discussion. As before we thank you for your review and insightful comments!

---

> > > ### Comment · Area_Chair_bPLj · 2025-08-08
> > >
> > > Dear Reviewer JLNp,
> > >
> > > As the author-reviewer discussion period is approaching its end, please review the rebuttal and engage in the discussion promptly. A note confirming your concerns are resolved is critical. Also, non-participating reviewers may face penalties under the Responsible Reviewing Initiative, affecting future invitations.
> > >
> > > Thanks,
> > >
> > > AC

---

### Comment · Area_Chair_bPLj · 2025-08-04
**Engage in Author-Reviewer Discussions**

Dear reviewers,

If you haven't done so already, please click the 'Mandatory Acknowledgement' button and actively participate in the rebuttal discussion with the authors after carefully reading all other reviews and the author responses.

Thanks,
AC

---

### Note · Authors · 2025-08-12

We thank all reviewers for their constructive feedback. We would like to reiterate some points raised in our rebuttal that did not receive a full response:

- Regarding concerns about the strongly log-concave assumption in our theory and the overall significance of our work, we clarified that:
    - Our methods are designed for challenging multimodal and high-dimensional targets, with experiments extending beyond the strongly log-concave case (including real world examples such as a 100-dimensional statistical physics model). Empirical results confirm competitiveness with state-of-the-art methods.

    - The asymptotic theoretical results in the strongly convex setting serve as a principled foundation and proof-of-concept. Developing a full theoretical framework for multiscale systems under weaker assumptions remains technically challenging and constitutes a substantial research direction beyond the scope of this paper.

In addition, we will incorporate the following experimental details and results that we provided to address specific reviewer comments:
- Ablation studies demonstrating robustness to hyperparameters and scalability with dimension.
- New comparisons to nested inner–outer loop algorithms in the heavy-tailed setting.
- Computational cost analyses showing our methods require fewer energy evaluations for comparable performance.
- Detailed grids for hyperparameter optimisation.

---

### Decision · Program_Chairs · 2025-09-17

**Decision:**

Accept (poster)

**Comment:**

This paper proposes a novel sampling method from a given unnormalized target distribution that is high-dimensional and has many modes through a diffusion path. In specific, it applies a fast stochastic differential equations (SDE) to efficiently estimate the drift term of the averaged intermediate score (target slow SDE) without any training. Experimental results on a number of toy examples and real-world datasets show that the proposed multi-scale dynamics-based sampling (coupled slow and fast SDEs) outperforms baseline samplers.

Overall, the use of multi-scale dynamics for diffusion-based sampling seems to be technically sound and novel. Most concerns raised by the reviewers are well addressed by the authors, including a clear description of the target sampling problem different from generative modeling, a computation cost, hyperparameters’ sensitivity, a limited theoretical analysis on the strict constraint and assumption. Based on the consensus of all positive ratings between the reviewers, I would recommend the paper to be accepted.

However, I think that one main concern regarding the practicality on real-world datasets still needs to be further resolved. For example, given a learned energy function (even though it is not perfectly correct) on mnist or cifar, the sampling performance comparison between the proposed fast inner process and other baselines or variants would be necessary. Moreover, it is necessary to analyze whether the proposed method yields greater benefits as the dimensionality increases or the number of modes grows.

In addition, regarding the paper’s organization, rather than allocating limited theoretical analysis to the main body, it would be more appropriate to present the details of the proposed sampling methods (Algorithms 1 and 2) there and to further strengthen the experimental section.